# COMBINATORIAL BANDIT BAYESIAN OPTIMIZATION FOR TENSOR OUTPUTS

**Jingru Huang[a], Haijie Xu[a], Jie Guo[b], Manrui Jiang[a], Chen Zhang[a]** [*]
[a] Department of Industrial Engineering, Tsinghua University, Beijing 100084, China
[b] College of Economics and Management, Nanjing University of Aeronautics and Astronautics, Nanjing 211106, China
`jingruhuang@tsinghua.edu.cn, xu-hj22@mails.tsinghua.edu.cn,`
`guojie25@nuaa.edu.cn, jiangmanrui@mail.tsinghua.edu.cn,`
`zhangchen01@tsinghua.edu.cn`

## ABSTRACT

Bayesian optimization (BO) has been widely used to optimize expensive and black-box functions across various domains. However, existing BO methods have not addressed tensor-output functions. To fill this gap, we propose a novel tensor-output BO framework. Specifically, we first introduce a tensor-output Gaussian process (TOGP) with two classes of tensor-output kernels as a surrogate model of the tensor-output function, which can effectively capture the structural dependencies within the tensor. Based on it, we develop an upper confidence bound (UCB) acquisition function to select query points. Furthermore, we introduce a more practical and challenging problem setting, termed combinatorial bandit Bayesian optimization (CBBO), where only a subset of the tensor outputs can be selected to contribute to the objective. To tackle this, we propose a tensor-output CBBO method, which extends TOGP to handle partially observed tensor outputs, and accordingly design a novel combinatorial multi-arm bandit-UCB2 (CMAB-UCB2) criterion to sequentially select both the query points and the output subset. We establish theoretical regret bounds for both methods, guaranteeing sublinear regret. Extensive experiments on synthetic and real-world datasets demonstrate the superiority of our methods.

## 1 INTRODUCTION

Bayesian optimization (BO) is a widely used strategy for optimizing expensive, black-box objective functions (Frazier, 2018; Wang et al., 2023). Its effectiveness has led to successful applications in various domains such as hyperparameter tuning, experimental design, and robotics (Snoek et al., 2012; Shields et al., 2021; Wang et al., 2022). Most existing BO methods focus on scalar outputs (Bull, 2011; Wu et al., 2017), while recent studies have extended BO to handle multi-output settings (Chowdhury & Gopalan, 2021; Tu et al., 2022; Maddox et al., 2021; Song et al., 2022). However, to the best of our knowledge, no prior work has addressed BO with tensor-valued outputs, where the system response is a multi-mode tensor. In contrast, tensor-valued data have been extensively explored in other areas, including tensor decomposition (Abed-Meraim et al., 2022), tensor regression (Lock, 2018), and tensor completion (Song et al., 2019), among others.

In current multi-output BO (MOBO) methods, a surrogate model, typically a multi-output Gaussian process (GP) or a collection of independent scalar-output GPs, is constructed from observed data, and an acquisition function is then optimized to sequentially select query points by balancing exploration and exploitation. A straightforward way to handle tensor-valued outputs is to vectorize the tensor and apply existing MOBO methods. However, vectorization ignores the intrinsic structural correlations of tensor data, especially mode-wise dependencies that are critical in many applications. As a result, MOBO methods may become less effective when optimizing the acquisition function to identify the global optimum. As such, it is important to develop GP surrogates that model tensor structure directly. Existing tensor-output GP models typically adopt fully separable covariance

---

[*]Corresponding author.

structures, in which the joint covariance factorizes as a Kronecker product of covariance matrices across the input and each output mode (Kia et al., 2018; Zhe et al., 2019; Belyaev et al., 2015). While computationally attractive, separability implies that correlations across tensor modes do not vary with the input, an assumption that is often violated in complex systems such as spatiotemporal processes (Hristopulos, 2023). This mismatch can lead to poor predictive accuracy, numerical instability in posterior inference, and ultimately degraded BO performance. Therefore, in this paper, we first aim to construct a more flexible and scalable GP model that can capture input-dependent correlations within tensor outputs. Then, we aim to design an acquisition function and a sequential querying policy tailored for tensor-output BO.

Furthermore, we consider a more complex and practical setting in which only a subset of tensor entries contributes to the objective. This naturally transforms the problem into a combinatorial multi-armed bandit (CMAB) setting. Specifically, each tensor element is treated as an base arm, and at each round, select a subset of arms, referred to as a super-arm. The objective value associated with the chosen super-arm is then observed, which we interpret as the reward. We term this novel problem as *combinatorial bandit Bayesian optimization* (CBBO). The goal of tensor-output CBBO is to jointly identify an optimal input in the search space and the corresponding best super-arm over tensor outputs. Recent work has combined BO with multi-armed bandits (MAB), often termed bandit Bayesian optimization (BBO), to handle mixed input spaces with both continuous and categorical variables (Nguyen et al., 2020; Ru et al., 2020; Huang et al., 2022). In these settings, categorical variables can be viewed as indexing discrete modes, and choosing one category per variable amounts to selecting a single arm along each mode, which is a special case of CBBO. However, existing BBO methods do not be directly extend to our CBBO setting for two reasons. First, they typically model the effects of categorical choices using independent GPs, and thus cannot capture the rich structural correlations inherent in tensor outputs. Second, their selection strategies effectively decompose the decision into multiple independent bandit problems (one per categorical variable), whereas CBBO requires joint selection of multiple, potentially correlated arms within a super-arm. Thus, these methods are not well suited for our tensor-output CBBO framework.

To address the aforementioned challenges, we propose a tensor-output Bayesian optimization framework, termed TOBO, together with its extension to the CBBO setting, termed TOCBBO. Our main contributions are summarized as follows:

- We propose a tensor-output Gaussian process (TOGP) with two classes of tensor-output kernels. The proposed kernels explicitly incorporate tensor structure by extending the linear model of coregionalization from vector-valued to tensor-valued outputs, thereby capturing dependencies both across tensor modes and over the input domain.

- Using the TOGP model as the surrogate, we develop a TOBO framework based on the upper confidence bound (UCB) acquisition. We establish a sublinear regret bound for TOBO, which provides the first Bayesian regret analysis tailored to tensor-valued outputs.

- We formulate a novel problem setting, referred to as CBBO. To address this problem, we design the TOCBBO framework by introducing a CMAB-UCB2 acquisition function, which couples UCB-based input selection with CMAB-UCB-based super-arm selection. We further establish a sublinear regret bound for TOCBBO.

- We demonstrate the efficiency of our methods on synthetic experiments and case studies.

Notably, compared with existing TOGP methods (Belyaev et al., 2015; Kia et al., 2018; Zhe et al., 2019), our model provides a more general kernel-construction framework. Specifically, existing tensor-output kernels can be interpreted as special cases induced by particular low-rank tensor decompositions, whereas our LMC-based formulation allows a broad class of tensor constraints to be encoded through the coregionalization structure. Compared with standard BO methods (Srinivas et al., 2009; Belakaria et al., 2019; Chowdhury & Gopalan, 2021), our work is the first to establish a BO framework for tensor outputs and further extend it to the proposed CBBO setting. Moreover, our contributions lie in deriving regret bounds for both TOBO and TOCBBO based on concentration inequalities specialized to the proposed TOGP under a Bayesian framework.

## 2 RELATED WORKS

**High-order Gaussian processes (HOGPs):** Most existing studies for modeling HOGPs adopt separable kernel structures. In particular, Belyaev et al. (2015) proposes a tensor-variate GP with separable covariance across tensor modes, Kia et al. (2018) develops a scalable multi-task GP for tensor outputs by factorizing the cross-covariance kernel into mode-wise and input components, and Zhe et al. (2019) introduces a scalable high-order GP framework based on Kronecker-structured kernels. Although these formulations simplify computation, they typically imply that correlations across tensor modes do not vary with the input, which limits their ability to capture input-dependent dependencies. Recent work has also developed non-separable kernels for multi-output GPs (MOGPs), including convolution-based kernels (Fricker et al., 2013), linear model of coregionalization (LMC) kernels (Li & Zhou, 2016; Bruinsma et al., 2020), and linear damped harmonic oscillator-based kernels (Hristopulos, 2023). A closely related viewpoint is multi-task GPs, where each task corresponds to one output component (Yu et al., 2018; Chowdhury & Gopalan, 2021; Maddox et al., 2021). However, these methods are primarily designed for vector-valued outputs and therefore cannot fully exploit the intrinsic multi-mode structure of tensor data, see Section 1.

**Multi-output Bayesian optimization (MOBO):** MOBO typically refers to either multi-task BO or multi-objective BO (Frazier, 2018; Wang et al., 2023). A desirable property of BO algorithms is no-regret, namely achieving cumulative regret $R(T) = o(T)$ after $T$ rounds (Srinivas et al., 2009; Chowdhury & Gopalan, 2021). Recent work has developed various MOBO methods with theoretically grounded acquisition functions. For multi-task BO, Chowdhury & Gopalan (2021); Dai et al. (2020); Sessa et al. (2023) employ UCB-type acquisition rules and establish $\mathcal{O}(\sqrt{T})$ regret. For multi-objective BO, Belakaria et al. (2019); Zhang et al. (2025) study max-value entropy search and obtain $\mathcal{O}(\sqrt{T})$ regret, while Daulton et al. (2022a) proposes a trust-region-based criterion with $\mathcal{O}(\sqrt{T \log T})$ regret. In parallel, a large body of MOBO methods emphasizes empirical performance without regret guarantees, including improvement-based criteria (Uhrenholt & Jensen, 2019; Daulton et al., 2020; 2022b), entropy-based search (Hernández-Lobato et al., 2014; 2016; Tu et al., 2022), and information-gain-based methods (Chowdhury & Gopalan, 2021). Although effective for multi-output problems, these methods typically do not exploit the intrinsic multi-mode correlations of tensor outputs and are therefore less suitable for our TOBO setting. Moreover, hypervolume-based and entropy-search-based multi-objective BO methods become computationally prohibitive as the number of objectives grows, which makes them impractical for tensor-output optimization.

**Bandit Bayesian optimization (BBO):** BBO combines BO with multi-armed bandit (MAB) algorithms to tackle optimization in mixed input spaces containing both continuous and categorical variables. Nguyen et al. (2020); Huang et al. (2022) study a setting with one continuous and one categorical variable, coupling BO with Thompson sampling for the categorical choice and establishing $\mathcal{O}(\sqrt{T^{\alpha+1} \log T})$ regret. Ru et al. (2020) extends this idea to multiple categorical variables, named CoCaBO, by employing EXP3 (Auer et al., 2002) to select categories and achieving $\mathcal{O}(\sqrt{T} \log T)$ regret. Such settings can be viewed as special cases of CBBO, yet they do not directly extend to our problem due to the tensor-output structure and the need to jointly select multiple correlated arms.

Table 1 summarizes a comprehensive comparison between our method and existing literature.

Table 1: Comparison of related works and our proposed method

| Type | Literature | GP | | | BO (Regret) | BBO (Regret) | CBBO (Regret) |
|---|---|---|---|---|---|---|---|
| | | Tensor structure | Separable (Independent modes) | Non-separable (Cross-mode correlations) | | | |
| HOGP | Belyaev et al. (2015) Kia et al. (2018) Zhe et al. (2019) | $\checkmark$ | $\checkmark$ | $\times$ | $\times$ | $\times$ | $\times$ |
| MOGP | Fricker et al. (2013) Li & Zhou (2016) Hristopulos (2023) | $\times$ | $\times$ | $\checkmark$ | $\times$ | $\times$ | $\times$ |
| MTBO | Dai et al. (2020) Sessa et al. (2023) | $\times$ | $\checkmark$ | $\times$ | $\checkmark$ ($\mathcal{O}(\sqrt{T})$) | $\times$ | $\times$ |
| MTBO | Chowdhury & Gopalan (2021) | $\times$ | $\checkmark$ | $\checkmark$ | $\checkmark$ ($\mathcal{O}(\sqrt{T})$) | $\times$ | $\times$ |
| MOBO | Belakaria et al. (2019) Zhang et al. (2025) | $\times$ | $\times$ | $\times$ | $\checkmark$ ($\mathcal{O}(\sqrt{T})$) | $\times$ | $\times$ |
| MOBO | Daulton et al. (2022a) | $\times$ | $\times$ | $\times$ | $\checkmark$ ($\mathcal{O}(\sqrt{T \log T})$) | $\times$ | $\times$ |
| BBO | Nguyen et al. (2020) Huang et al. (2022) | $\times$ | $\checkmark$ | $\times$ | $\checkmark$ ($\mathcal{O}(\sqrt{T^{\alpha+1} \log T})$) | $\checkmark$ ($\mathcal{O}(\sqrt{T^{\alpha+1} \log T})$) | $\times$ |
| BBO | Ru et al. (2020) | $\times$ | $\checkmark$ | $\times$ | $\checkmark$ ($\mathcal{O}(\sqrt{T} \log T)$) | $\checkmark$ ($\mathcal{O}(\sqrt{T} \log T)$) | $\times$ |
| TOBO+TOCBBO | **Our proposed method** | $\checkmark$ | $\checkmark$ | $\checkmark$ | $\checkmark$ ($\mathcal{O}(\sqrt{T} \log T)$) | $\checkmark$ ($\mathcal{O}(\sqrt{T} \log T)$) | $\checkmark$ ($\mathcal{O}(\sqrt{T} \log T)$) |

## 3 TENSOR-OUTPUT BAYESIAN OPTIMIZATION

In this section, we propose a novel tensor-output Bayesian optimization (TOBO) framework for optimizing expensive black-box systems with tensor-valued responses. Let $\mathbf{f} : \mathcal{X} \to \mathcal{Y}$ denote the objective function, where the input $\boldsymbol{x} = (x_1, \ldots, x_d)$ lies in a compact and convex domain $\mathcal{X} \subset \mathbb{R}^d$, and the output $\mathbf{f}(\boldsymbol{x}) \in \mathcal{Y} \subset \mathbb{R}^{t_1 \times \cdots \times t_m}$ is an $m$-mode tensor. Denote $f_{i_1, \ldots, i_m}(\boldsymbol{x})$ as the $(i_1, \ldots, i_m)$-th entry of the tensor, where $i_l = 1, \ldots, t_l$ for $l = 1, \ldots, m$, and let $T = \prod_{l=1}^m t_l$ be the total number of elements. To optimize tensor-output systems, an intuitive way is to map the tensor-valued objective into a scalar function. To this end, we introduce a bounded linear operator $L_f \in \mathcal{L}(\mathcal{Y}, \mathbb{R})$, where $\mathcal{L}(\mathcal{Y}, \mathbb{R})$ denotes the set of bounded linear operators from $\mathcal{Y}$ to $\mathbb{R}$. The optimization problem is thus

$$\boldsymbol{x}^\star = \arg\max_{\boldsymbol{x} \in \mathcal{X}} L_f \mathbf{f}(\boldsymbol{x}). \tag{1}$$

To solve this problem, the proposed TOBO aims to sequentially select inputs $\boldsymbol{x}_i$ and observe the corresponding tensor outputs,

$$\mathbf{y}_i = \mathbf{f}(\boldsymbol{x}_i) + \boldsymbol{\varepsilon}_i, \qquad \forall i = 1, 2, \ldots \tag{2}$$

where $\boldsymbol{\varepsilon}_i \in \mathbb{R}^{t_1 \times \cdots \times t_m}$ denotes i.i.d. measurement noise with $\text{vec}(\boldsymbol{\varepsilon}_i) \sim \mathcal{N}(0, \tau^2 \mathbf{I}_T)$, and $\text{vec} : \mathcal{Y} \to \mathbb{R}^T$ is the vectorization operator.

Based on the collected data, we fit a tensor-output Gaussian process (TOGP) surrogate for $\mathbf{f}$ using two classes of tensor-output kernels, as detailed in Subsection 3.1. Then, we develop a UCB-based acquisition strategy to efficiently identify the maximizer $\boldsymbol{x}^\star$, as presented in Subsection 3.2.

### 3.1 TENSOR-OUTPUT GAUSSIAN PROCESS

Define the prior of $\mathbf{f} : \mathcal{X} \to \mathcal{Y}$ as a tensor-output Gaussian process (TOGP):

$$\text{vec}(\mathbf{f}(\boldsymbol{x})) \sim \mathcal{TOGP}\left(\boldsymbol{\mu}, \sigma^2 \mathbf{K}(\boldsymbol{x}, \boldsymbol{x}')\right), \qquad \forall \boldsymbol{x}, \boldsymbol{x}' \in \mathcal{X}, \tag{3}$$

where $\boldsymbol{\mu} \in \mathbb{R}^T$ is the prior mean, $\sigma^2 > 0$ is a variance hyperparameter, and $\mathbf{K}(\boldsymbol{x}, \boldsymbol{x}') \in \mathbb{R}^{T \times T}$ is a symmetric and positive semi-definite kernel function. The classes of $\mathbf{K}(\boldsymbol{x}, \boldsymbol{x}')$ are discussed later.

Given $n$ observations $\mathbf{X}_n = (\boldsymbol{x}_1, \ldots, \boldsymbol{x}_n)^\top$ and $\boldsymbol{Y}_n = \left(vec(\mathbf{y}_1)^\top, \ldots, vec(\mathbf{y}_n)^\top\right)^\top$, the posterior distribution of $\text{vec}(\mathbf{f}(\boldsymbol{x}))$ at a new input $\boldsymbol{x} \in \mathcal{X}$ is Gaussian in $\mathbb{R}^T$ with mean and covariance

$$\hat{\boldsymbol{\mu}}_n(\boldsymbol{x}) = \boldsymbol{\mu} + \mathbf{K}_n^\top(\boldsymbol{x}) \left(\mathbf{K}_n + \eta \mathbf{I}_{nT}\right)^{-1} \left(\boldsymbol{Y}_n - \mathbf{1}_n \otimes \boldsymbol{\mu}\right), \tag{4}$$

$$\hat{\mathbf{K}}_n(\boldsymbol{x}, \boldsymbol{x}) = \sigma^2 \left[\mathbf{K}(\boldsymbol{x}, \boldsymbol{x}) - \mathbf{K}_n^\top(\boldsymbol{x}) \left(\mathbf{K}_n + \eta \mathbf{I}_{nT}\right)^{-1} \mathbf{K}_n(\boldsymbol{x})\right], \tag{5}$$

where $\mathbf{K}_n(\boldsymbol{x}) \in \mathbb{R}^{nT \times T}$ is the block column matrix whose $i$-th block is $\mathbf{K}(\boldsymbol{x}_i, \boldsymbol{x})$, $\mathbf{K}_n \in \mathbb{R}^{nT \times nT}$ is the block matrix with $(i, j)$-block $\mathbf{K}(\boldsymbol{x}_i, \boldsymbol{x}_j)$, $\mathbf{1}_n$ is the $n$-vector of ones, and $\eta = \tau^2 / \sigma^2$.

To specify the tensor-output kernel in TOGP, we introduce two classes of kernels. The first class consists of non-separable tensor-output kernels constructed via a tensor extension of the linear model of coregionalization (LMC) (Fricker et al., 2013; Li & Zhou, 2016). Let $\text{vec}(\mathbf{f}(\boldsymbol{x})) = \sum_{\ell=1}^m \sum_{j=1}^{r_\ell} \boldsymbol{a}_\ell u_{\ell j}(\boldsymbol{x})$, where $\boldsymbol{a}_{\ell j} \in \mathbb{R}^T$ are loading vectors and $\{u_{\ell j}\}$ are independent scalar GPs with zero mean and covariance functions $\text{Cov}(u_{\ell j}(\boldsymbol{x}), u_{\ell j}(\boldsymbol{x}')) = k_{\ell j}(\boldsymbol{x}, \boldsymbol{x}')$. We parameterize the loading vectors by tensor cores, namely $\boldsymbol{a}_{\ell j} = \text{vec}(\mathbf{A}_{\ell j})$ with $\mathbf{A}_{\ell j} \in \mathbb{R}^{t_1 \times \cdots \times t_m}$. Then $\mathbf{f}(\boldsymbol{x})$ has zero mean and covariance $\text{Cov}(\text{vec}(\mathbf{f}(\boldsymbol{x})), \text{vec}(\mathbf{f}(\boldsymbol{x}'))) = \sum_{\ell=1}^m \sum_{j=1}^{r_\ell} \text{vec}(\mathbf{A}_\ell) \text{vec}(\mathbf{A}_\ell)^\top k_{\ell j}(\boldsymbol{x}, \boldsymbol{x}')$. This yields a non-separable covariance structure since the cross-entry correlations are governed by a mixture of base kernels $\{k_{\ell j}\}$ and therefore can vary with the input. The non-separable tensor-output kernel is defined as

**Definition 1.** *Define the non-separable tensor-output kernel* $\mathbf{K}(\boldsymbol{x}, \boldsymbol{x}')$ *for any* $\boldsymbol{x}, \boldsymbol{x}' \in \mathcal{X}$:

$$\mathbf{K}(\boldsymbol{x}, \boldsymbol{x}') = \sum_{l=1}^m \sum_{j=1}^{t_l} \text{vec}(\mathbf{A}_l) \text{vec}(\mathbf{A}_l)^\top k_{lj}(\boldsymbol{x}, \boldsymbol{x}'), \tag{6}$$

*where* $\mathbf{A}_l \in \mathbb{R}^{t_1 \times \cdots \times t_m}$ *are core tensors and* $k_{lj}(\boldsymbol{x}, \boldsymbol{x}')$ *are base kernels on* $\mathcal{X}$ *(e.g., Matérn).*

Furthermore, if all $w_{i_1,\ldots,i_m}(\boldsymbol{z})$ share the same covariance $k(\boldsymbol{z},\boldsymbol{z}')$ for $i_l \in [t_l]$, $l = 1,\ldots,m$, and the convolution degenerates to $g_{i_1,\ldots,i_m}(\boldsymbol{z}) = A_{i_1,\ldots,i_m}\delta(\boldsymbol{z}-\boldsymbol{x})$, then the induced tensor-output kernel reduces to a separable structure in which the correlations across tensor modes are independent of the input.

**Definition 2.** *For any $\boldsymbol{x}, \boldsymbol{x}' \in \mathcal{X}$, define the separable tensor-output kernel*

$$\mathbf{K}(\boldsymbol{x},\boldsymbol{x}') = \mathrm{vec}(\mathbf{A})\mathrm{vec}(\mathbf{A})^\top k(\boldsymbol{x},\boldsymbol{x}'). \tag{7}$$

*where $\mathbf{A} \in \mathbb{R}^{t_1 \times \cdots \times t_m}$ is a core tensor, and $k(\cdot,\cdot)$ is a base kernel on $\mathcal{X}$.*

Note that $\mathbf{K}(\boldsymbol{x},\boldsymbol{x}')$ is designed to capture both input correlations and dependencies among tensor entries. For the non-separable kernel in (6), each base kernel $k_{lj}(\boldsymbol{x},\boldsymbol{x}')$ models correlations in the input space, while the matrix $\mathrm{vec}(\mathbf{A}_l)\mathrm{vec}(\mathbf{A}_l)^\top$ encodes the dependency structure across tensor entries. For the separable kernel in (7), $k(\boldsymbol{x},\boldsymbol{x}')$ captures input correlations and $\mathrm{vec}(\mathbf{A})\mathrm{vec}(\mathbf{A})^\top$ specifies the dependency structure within the tensor output.

**Proposition 1.** *The kernels in Definitions 1 and 2 are symmetric and positive semidefinite on $\mathcal{X}$, i.e., (1) $\forall \boldsymbol{x}, \boldsymbol{x}' \in \mathcal{X}$, $\mathbf{K}(\boldsymbol{x},\boldsymbol{x}') = \mathbf{K}(\boldsymbol{x}',\boldsymbol{x})^\top$; (2) $\forall n \in \mathbb{N}$, any $\boldsymbol{x}_1,\ldots,\boldsymbol{x}_n \in \mathcal{X}$, and any $\mathbf{y}_1,\ldots,\mathbf{y}_n \in \mathcal{Y}$, $\sum_{i,j=1}^n vec(\mathbf{y}_i)^\top \mathbf{K}(\boldsymbol{x}_i,\boldsymbol{x}_j)vec(\mathbf{y}_j) \geq 0$.*

Its proof is given in Appendix E. Proposition 1 ensures that both classes of tensor-output kernels define valid covariance functions for TOGP, and hence the induced TOGP prior is well defined.

**Remark 1.** *The core tensors $\{\mathbf{A}_l\}_{l=1}^m$ in (6) and $\mathbf{A}$ in (7) contain $mT$ and $T$ free parameters, respectively. To reduce the parameter complexity, low-rank decomposition-based structures can be imposed on these cores, such as CANDECOMP/PARAFAC (CP) decomposition (Goulart et al., 2015) or tensor-train (TT) decomposition (Oseledets, 2011). Details are shown in Appendix B.*

Denote the hyperparameters by $\boldsymbol{\Theta} = \{\boldsymbol{\theta}, \mathbf{a}, \sigma^2, \tau^2\}$. For the non-separable kernel in (6), $\boldsymbol{\theta} = \{\theta_{l,j}\}_{l,j=1}^{m,t_l}$ are the parameters of the base kernel $\{k_{l,j}(\cdot,\cdot)\}$, and $\boldsymbol{a} = \{a_{lij}\}_{l,i,j}^{m,m,t_l}$ are the entries of $\{\mathbf{A}_l\}_{l=1}^m$. For the kernel in (7), $\boldsymbol{\theta} = \{\theta_1,\ldots,\theta_d\}$ is the parameters of $k(\cdot,\cdot)$, and $\boldsymbol{a} = \{a_{ij}\}_{i,j}^{m,t_l}$ are the entries of $\mathbf{A}$. We estimate $\boldsymbol{\Theta}$ by maximum likelihood. The detailed estimation, algorithm, and complexity analysis are provided in Appendix C.

**Remark 2.** *The ranks of the core tensors $\{\mathbf{A}_\ell\}_{\ell=1}^m$ are selected via cross-validation. Let $\mathbf{R}_c = \{\boldsymbol{r}_1,\ldots,\boldsymbol{r}_c\}$ be the set of candidate ranks, we fit the TOGP model and evaluate its predictive performance on a held-out validation set using the mean absolute error (MAE) criterion. We select $\boldsymbol{r}^\star = \arg\min_{\boldsymbol{r}\in\mathbf{R}_c} MAE(\boldsymbol{r})$, where $MAE(\boldsymbol{r}) = \frac{1}{n_{test}}\sum_{i=1}^{n_{test}}\left\|\frac{\mathbf{f}_i - \hat{\mathbf{f}}_i(\boldsymbol{r})}{\mathbf{f}_i}\right\|$. The data-driven choice balances model flexibility and complexity, mitigating overfitting while preserving expressive capacity.*

**Remark 3.** *To improve scalability for large $nT$, we adopt a Nyström low-rank approximation of the block covariance matrix $\mathbf{K}_n \in \mathbb{R}^{nT \times nT}$. Specifically, it selects $n_\ell \ll nT$ landmark columns to construct a rank-$n_\ell$ approximation $\mathbf{K}_n \approx \tilde{\mathbf{U}}\tilde{\mathbf{\Lambda}}\tilde{\mathbf{U}}^\top$, which can be viewed as approximating the leading spectral components of $\mathbf{K}_n$. We choose the effective rank via cumulative explained variance, i.e., $l = \min_{l_0}\left\{l_0 \in \{1,\ldots,nT\} : \sum_{i=1}^{l_0}\lambda_i / \sum_{i=1}^{nT}\lambda_i \geq c\right\}$, where $\{\lambda_i\}$ denote the eigenvalues of $\mathbf{K}_n$ in non-increasing order. Then, evaluating $(\mathbf{K}_n + \eta\mathbf{I}_{nT})^{-1}$ can be reduced to $\mathcal{O}(nT\,n_\ell^2 + n_\ell^3)$ (Williams & Seeger, 2000). Thus, the overall training complexity is reduced to $\mathcal{O}\left((nT\,n_\ell^2 + n_\ell^3 + n^2T^2m_h)\log n\right)$.*

### 3.2 UPPER CONFIDENCE BOUND ACQUISITION STRATEGY

Building on TOGP as a surrogate for $\mathbf{f}$, we develop a UCB-based acquisition strategy. At round $n+1$, given past observations $\mathbf{X}_n$ and $\mathbf{Y}_n$, we update the hyperparameters $\boldsymbol{\Theta}_n$ and the posterior mean (4) and covariance (5). The UCB acquisition function for the scalarization-based objective in (1) is defined as

$$\alpha_{UCB}(\boldsymbol{x} \mid \mathcal{D}_n) = L_f\hat{\boldsymbol{\mu}}_n(\boldsymbol{x}) + \beta_n\left\|\hat{\mathbf{K}}_n(\boldsymbol{x},\boldsymbol{x})\right\|^{1/2}, \tag{8}$$

where $\beta_n > 0$ balances exploration and exploitation. This criterion encourages exploration in directions with greater predictive uncertainty under the tensor-output setting. The next query is then

selected by maximizing (8):

$$\boldsymbol{x}_{n+1} = \arg\max_{\boldsymbol{x}\in\mathcal{X}} \alpha_{UCB}(\boldsymbol{x} \mid \mathcal{D}_n). \tag{9}$$

The complete TOBO algorithm and its complexity analysis are given in Appendix D.

We now study theoretical properties of TOBO under the following conditions. We first define two commonly used notions of regret.

**Definition 3.** *At each round $n$, the TOBO method selects a queried input $\boldsymbol{x}_n \in \mathcal{X}$. The instantaneous regret is defined as $r_n = L_f\mathbf{f}(\boldsymbol{x}^\star) - L_f\mathbf{f}(\boldsymbol{x}_n)$, and the cumulative regret up to round $N$ is defined as $R_N = \sum_{n=1}^{N} [L_f\mathbf{f}(\boldsymbol{x}^\star) - L_f\mathbf{f}(\boldsymbol{x}_n)]$.*

The regret quantifies the gap from not knowing the objective in advance. A good strategy can achieve a sub-linear cumulative regret so that the average regret per round converges to zero as $N \to \infty$.

**Definition 4.** *At round $n$, TOBO selects an input $\boldsymbol{x}_n \in \mathcal{X}$. The instantaneous regret is $r_n = L_f\mathbf{f}(\boldsymbol{x}^\star) - L_f\mathbf{f}(\boldsymbol{x}_n)$, and the cumulative regret up to round $N$ is $R_N = \sum_{n=1}^{N} [L_f\mathbf{f}(\boldsymbol{x}^\star) - L_f\mathbf{f}(\boldsymbol{x}_n)]$.*

The regret quantifies the gap due to not knowing the objective in advance. A good strategy achieves sublinear cumulative regret so that the average regret $R_N/N$ converges to zero as $N \to \infty$.

**Assumption 1.** *The unknown function $\mathbf{f}$ is a TOGP prior with kernel $\mathbf{K}$ as defined in (6)–(7).*

**Assumption 2.** *The scalarized objective $\boldsymbol{x} \mapsto L_f(\mathbf{f}(\boldsymbol{x}))$ is L-Lipschitz on $\mathcal{X}$, i.e., for any $\boldsymbol{x}, \boldsymbol{x}' \in \mathcal{X}$, $|L_f(\mathbf{f}(\boldsymbol{x}_i)) - L_f(\mathbf{f}(\boldsymbol{x}_j))| \leq L\|\mathbf{f}(\boldsymbol{x}_i) - \mathbf{f}(\boldsymbol{x}_j)\|$ holds.*

Assumption 1 specifies the Bayesian setting under which $\mathbf{f}$ follows a TOGP prior. Assumption 2 ensures that the scalarized objective is stable under small perturbations of the input.

**Lemma 1.** *Let $\partial vec(\mathbf{f})/\partial x_j \in \mathbb{R}^T$ be the gradient of $vec(\mathbf{f})$ with respect to the $j$-th coordinate of $\boldsymbol{x} \in \mathcal{X}$. Then, $\partial vec(\mathbf{f})/\partial x_j$ is a GP with covariance $\mathbf{K}^\nabla(x_j, x'_j)$. Under Assumptions 1, given data $\mathbf{X}_n$ and $\mathbf{Y}_n$ with $n \geq 1$, there exist constants $a, b > 0$ such that for any $L' > 0$,*

$$\Pr\left(\sup_{\boldsymbol{x}\in\mathcal{X}} \|\partial vec(\mathbf{f})/\partial x_j\| > L' + C_\nabla\right) \leq a\exp(-L'^2/b^2), \quad j = 1, \cdots, d. \tag{10}$$

*where $C_\nabla = \sup_{\boldsymbol{x}\in\mathcal{X}} \sqrt{\text{tr}(\hat{\mathbf{K}}_n^\nabla(x_j, x_j))}$.*

Its proof is given in Appendix F. Lemma 1 shows that the derivative of the vectorized TOGP remains Gaussian and holds high-probability confidence bounds.

**Theorem 1.** *Under Assumptions 1–2, define $C_n = \sup_{\boldsymbol{x}\in\mathcal{X}} \text{tr}(\hat{\mathbf{K}}_n(\boldsymbol{x}, \boldsymbol{x}))/\lambda_{\max}^{(n)}(\boldsymbol{x})$, where $\lambda_{\max}^{(n)}(\boldsymbol{x})$ is the largest eigenvalue of $\hat{\mathbf{K}}_n(\boldsymbol{x}, \boldsymbol{x})$. Suppose $\mathcal{X} \subseteq [0, r]^d$. Then, for any $\delta \in (0, 1)$, TOBO with $\beta_n = \sqrt{C_n} + 2d\log(rdn^2(b\sqrt{\log(da/\delta)} + C_\nabla)/\delta)$ holds that, $\Pr\left(R_N \leq L\left(\sqrt{C_1 N\gamma_N(\mathbf{K}, \eta)}\,\beta_N + \frac{\pi^2}{6}\right)\right) \geq 1-\delta$, where $C_1 > 0$ is a constant and $\gamma_N(\mathbf{K}, \eta) := \max_{\mathcal{X}_N \subset \mathcal{X}} \frac{1}{2}\log\det\left(\mathbf{I}_{NT} + \eta^{-1}\mathbf{K}_N\right)$ denotes the maximum information gain.*

Its proof is given in Appendix G. Theorem 1 implies that TOBO achieves sublinear cumulative regret with high probability.

**Proposition 2.** *If $\mathbf{K}(\boldsymbol{x}, \boldsymbol{x}')$ is the separable kernel in Definition 2, then the maximum information gain satisfies $\gamma_n(\mathbf{K}, \eta) = \mathcal{O}\left(T\log(n)^{d+1}\right)$ when $k(\boldsymbol{x}, \boldsymbol{x}')$ is a Gaussian kernel, and $\gamma_n(\mathbf{K}, \eta) = \mathcal{O}(Tn^{d(d+1)/(2\nu+d(d+1))}\log(n))$ when $k(\boldsymbol{x}, \boldsymbol{x}')$ is a Matérn kernel with $\nu > 1$. Details are provided in Appendix H.*

## 4 TENSOR-OUTPUT COMBINATORIAL BANDIT BAYESIAN OPTIMIZATION

We now consider a more challenging setting in which only $k < T$ entries of the tensor output contribute to the objective. Formally, we define $\tilde{\boldsymbol{f}}(\boldsymbol{x}, \boldsymbol{\lambda}) = \mathbf{e}(\boldsymbol{\lambda})vec(\mathbf{f}(\boldsymbol{x})) \in \tilde{\mathcal{Y}}$, where $\boldsymbol{\lambda} = $

$(\lambda_1, \ldots, \lambda_T)^\top$ is a binary indicator vector in $\Lambda = \{\boldsymbol{\lambda} \in \{0,1\}^T : \mathbf{1}_n^\top \boldsymbol{\lambda} = k\}$ and $\mathbf{e}(\boldsymbol{\lambda}) \in \{0,1\}^{k \times T}$ is a binary selection matrix whose $j$-th row selects the $i_j$-th coordinate of $\mathrm{vec}(\mathbf{f}(\boldsymbol{x}))$. The goal is to jointly identify $\boldsymbol{x}^\star \in \mathcal{X}$ and the optimal subset of $k$ elements, encoded by $\boldsymbol{\lambda}^\star \in \Lambda$, that maximize

$$(\boldsymbol{x}^\star, \boldsymbol{\lambda}^\star) = \arg \max_{\boldsymbol{x} \in \mathcal{X}, \boldsymbol{\lambda} \in \Lambda} H_f \tilde{\boldsymbol{f}}(\boldsymbol{x}, \boldsymbol{\lambda}), \tag{11}$$

where $H_f$ is a bounded linear operator $H_f \in \mathcal{L}(\tilde{\mathcal{Y}}, \mathbb{R})$. By interpreting each tensor element as a base arm, any $\boldsymbol{\lambda} \in \Lambda$ corresponds to a super-arm $\mathcal{S} \subseteq [T]$ of size $k$, where $j \in \mathcal{S}$ if $\lambda_j = 1$ and $j \notin \mathcal{S}$ otherwise. At each round $i \in [N]$, the learner selects an input $\boldsymbol{x}_i$ together with a super-arm $\mathcal{S}_i = \{i_1, \ldots, i_k\}$. The observation is the partial output $\tilde{\boldsymbol{y}}_i \in \mathbb{R}^k$ indexed by $\mathcal{S}_i$, while entries with indices $j \notin \mathcal{S}_i$ remain unobserved.

In this section, we develop a tensor-output combinatorial bandit Bayesian optimization (TOCBBO) framework to solve (11). In Subsection 4.1, we extend TOGP for partially observed outputs. In Subsection 4.2, we develop an efficient CMAB-UCB2 acquisition strategy that couples UCB-based input selection with CMAB-UCB-based super-arm selection.

## 4.1 Partially observed tensor-output Gaussian process

From the proposed TOGP in (3), the prior of $\tilde{\boldsymbol{f}} : \mathcal{X} \times \Lambda \to \tilde{\mathcal{Y}}$ is a partially observed TOGP (PTOGP):

$$\tilde{\boldsymbol{f}}(\boldsymbol{x}, \boldsymbol{\lambda}) \sim \mathcal{PTOGP}\left(\mathbf{e}(\boldsymbol{\lambda})\boldsymbol{\mu}(\boldsymbol{x}), \tau^2 \mathbf{e}(\boldsymbol{\lambda})\mathbf{K}(\boldsymbol{x}, \boldsymbol{x}')\mathbf{e}(\boldsymbol{\lambda}')^\top\right), \quad \forall \boldsymbol{x}, \boldsymbol{x}' \in \mathcal{X}, \ \boldsymbol{\lambda}, \boldsymbol{\lambda}' \in \Lambda. \tag{12}$$

Let $\mathbf{X}_n = (\boldsymbol{x}_1, \ldots, \boldsymbol{x}_n)^\top$, $\boldsymbol{\Lambda}_n = (\boldsymbol{\lambda}_1, \ldots, \boldsymbol{\lambda}_n)^\top$, and $\tilde{\boldsymbol{Y}}_n = (\tilde{\boldsymbol{y}}_1, \ldots, \tilde{\boldsymbol{y}}_n)^\top$ be $n$ partial observations, where $\boldsymbol{\lambda}_i$ encodes the selected super-arm $\mathcal{S}_i$, and $\tilde{\boldsymbol{y}}_i = \tilde{\boldsymbol{f}}(\boldsymbol{x}_i, \boldsymbol{\lambda}_i) + \tilde{\boldsymbol{\varepsilon}}_i$ with $\tilde{\boldsymbol{\varepsilon}}_i \overset{\text{i.i.d.}}{\sim} \mathcal{N}(\mu, \tau^2 \mathbf{I}_k)$. Then, for a new pair $(\boldsymbol{x}, \boldsymbol{\lambda})$, the posterior of $\tilde{\boldsymbol{f}}(\boldsymbol{x}, \boldsymbol{\lambda})$ is Gaussian in $\mathbb{R}^k$ with mean and covariance

$$\tilde{\boldsymbol{\mu}}_n(\boldsymbol{x}, \boldsymbol{\lambda}) = \mathbf{e}(\boldsymbol{\lambda})\boldsymbol{\mu}(\boldsymbol{x}) + \sigma^2 \mathbf{e}(\boldsymbol{\lambda})\mathbf{K}_n^\top(\boldsymbol{x})\mathbf{E}_n^\top \tilde{\boldsymbol{\Sigma}}_n^{-1} \left(\mathrm{vec}(\tilde{\boldsymbol{Y}}_n) - \mathbf{E}_n(\mathbf{1}_n \otimes \boldsymbol{\mu})\right), \tag{13}$$

$$\tilde{\mathbf{K}}_n(\boldsymbol{x}, \boldsymbol{x}'; \boldsymbol{\lambda}, \boldsymbol{\lambda}') = \sigma^2 \left[\mathbf{e}(\boldsymbol{\lambda})\mathbf{K}(\boldsymbol{x}, \boldsymbol{x}')\mathbf{e}(\boldsymbol{\lambda}')^\top - \sigma^2 \mathbf{e}(\boldsymbol{\lambda})\mathbf{K}_n^\top(\boldsymbol{x})\mathbf{E}_n \tilde{\boldsymbol{\Sigma}}_n^{-1}\mathbf{E}_n^\top \mathbf{K}_n(\boldsymbol{x}')\mathbf{e}(\boldsymbol{\lambda}')^\top\right], \tag{14}$$

where $\tilde{\boldsymbol{\Sigma}}_n = \sigma^2 \mathbf{E}_n \mathbf{K}_n \mathbf{E}_n^\top + \tau^2 \mathbf{I}_{nk}$, and $\mathbf{E}_n \in \mathbb{R}^{nk \times nT}$ is a $n \times n$ block-diagonal matrix with the $i$-block given by $\mathbf{e}(\boldsymbol{\lambda}_i)$. It is easy to verify that the posterior covariance is symmetric and positive semidefinite. We estimate the hyperparameters of PTOGP via maximum likelihood. Detailed estimation, algorithm, and complexity analysis are presented in Appendix C.

## 4.2 CMAB-UCB2 Acquisition Strategy

Building on the PTOGP, we develop TOCBBO to sequentially select query inputs $\{\boldsymbol{x}_1, \ldots, \boldsymbol{x}_N\}$ together with their associated super-arm indicators $\{\boldsymbol{\lambda}_1, \ldots, \boldsymbol{\lambda}_N\}$. Directly optimizing (11) over both $\boldsymbol{x}$ and $\boldsymbol{\lambda}$ is computationally intractable, since identifying the optimal super-arm of size $k$ from $T$ arms requires a combinatorial search over $\binom{T}{k}$ candidates, which becomes prohibitive when coupled with continuous optimization over $\mathcal{X}$. To address this challenge, we propose a CMAB-UCB2 criterion that decouples the joint optimization into two sequential steps.

At round $n + 1$, let $(\boldsymbol{x}_n^\star, \boldsymbol{\lambda}_n^\star)$ denote the best pair observed so far, that is, $\{\boldsymbol{x}_n^\star, \boldsymbol{\lambda}_n^\star\} = \arg\max_{\{\boldsymbol{x}_i, \boldsymbol{\lambda}_i\}, i=1,\ldots,n} H_f \tilde{\boldsymbol{f}}(\boldsymbol{x}_i, \boldsymbol{\lambda}_i)$. In the first step, we fix the super-arm to $\boldsymbol{\lambda}_n^\star$ and select the next input by maximizing a UCB acquisition function under this fixed super-arm:

$$\boldsymbol{x}_{n+1} = \arg\max_{\boldsymbol{x} \in \mathcal{X}} H_f \tilde{\boldsymbol{\mu}}_n(\boldsymbol{x}, \boldsymbol{\lambda}_n^\star) + \tilde{\beta}_n \|\tilde{\mathbf{K}}_n(\boldsymbol{x}, \boldsymbol{x}; \boldsymbol{\lambda}_n^\star, \boldsymbol{\lambda}_n^\star)\|^{1/2}. \tag{15}$$

In the second step, given the chosen input $\boldsymbol{x}_{n+1}$, selecting $\boldsymbol{\lambda}_{n+1}$ reduces to a combinatorial bandit problem. To this end, we adopt the CMAB-UCB criterion by constructing a UCB for each super-arm and selecting the one that maximizes the upper confidence value:

$$\boldsymbol{\lambda}_{n+1} = \arg\max_{\boldsymbol{\lambda} \in \Lambda} H_f \tilde{\boldsymbol{\mu}}_n(\boldsymbol{x}_{n+1}, \boldsymbol{\lambda}) + \tilde{\rho}_n \|\tilde{\mathbf{K}}_n(\boldsymbol{x}_{n+1}, \boldsymbol{x}_{n+1}; \boldsymbol{\lambda}, \boldsymbol{\lambda})\|^{1/2}. \tag{16}$$

Here, $\tilde{\beta}_n$ and $\tilde{\rho}_n$ are tuning parameters controlling the trade-off between exploration and exploitation. Detailed algorithm and its computational complexity analysis are provided in Appendix D.

We further analyze the regret bound of TOCBBO under the following conditions. The regret for CBBO is defined as follows.

**Definition 5.** *At round $n$, TOCBBO selects an input an input $\boldsymbol{x}_n \in \mathcal{X}$ and a super-arm $\mathcal{S}_n$ encoded by $\boldsymbol{\lambda}_n \in \Lambda$. The instantaneous regret is $r_n = H_f \tilde{\boldsymbol{f}}(\boldsymbol{x}^\star, \boldsymbol{\lambda}^\star) - H_f \tilde{\boldsymbol{f}}(\boldsymbol{x}_n, \boldsymbol{\lambda}_n)$, aand the cumulative regret up to round $N$ is $R_N = \sum_{n=1}^{N} \left[ H_f \tilde{\boldsymbol{f}}(\boldsymbol{x}^\star, \boldsymbol{\lambda}^\star) - H_f \tilde{\boldsymbol{f}}(\boldsymbol{x}_n, \boldsymbol{\lambda}_n) \right]$.*

**Assumption 3.** *The operator $H_f$ in (11) is $H$-Lipschitz with respect to $\tilde{\boldsymbol{f}}$ under the $l_2$ norm, i.e., for $\forall \boldsymbol{x}_i, \boldsymbol{x}_j \in \mathcal{X}$ and $\boldsymbol{\lambda}_i, \boldsymbol{\lambda}_j \in \Lambda$, $|H_f \tilde{\boldsymbol{f}}(\boldsymbol{x}_i, \boldsymbol{\lambda}_i) - H_f \tilde{\boldsymbol{f}}(\boldsymbol{x}_j, \boldsymbol{\lambda}_j)| \le H \|\tilde{\boldsymbol{f}}(\boldsymbol{x}_i, \boldsymbol{\lambda}_i) - \tilde{\boldsymbol{f}}(\boldsymbol{x}_j, \boldsymbol{\lambda}_j)\|$ holds.*

Assumption 3 ensures that $H_f \tilde{\boldsymbol{f}}$ varies smoothly to changes for partially observed tensor outputs.

**Theorem 2.** *Under Assumption 1 and Assumption 2, denote $\tilde{C}_n = \sup_{\boldsymbol{x} \in \mathcal{X}} \frac{tr(\tilde{\mathbf{K}}_n(\boldsymbol{x}, \boldsymbol{x}; \boldsymbol{\lambda}_n^\star, \boldsymbol{\lambda}_n^\star)))}{\lambda_{max}^{(n)}(\boldsymbol{x}, \boldsymbol{\lambda}_n^\star)}$, where $\lambda_{max}^{(n)}(\boldsymbol{x}, \boldsymbol{\lambda}_n^\star)$ is the largest eigenvalue of $\tilde{\mathbf{K}}_n(\boldsymbol{x}, \boldsymbol{x}; \boldsymbol{\lambda}_n^\star, \boldsymbol{\lambda}_n^\star)$. For any $\delta \in (0, 1)$ and $\eta > 0$, TOCBBO with $\tilde{\beta}_n = \sqrt{\tilde{C}_n} + 2d \log(\frac{rdn^2(b\sqrt{\log(da/\delta)} + \tilde{C}_\nabla)}{l} \delta)$ and $\tilde{\rho}_n = \sqrt{2 \log\left(\frac{NT\pi_n}{\delta}\right)}$ holds that, $\Pr\left(R_N \le H\left(\sqrt{C_3 N \gamma_N(\tilde{\mathbf{K}}, \eta)} \tilde{\beta}_N + \frac{\pi^2}{6} + \sqrt{C_4 T N \tilde{\gamma}_N(\tilde{\mathbf{K}}, \eta)} \tilde{\rho}_N\right)\right) \ge 1 - \delta$, where $C_3, C_4 > 0$ are constants, $\pi_n > 0$ is a sequence such that $\sum_{n=1}^{\infty} 1/\pi_n = 1$, $\gamma_n(\tilde{\mathbf{K}}, \eta) = \max_{\mathbf{X}_N \subset \mathcal{X}} \frac{1}{2} \log \det\left(\mathbf{I}_{kN} + \eta^{-1} \mathbf{E}_N \mathbf{K}_N \mathbf{E}_N^\top\right)$, and $\tilde{\gamma}_N(\tilde{\mathbf{K}}, \eta) = \max_{\boldsymbol{\Lambda}_N \subset \Lambda} \frac{1}{2} \log \det\left(\mathbf{I}_{kN} + \eta^{-1} \mathbf{E}_N \tilde{\mathbf{K}}_N \mathbf{E}_N^\top\right)$ is the maximum information gain for super-arms.*

The proof is provided in Appendix I. Theorem 2 implies that TOCBBO achieves a sublinear regret upper bound with high probability.

**Proposition 3.** *Suppose the kernel $\mathbf{K}(\boldsymbol{x}, \boldsymbol{x}')$ follows the separable structure specified in Definition 2. The maximum information gain $\gamma_n(\tilde{\mathbf{K}}, \eta)$ and $\tilde{\gamma}_n(\tilde{\mathbf{K}}, \eta)$ are $\mathcal{O}\left(T(\log n)^{d+1}\right)$ and $\mathcal{O}\left(T n^{\frac{d(d+1)}{2\nu + d(d+1)}} \log n\right)$ for the Gaussian kernel and the Matérn kernel ($\nu > 1$), respectively, where $d$ denotes the input dimension. Details are provided in Appendix J.*

## 5 EXPERIMENTS

We evaluate TOBO and TOCBBO on both synthetic and real-case data, and compare them with baselines where the tensor output is vectorized and MOGPs are used as surrogate models. Specifically, we consider three GPs in the literature: (1) sMTGP: the scalable multi-task GP Kia et al. (2018); (2) MLGP: the multi-linear GP Yu et al. (2018); and (3) MVGP: the multi-variate GP Chen et al. (2020). For each GP, we examine two sequential BO sampling strategies: UCB-based BO and random sampling. In addition, we replace the UCB acquisition in TOBO and TOCBBO with random sampling to construct an ablation baseline, denoted as TOGP-RS. Details of all baseline settings are provided in Appendix K.

### 5.1 SYNTHETIC EXPERIMENTS

We generate synthetic functions of the form $\mathbf{f}(\boldsymbol{x}) = \mathbf{B} \otimes_1 \mathbf{U}_1 \otimes_2 \ldots \otimes_{m-1} \mathbf{U}_{m-1} \otimes_m \mathbf{g}(\boldsymbol{x})$, where each element of $\mathbf{B} \in \mathbb{R}^{P_1 \times \ldots P_m}$ is independently sampled from $U(0, 1)$, the $ij$-th element of $\mathbf{U}_l \in \mathbb{R}^{P_l \times T_l}$ is defined as $li \cos(ijl/2) + \sin(li)$, and $\mathbf{g}(\boldsymbol{x}) = (\sin(5\boldsymbol{x}), \cos(\boldsymbol{x})) \in \mathbb{R}^{P_m \times T_m}$. Here $P_m = d$ and $\boldsymbol{x} \in [0, 1]^d$. We consider three parameter settings: $\mathbf{f}(\boldsymbol{x})$: (1) $m = 3$, $(T_1, T_2, T_3) = (2, 4, 2)$, $(P_1, P_2, P_3) = (3, 3, 3)$; (2) $m = 2$, $(T_1, T_2) = (3, 2)$, $(P_1, P_2) = (3, 2)$; and (3) $m = 3$, $(T_1, T_2, T_3) = (4, 5, 2)$, $(P_1, P_2, P_3) = (3, 3, 3)$. Observations are collected as $\mathbf{y}_i = \mathbf{f}(\boldsymbol{x}_i) + \boldsymbol{\varepsilon}_i$, where $\boldsymbol{\varepsilon}_i \overset{\text{i.i.d.}}{\sim} N(0, 0.1^2 \mathbf{I})$. For CBBO tasks, we set $k = T/6$.

We generate $n_{\text{train}} = 10d$ training samples and $n_{\text{test}} = 5d$ testing samples via a Latin hypercube design (Santner et al., 2003). The training set is used for hyperparameter estimation, and predictive performance is evaluated on the test set in terms of NLL, MAE, and $\|\text{Cov}\|$, with details provided in Appendix K. To balance modeling flexibility and computational cost, we use the separable tensor-output kernel in (7) for Settings (1) and (2), and the non-separable tensor-output kernel in (6) for Setting (3). The results are summarized in Table 2. As shown, our proposed method achieves the lowest NLL and MAE, indicating that the TOGP model provides the highest prediction accuracy. Among the three baselines, MLGP performs the worst because its fitted covariance matrix becomes

Table 2: The prediction performance of GPs in the three synthetic settings.

| | Setting (1) | | | Setting (2) | | | Setting (3) | | |
|---|---|---|---|---|---|---|---|---|---|
| | NLL | MAE | ‖Cov‖ | NLL | MAE | ‖Cov‖ | NLL | MAE | ‖Cov‖ |
| TOGP | **503.0** | **0.1571** | 2.02 | **-18.1** | **0.1052** | **0.04** | **-3923.1** | **0.1372** | 2.82 |
| sMTGP | 749.4 | 0.1684 | **1.44** | -5.0 | 0.1566 | 0.06 | -3743.0 | 0.1501 | 22.01 |
| MLGP | 707937.1 | 0.9428 | 67.00 | 7066.9 | 0.8789 | 5.12 | -55800.7 | 1.1670 | **0.06** |
| MVGP | 11152.2 | 0.6746 | 22.20 | 46.54 | 0.1784 | 0.10 | -2583.1 | 1.0000 | 142.72 |

singular under our experimental configuration. sMTGP outperforms MVGP because it considers modeling each mode of the tensor output, and MVGP ignores the tensor structure by vectorizing it.

Table 3 summarizes the optimization performance of different methods for the BO and CBBO tasks. We set $N = 10d$ and evaluate each method using $\text{MSE}_x$, $\text{MAE}_y$, and Acc, as defined in Appendix K. For each surrogate, the UCB-based strategy consistently outperforms random sampling. This is

Table 3: The optimization performance of different methods in the three synthetic settings.

| | | Setting (1) | | | Setting (2) | | | Setting (3) | | |
|---|---|---|---|---|---|---|---|---|---|---|
| | | $\text{MSE}_x$ | $\text{MAE}_y$ | Acc | $\text{MSE}_x$ | $\text{MAE}_y$ | Acc | $\text{MSE}_x$ | $\text{MAE}_y$ | Acc |
| BO | TOBO | **0.0000** | **0.0008** | - | **0.0003** | **0.0350** | - | **0.0001** | **0.0050** | - |
| | sMTGP-UCB | 0.0001 | 0.0031 | - | **0.0003** | 0.0361 | - | 0.0048 | 0.0590 | - |
| | MLGP-UCB | 0.0433 | 0.3793 | - | 0.0512 | 0.9295 | - | 0.0342 | 0.6263 | - |
| | MVGP-UCB | 0.0015 | 0.0523 | - | 0.0044 | 0.0550 | - | 0.0342 | 0.6263 | - |
| | TOGP-RS | 0.0893 | 0.3145 | - | 0.0026 | 0.0351 | - | 0.0106 | 0.1526 | - |
| | sMTGP-RS | 0.0251 | 0.3242 | - | 0.0206 | 0.3684 | - | 0.0084 | 0.1223 | - |
| | MLGP-RS | 0.0433 | 0.3793 | - | 0.0435 | 0.7976 | - | 0.0075 | 0.0934 | - |
| | MVGP-RS | 0.0148 | 0.2036 | - | 0.0157 | 0.2697 | - | 0.0084 | 0.1223 | - |
| CBBO | TOCBBO | **0.0023** | **0.0172** | **1.00** | **0.0000** | **0.0000** | **1.00** | **0.0021** | **0.0145** | **1.00** |
| | sMTGP-UCB | 0.1832 | 0.5614 | 0.67 | **0.0000** | **0.0000** | **1.00** | 0.0075 | 0.2171 | 0.86 |
| | MLGP-UCB | 0.0667 | 0.6527 | 0.33 | 0.1826 | 0.0779 | **1.00** | 0.2070 | 0.7105 | 0.43 |
| | MVGP-UCB | 0.0032 | 0.0285 | **1.00** | 0.0725 | 0.0312 | **1.00** | 0.0151 | 0.1988 | 0.71 |
| | TOGP-RS | 0.0438 | 0.5319 | 0.67 | 0.0908 | 0.0395 | **1.00** | 0.0512 | 0.8793 | 0.43 |
| | sMTGP-RS | 0.3151 | 0.5882 | 0.67 | 0.1826 | 0.0779 | **1.00** | 0.0117 | 0.9453 | 0.29 |
| | MLGP-RS | 0.1053 | 0.6909 | 0.33 | 0.1826 | 0.0779 | **1.00** | 0.0117 | 0.9453 | 0.29 |
| | MVGP-RS | 0.1313 | 0.5975 | 0.33 | 0.1489 | 0.0654 | 0.00 | 0.0512 | 0.8793 | 0.43 |

intuitive due to UCB's better theoretical guarantees. Across all GPs, TOBO and TOCBBO achieve the smallest $\text{MSE}_x$ and $\text{MAE}_y$, indicating that the selected pairs yield objective values closest to the true optimum. Among the baseline methods, sMTGP-UCB delivers the second-best performance, followed by MVGP-UCB, while MLGP-UCB performs the worst. This result is consistent with their modeling abilities shown in Table 2.

Finally, we provide each round's logarithmic instantaneous regret for different methods for BO and CBBO in Figure 1. We can observe that TOBO and TOCBBO consistently achieve the lowest

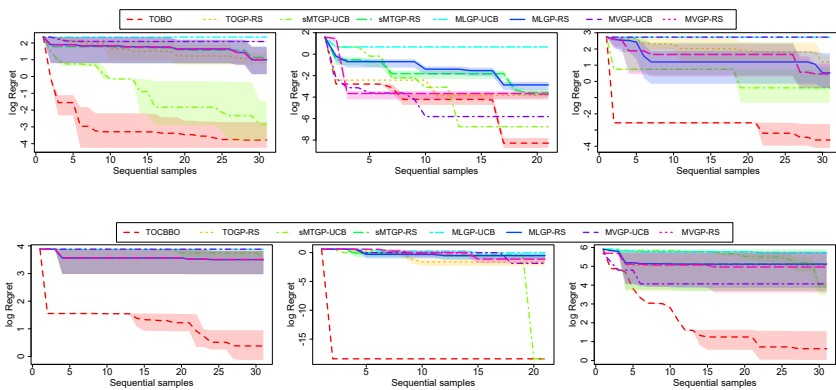

Figure 1: Each round's logarithmic instantaneous regret of different methods in the Setting (1) (L), (2) (M), and (3) (R) for BO (Top row) and CBBO (Bottom row).

instantaneous regret across all three settings, highlighting their superiority. Additional synthetic results are provided in Appendix M.

## 5.2 CASE STUDIES

We further apply apply TOBO and TOCBBO to four real-world datasets: (1) **CHEM** (Shields et al., 2021): input $x \in \mathbb{R}^2$ and output $\mathbf{y} \in \mathbb{R}^{4 \times 3 \times 3}$; (2) **MAT** (Wang et al., 2020): input $x \in \mathbb{R}^4$ and output $\mathbf{y} \in \mathbb{R}^{5 \times 4 \times 4}$; (3) **PRINT** (Zhai et al., 2023): input $x \in \mathbb{R}^5$ and output $\mathbf{y} \in \mathbb{R}^{3 \times 4 \times 3}$; (4) **REEN**: input $x \in \mathbb{R}^6$ and output $\mathbf{y} \in \mathbb{R}^{10 \times 2}$. Detailed description of these datasets is provided in Appendix K.

Since the REEN dataset provides fully observed tensor outputs, we first evaluate surrogate modeling performance by randomly selecting 30 samples for training and 5 samples for testing. Table 4 reports the predictive performance of the four GP surrogates on the test set. TOGP achieves the best accuracy in terms of NLL and MAE. Figure 2 further compares optimization performance on both BO (left) and CBBO (right), where TOBO and TOCBBO consistently perform the best, demonstrating their effectiveness on real-world black-box systems.

Table 4: The prediction performance of GPs in the REEN dataset.

|  | TOGP | sMTGP | MLGP | MVGP |
|---|---|---|---|---|
| NLL | **15.6664** | 33.3198 | 88.2722 | 48.7167 |
| MAE | **0.0883** | 0.0993 | 0.0929 | 0.1054 |
| ‖Cov‖ | 0.4918 | **0.3555** | 0.4318 | 0.3711 |

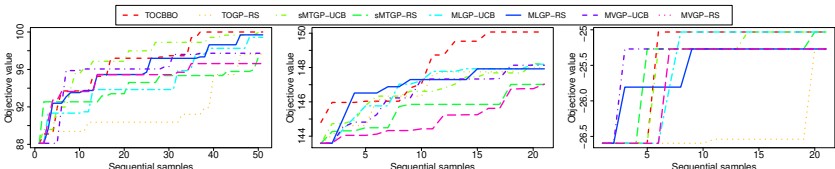

Figure 2: Each round's optimal objective value in REEN for BO (L) and CBBO (R).

The remaining three datasets contain only partially observed tensor outputs, we only evaluate optimization performance under the CBBO setting. Figure 2 shows that TOCBBO can identify the optimal input consistently using fewer rounds than the baselines, further demonstrating its superior effectiveness.

Figure 3: Each round's optimal objective value in CHEM (L), MAT (M), and PRINT (R).

## 6 CONCLUSION AND DISCUSSION

In this work, we develop two BO methods for tensor-output systems. TOBO employs two classes of kernel-based TOGP as a surrogate model and selects query points via a UCB acquisition function. TOCBBO extends TOGP to the partially observed tensor outputs and adopts a CMAB-UCB2 criterion to sequentially select both the query input and the super-arm. We establish theoretical regret bounds for both methods and demonstrate their effectiveness on extensive synthetic and real-world experiments. Future work could consider integrating the proposed tensor-output kernels with sparse techniques, such as sparse GPs (Snelson & Ghahramani, 2005) and scalable LMC (Bruinsma et al., 2020), to improve the computational efficiency of TOGP. It would be valuable to explore alternative acquisition functions within this framework. For example, one may combine the TOGP model with improvement-based criteria such as EI or PI (Frazier, 2018), and further extend them to the TOCBBO setting. Moreover, acquisition functions based on improvement (Uhrenholt & Jensen, 2019) or information-theoretic criteria (Tu et al., 2022) are also be considered in our framework, provided that the computational challenges for tensor outputs can be addressed. Finally, it is worth exploring more meaningful tensor structures for our proposed framework, such as a spatiotemporal system.

## ACKNOWLEDGMENTS

This work was supported by National Natural Science Foundation of China (Grant No.72271138) and Tsinghua-National University of Singapore Joint Funding (Grant No.20243080039). We gratefully acknowledge this support.

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

## A    USE OF LLMs

Large Language Models (LLMs) were used to aid in the writing and polishing of the manuscript. Specifically, we used an LLM to assist in refining the language, improving readability, and ensuring clarity in various sections of the paper. The model helped with tasks such as sentence rephrasing, grammar checking, and enhancing the overall flow of the text.

It is important to note that the LLM was not involved in the ideation, research methodology, or experimental design. All research concepts, ideas, and analyses were developed and conducted by the authors. The contributions of the LLM were solely focused on improving the linguistic quality of the paper, with no involvement in the scientific content or data analysis.

The authors take full responsibility for the content of the manuscript, including any text generated or polished by the LLM. We have ensured that the LLM-generated text adheres to ethical guidelines and does not contribute to plagiarism or scientific misconduct.

## B    THE DETAILS OF THE TENSOR DECOMPOSITION IN REMARK 1

As discussed in Remark 1, directly estimating the entries of the core tensors $\{\mathbf{A}_l\}_{l=1}^{m}$ or $\mathbf{A}$ requires $mT$ parameters for the non-separable tensor-output kernel and $T$ parameters for the separable tensor-output kernel, which becomes intractable for large-scale or high-order tensors. To reduce the parameter complexity, we impose low-rank tensor structures and adopt different decompositions depending on the tensor order.

**Low-order tensors** ($m \leq 3$):    When the tensor order is small, we employ the CP decomposition:

$$\mathbf{A}_l = \sum_{r=1}^{R_l} \boldsymbol{a}_{lr1} \circ \boldsymbol{a}_{lr2} \circ \cdots \circ \boldsymbol{a}_{lrm} \quad \text{for the non-separable kernel,} \tag{17}$$

$$\mathbf{A} = \sum_{r=1}^{R} \boldsymbol{a}_{r1} \circ \boldsymbol{a}_{r2} \circ \cdots \circ \boldsymbol{a}_{rm} \quad \text{for the separable kernel,} \tag{18}$$

where $\boldsymbol{a}_{lri}, \boldsymbol{a}_{ri} \in \mathbb{R}^{t_l}$ for $r = 1, \ldots, R$ and $i = 1, \ldots, m$ for $r = 1, \ldots, R$ and $i = 1, \ldots, m$. The number of free parameters becomes $\sum_{l=1}^{m} \sum_{i=1}^{m} R_l t_i$ for the non-separable kernel and $R \sum_{i=1}^{m} t_i$ for the separable kernel, which scales linearly with each mode size $t_i$. Thus, CP provides a compact representation when $m \leq 3$. However, CP can be ill-posed for higher-order tensors because the set of tensors with bounded CP rank is not closed, and a best low-rank approximation may fail to exist (De Silva & Lim, 2008). Moreover, the factor matrices can easily become ill-conditioned as $m$ increases, leading to numerical instability (Chi & Kolda, 2012).

**High-order tensors** ($m > 3$):    When the tensor order is large, we employ the TT decomposition:

$$\mathbf{A}_l = \mathbf{G}_{l1}(t_1)\mathbf{G}_{l2}(t_2) \cdots \mathbf{G}_{lm}(t_m) \quad \text{for the non-separable kernel,} \tag{19}$$

$$\mathbf{A} = \mathbf{G}_1(t_1)\mathbf{G}_2(t_2) \cdots \mathbf{G}_m(t_m) \quad \text{for the separable kernel,} \tag{20}$$

where each $\mathbf{G}_{lj}(t_j)$ is a $r_{l,j-1} \times t_j \times r_{l,j}$ three-mode tensor and the TT-ranks satisfy $r_{l,0} = r_{l,m} = 1$ for $l = 1, \ldots, m$. Similarly, $\mathbf{G}_j(t_j)$ is a $r_{j-1} \times t_j \times r_j$ three-mode tensor that satisfies $r_0 = r_m = 1$. The total number of free parameters is $\sum_{l=1}^{m} \sum_{j=1}^{m} r_{l,j-1} t_j r_{l,j}$ for the non-separable tensor-output kernel in (6) and $\sum_{j=1}^{m} r_{j-1} t_j r_j$ for the separable tensor-output kernel in (7). It scales linearly with the tensor order $m$, instead of exponentially as in the full tensor. This makes TT decomposition highly suitable for high-order tensors ($m > 3$), as it balances modeling flexibility with computational scalability and avoids the instability of CP.

In summary, we use CP for low-order cores and TT for high-order cores, ensuring both efficiency and numerical robustness across different tensor settings.

## C    THE ESTIMATION OF HYPERPARAMETERS FOR THE TOGP AND PTOGP

In this appendix, we provide the details of hyperparameter estimation for both TOGP and PTOGP. Without loss of generality, we assume a zero prior mean $\boldsymbol{\mu} = 0$ in (3) and (12).

### C.1 PARAMETER ESTIMATION FOR TRAINING TOGP

Given the training data $\mathbf{X}_n = (\boldsymbol{x}_1, \ldots, \boldsymbol{x}_n)$ and $\boldsymbol{Y}_n = \left(vec(\mathbf{y}_1)^\top, \ldots, vec(\mathbf{y}_n)^\top\right)^\top$, the log marginal likelihood of TOGP is given by:

$$\log L(\boldsymbol{\Theta}) = -\frac{1}{2}\log|\boldsymbol{\Sigma}_n| - \frac{1}{2}\boldsymbol{Y}_n^\top \boldsymbol{\Sigma}_n^{-1}\boldsymbol{Y}_n, \tag{21}$$

where $\boldsymbol{\Sigma}_n = \sigma^2 \mathbf{K}_n + \tau^2 \mathbf{I}_{nT}$. Then, (21) can be optimized by applying gradient-based optimization methods, such as L-BFGS algorithm.

The gradients of the log-likelihood function in (21) with respect to the hyperparameters $\tau^2$, $\sigma^2$, $\boldsymbol{\theta}$, and $\mathbf{a}$ is given by

$$\frac{\partial \log L}{\partial \tau^2} = \frac{1}{2}\text{tr}\left(\boldsymbol{\Sigma}_n^{-1}\mathbf{K}_n\right) - \frac{1}{2}\boldsymbol{Y}_n^\top \boldsymbol{\Sigma}_n^{-1}\mathbf{K}_n\boldsymbol{\Sigma}_n^{-1}\boldsymbol{Y}_n, \tag{22}$$

$$\frac{\partial \log L}{\partial \sigma^2} = \frac{1}{2}\text{tr}\left(\boldsymbol{\Sigma}_n^{-1}\right) - \frac{1}{2}\boldsymbol{Y}_n^\top \boldsymbol{\Sigma}_n^{-1}\boldsymbol{\Sigma}_n^{-1}\boldsymbol{Y}_n, \tag{23}$$

$$\frac{\partial \log L}{\partial \theta_{lij}} = \frac{\tau^2}{2}\text{tr}\left(\boldsymbol{\Sigma}_n^{-1}\frac{\partial \mathbf{K}_n}{\partial \theta_{lij}}\right) - \frac{\tau^2}{2}\boldsymbol{Y}_n^\top \boldsymbol{\Sigma}_n^{-1}\frac{\partial \mathbf{K}_n}{\partial \theta_{lij}}\boldsymbol{\Sigma}_n^{-1}\boldsymbol{Y}_n, \tag{24}$$

$$\frac{\partial \log L}{\partial a_{lij}} = \frac{1}{2}\text{tr}\left(\boldsymbol{\Sigma}_n^{-1}\tau^2\frac{\partial \mathbf{K}_n}{\partial a_{lij}}\right) - \frac{\tau^2}{2}\boldsymbol{Y}_n^\top \boldsymbol{\Sigma}_n^{-1}\frac{\partial \mathbf{K}_n}{\partial a_{lij}}\boldsymbol{\Sigma}_n^{-1}\boldsymbol{Y}_n, \tag{25}$$

where $\theta_{lij}$ represents the scale parameters of $k_{lj}$, the kernel function associated with the $l$-th mode. The matrices $\frac{\partial \mathbf{K}_n}{\partial \theta_{lij}}$ and $\frac{\partial \mathbf{K}_n}{\partial a_{lij}}$ are the partial derivatives of the kernel matrix with respect to the corresponding kernel parameters.

The detailed algorithm for training TOGP is given as follows:

---

**Algorithm 1** Parameter estimation for training TOGP

---

**Input:** Training data $\mathbf{X}_n$ and $\boldsymbol{Y}_n$, initial hyperparameters $\boldsymbol{\Theta}_0 = \{\sigma_0^2, \tau_0^2, \boldsymbol{\theta}_0, \mathbf{a}_0\}$;
**Initialize:** $\sigma^2 \leftarrow \sigma_0^2, \tau^2 \leftarrow \tau_0^2, \boldsymbol{\theta} \leftarrow \boldsymbol{\theta}_0 \; \mathbf{a} \leftarrow \mathbf{a}_0$;
 1: **while** $\tau^2, \sigma^2, \boldsymbol{\theta}, \boldsymbol{\omega}$ not converge **do**
 2:     Update $\tau^2$ based on (22);
 3:     Update $\sigma^2$ based on (23);
 4:     Update $\boldsymbol{\theta}$ based on (24);
 5:     Update $\mathbf{a}$ based on (25).
 6: **end while**

---

**Remark 4.** *For the non-separable tensor-output kernel in Definition 1, the total number of hyperparameters to be estimated in TOGP is $m_h = 2 + T + \sum_{l=1}^m \sum_{i=1}^m R_l t_i$ when $m \leq 3$ and $m_h = 2 + T + \sum_{l=1}^m \sum_{j=1}^m r_{l,j-1} t_j r_{l,j}$ when $m > 3$. For the separable tensor-output kernel in Definition 2, the total number of hyperparameters to be estimated in TOGP is $m_h = 2 + T + R\sum_{i=1}^m t_i$ when $m \leq 3$ and $m_h = 2 + T + \sum_{j=1}^m r_{j-1} t_j r_j$ when $m > 3$. The computational complexity of computing the gradient of $\log L(\boldsymbol{\Theta})$ with respect to all $m_h$ parameters is $\mathcal{O}(n^3 T^3 + n^2 T^2 m_h)$. When using the L-BFGS algorithm to optimize the likelihood function results, the number of iterations typically scales as $\mathcal{O}(\log(n))$ (Bottou, 2010). Therefore, the overall computational complexity for training the TOGP takes $\mathcal{O}\left(n^3 T^3 \log(n) + n^2 T^2 m_h \log(n)\right)$ computational complexity.*

### C.2 PARAMETER ESTIMATION FOR TRAINING PTOGP

Given the partially observed training data $\mathbf{X}_n = (\boldsymbol{x}_1, \ldots, \boldsymbol{x}_n)$, $\boldsymbol{\Lambda}_n = (\boldsymbol{\lambda}_1, \ldots, \boldsymbol{\lambda}_n)^\top$, and $\tilde{\boldsymbol{Y}}_n = (\tilde{\boldsymbol{y}}_1, \ldots, \tilde{\boldsymbol{y}}_n)^\top$, the log marginal likelihood of PTOGP is given by:

$$\log \tilde{L}(\boldsymbol{\Theta}) = -\frac{1}{2}\log|\tilde{\boldsymbol{\Sigma}}_n| - \frac{1}{2}\text{vec}(\tilde{\boldsymbol{Y}}_n)^\top \tilde{\boldsymbol{\Sigma}}_n^{-1}\text{vec}(\tilde{\boldsymbol{Y}}_n), \tag{26}$$

where $\tilde{\boldsymbol{\Sigma}}_n = \sigma^2 \mathbf{E}_n \mathbf{K}_n \mathbf{E}_n^\top + \tau^2 \mathbf{I}_{nk}$. Then, (26) can also be optimized by applying gradient-based optimization methods.

The gradients of the log-likelihood function in (26) with respect to the hyperparameters $\sigma^2$, $\tau^2$, $\boldsymbol{\theta}$, and $\mathbf{a}$ is given by

$$\frac{\partial \tilde{L}}{\partial \tau^2} = \frac{1}{2}\mathrm{tr}\left(\tilde{\boldsymbol{\Sigma}}_n^{-1}\mathbf{E}_n\mathbf{K}_n\mathbf{E}_n^\top\right) - \frac{1}{2}\tilde{Y}_n^\top\tilde{\boldsymbol{\Sigma}}_n^{-1}\mathbf{E}_n\mathbf{K}_n\mathbf{E}_n^\top\tilde{\boldsymbol{\Sigma}}_n^{-1}\tilde{Y}_n, \tag{27}$$

$$\frac{\partial \tilde{L}}{\partial \sigma^2} = \frac{1}{2}\mathrm{tr}\left(\tilde{\boldsymbol{\Sigma}}_n^{-1}\right) - \frac{1}{2}\tilde{Y}_n^\top\tilde{\boldsymbol{\Sigma}}_n^{-1}\tilde{\boldsymbol{\Sigma}}_n^{-1}\tilde{Y}_n, \tag{28}$$

$$\frac{\partial \tilde{L}}{\partial \theta_{lij}} = \frac{\tau^2}{2}\mathrm{tr}\left(\tilde{\boldsymbol{\Sigma}}_n^{-1}\mathbf{E}_n\frac{\partial \mathbf{K}_n}{\partial \theta_{lij}}\mathbf{E}_n^\top\right) - \frac{\tau^2}{2}\tilde{Y}_n^\top\tilde{\boldsymbol{\Sigma}}_n^{-1}\mathbf{E}_n\frac{\partial \mathbf{K}_n}{\partial \theta_{lij}}\mathbf{E}_n^\top\tilde{\boldsymbol{\Sigma}}_n^{-1}\tilde{Y}_n, \tag{29}$$

$$\frac{\partial \tilde{L}}{\partial a_{lij}} = \frac{\tau^2}{2}\mathrm{tr}\left(\tilde{\boldsymbol{\Sigma}}_n^{-1}\mathbf{E}_n\frac{\partial \mathbf{K}_n}{\partial a_{lij}}\mathbf{E}_n^\top\right) - \frac{\tau^2}{2}\tilde{Y}_n^\top\tilde{\boldsymbol{\Sigma}}_n^{-1}\mathbf{E}_n\frac{\partial \mathbf{K}_n}{\partial a_{lij}}\mathbf{E}_n^\top\tilde{\boldsymbol{\Sigma}}_n^{-1}\tilde{Y}_n, \tag{30}$$

The detailed algorithm for training PTOGP is given as follows:

---

**Algorithm 2** Parameter estimation for training PTOGP

---

**Input:** Training data $\mathbf{X}_n$, $\boldsymbol{\Lambda}_n$ and $\tilde{Y}_n$, initial hyperparameters $\boldsymbol{\Theta}_0 = \{\sigma_0^2, \tau_0^2, \boldsymbol{\theta}_0, \mathbf{a}_0\}$;
**Initialize:** $\sigma^2 \leftarrow \sigma_0^2, \tau^2 \leftarrow \tau_0^2, \boldsymbol{\theta} \leftarrow \boldsymbol{\theta}_0\ \mathbf{a} \leftarrow \mathbf{a}_0$;
 1: **while** $\sigma^2, \tau^2, \boldsymbol{\theta}, \mathbf{a}$ not converge **do**
 2:     Update $\tau^2$ based on (27);
 3:     Update $\sigma^2$ based on (28);
 4:     Update $\boldsymbol{\theta}$ based on (29);
 5:     Update $\mathbf{a}$ based on (30).
 6: **end while**

---

**Remark 5.** *For the non-separable tensor-output kernel in Definition 1, the total number of hyperparameters to be estimated in PTOGP is $m_h = 2 + T + \sum_{l=1}^m \sum_{i=1}^m R_l t_i$ when $m \leq 3$ and $m_h = 2 + T + \sum_{l=1}^m \sum_{j=1}^m r_{l,j-1} t_j r_{l,j}$ when $m > 3$. For the separable tensor-output kernel in Definition 2, the total number of hyperparameters to be estimated in PTOGP is $m_h = 2 + T + R\sum_{i=1}^m t_i$ when $m \leq 3$ and $m_h = 2 + T + \sum_{j=1}^m r_{j-1} t_j r_j$ when $m > 3$. The computational complexity of computing the gradient of $\tilde{L}(\boldsymbol{\Theta})$ with respect to all $m_h$ parameters is $\mathcal{O}(k^3 T^3 + n^2 kT m_h)$. When using the L-BFGS algorithm to optimize the likelihood function results, the number of iterations typically scales as $\mathcal{O}(\log(n))$ Bottou (2010). Therefore, the overall computational complexity for training the PTOGP takes $\mathcal{O}\left(n^3 k^3 \log(n) + n^2 kT m_h \log(n)\right)$ computational complexity.*

## D   THE PROPOSED ALGORITHMS AND COMPUTATIONAL COMPLEXITY ANALYSIS

### D.1   UCB-BASED TOBO ALGORITHM

The detailed procedure of the proposed TOBO method is given in Algorithm 3.

---

**Algorithm 3** UCB-based TOBO

---

**Input:** Total rounds $N$, initial dataset $\mathcal{D}_0 = \varnothing$, initial hyperparameters $\boldsymbol{\Theta}_0$;
 1: **for** round $n = 1, \ldots, N$ **do**
 2:     Update the posterior mean (4) and covariance (5) of $\mathbf{f}$ given $\boldsymbol{\Theta}_{n-1}$;
 3:     Select the next input $\boldsymbol{x}_n \leftarrow \arg\max_{\boldsymbol{x} \in \mathcal{X}} \alpha_{UCB}(\boldsymbol{x} \mid \mathcal{D}_{n-1})$;
 4:     Evaluate the black-box system and observe output $\mathbf{y}_n$;
 5:     Update dataset $\mathcal{D}_n \leftarrow \mathcal{D}_{n-1} \cup \{\boldsymbol{x}_n, \mathbf{y}_n\}$;
 6:     Update hyperparameters $\boldsymbol{\Theta}_n$ by maximizing (21) with L-BFGS;
 7: **end for**
 8: Identify $i^\star = \arg\max_{i \in [N]} L_f \mathbf{f}(\boldsymbol{x}_i)$;
 9: **Output:** Optimal input $\boldsymbol{x}_{i^\star}$ and corresponding output $\mathbf{y}_{i^\star}$.

---

**Remark 6.** *When using the TOBO method to select $\boldsymbol{x}^\star$, the computational complexity of updating TOGP is $\mathcal{O}\left((n-1)^3 T^3\right)$ at round $n$. Then, the computational complexity of querying the next point is $\mathcal{O}\left((n-1)^2 T^2 \log(n)\right)$ by using the L-BFGS method. After updating the design dataset, the computational complexity of updating the hyperparameters $\boldsymbol{\Theta}$ is $\mathcal{O}\left(n^3 T^3 \log(n) + n^2 T^2 m_h \log(n)\right)$. Thus, the computational complexity of TOBO at round $n$ is $\mathcal{O}\left(n^3 T^3 \log(n) + n^2 T^2 m_h \log(n)\right)$. Suppose that there are $N$ points needed to query, the total computational complexity of Algorithm 3 is $\mathcal{O}\left(\sum_{n=1}^{N}\left[n^3 T^3 \log(n) + n^2 T^2 m_h \log(n)\right]\right)$.*

## D.2 CMAB-UCB2-BASED TOCBBO ALGORITHM

The detailed algorithm of TOCBBO is provided in Algorithm 4.

---

**Algorithm 4** CMAB-UCB2-based TOCBBO

---

**Input:** Total rounds $N$, initial dataset $\tilde{\mathcal{D}}_0 = \varnothing$, initial hyperparameters $\boldsymbol{\Theta}_0$;
1: **for** round $n = 1, \ldots, N$ **do**
2:      Update the posterior mean (13) and covariance (14);
3:      Select the next input $\boldsymbol{x}_n$ using (15);
4:      Select the super-arm $\boldsymbol{\lambda}_n$ using (16);
5:      Evaluate $\mathbf{f}$ under $(\boldsymbol{x}_n, \boldsymbol{\lambda}_n)$ and observe $\tilde{\boldsymbol{y}}_n$;
6:      Update dataset $\tilde{\mathcal{D}}_n \leftarrow \tilde{\mathcal{D}}_{n-1} \cup \{(\boldsymbol{x}_n, \boldsymbol{\lambda}_n), \tilde{\boldsymbol{y}}_n\}$;
7:      Update hyperparameters $\boldsymbol{\Theta}_n$ by maximizing $\log \tilde{L}(\boldsymbol{\Theta})$ with L-BFGS;
8:      Update incumbent solution $\{\boldsymbol{x}_n^\star, \boldsymbol{\lambda}_n^\star\} = \arg\max_{i=1,\ldots,n} H_f \tilde{\boldsymbol{f}}(\boldsymbol{x}_i, \boldsymbol{\lambda}_i)$;
9: **end for**
10: Identify $i^\star = \arg\max_{i \in [N]} H_f \tilde{\boldsymbol{f}}(\boldsymbol{x}_i, \boldsymbol{\lambda}_i)$;
11: **Output:** Optimal input $\boldsymbol{x}_{i^\star}$, optimal super-arm $\boldsymbol{\lambda}_{i^\star}$, and output $\tilde{\boldsymbol{y}}_{i^\star}$.

---

**Remark 7.** *When using the proposed TOCBBO method to jointly select $\boldsymbol{x}^\star$ and $\boldsymbol{\lambda}^\star$, the computational complexity of updating the PTOGP at round $n$ is $\mathcal{O}\left((n-1)^3 k^3\right)$. Then, based on the proposed CMAB-UCB2 criterion, the computational complexity of querying the next input by using L-BFGS method is $\mathcal{O}\left((n-1)^2 k^2\right)$ and selecting the next super-arm by using greedy Top-$k$ method is $\mathcal{O}(kT^3)$, respectively. After updating the current design dataset, the computational complexity of updating hyperparameters $\boldsymbol{\Theta}$ is $\mathcal{O}(n^3 k^3 \log(n) + n^2 kT m_h \log(n))$. Therefore, at round $n$, the computational complexity of TOCBBO method is $\mathcal{O}(n^3 k^3 \log(n) + n^2 kT m_h \log(n) + kT^3)$. Assuming a total of $N$ rounds, the overall computational complexity of Algorithm 4 is $\mathcal{O}\left(\sum_{n=1}^{N}\left[n^3 k^3 \log(n) + n^2 kT m_h \log(n) + kT^3\right]\right)$.*

## E THE PROOF OF PROPOSITION 1

*Proof.* We prove symmetry and positive semi-definite for the two kernel classes in Definitions 1–2.

**Non-separable kernel (Definition 1):** Denote $\boldsymbol{a}_\ell := \text{vec}(\mathbf{A}_\ell) \in \mathbb{R}^T$ (resp. $\boldsymbol{a} := \text{vec}(\mathbf{A}) \in \mathbb{R}^T$). The tensor-output kernel is

$$\mathbf{K}(\boldsymbol{x}, \boldsymbol{x}') = \sum_{\ell=1}^{m} \sum_{j=1}^{t_\ell} \boldsymbol{a}_\ell \boldsymbol{a}_\ell^\top \, k_{\ell j}(\boldsymbol{x}, \boldsymbol{x}'), \qquad \boldsymbol{x}, \boldsymbol{x}' \in \mathcal{X}, \tag{31}$$

where each $k_{\ell j}$ is a scalar positive semi-definite kernel.

For any $\boldsymbol{x}, \boldsymbol{x}' \in \mathcal{X}$, we have

$$\mathbf{K}(\boldsymbol{x}, \boldsymbol{x}')^\top = \sum_{\ell, j} \left(\boldsymbol{a}_\ell \boldsymbol{a}_\ell^\top\right)^\top k_{\ell j}(\boldsymbol{x}, \boldsymbol{x}') = \sum_{\ell, j} \boldsymbol{a}_\ell \boldsymbol{a}_\ell^\top \, k_{\ell j}(\boldsymbol{x}, \boldsymbol{x}')$$

$$= \sum_{\ell, j} \boldsymbol{a}_\ell \boldsymbol{a}_\ell^\top \, k_{\ell j}(\boldsymbol{x}', \boldsymbol{x}) = \mathbf{K}(\boldsymbol{x}', \boldsymbol{x}),$$

This shows that the full tensor-output kernel $\mathbf{K}$ is symmetric.

Let $\{\boldsymbol{x}_i\}_{i=1}^n \subset \mathcal{X}$ and $\{\boldsymbol{y}_i\}_{i=1}^n \subset \mathbb{R}^T$ be arbitrary. Then, we have

$$
\begin{aligned}
\sum_{i,j=1}^n \boldsymbol{y}_i^\top \mathbf{K}(\boldsymbol{x}_i, \boldsymbol{x}_j) \boldsymbol{y}_j &= \sum_{\ell,j} \sum_{i,j} \boldsymbol{y}_i^\top \big(\boldsymbol{a}_\ell \boldsymbol{a}_\ell^\top\big) \boldsymbol{y}_j \, k_{\ell j}(\boldsymbol{x}_i, \boldsymbol{x}_j) \\
&= \sum_{\ell,j} \sum_{i,j} s_{\ell j,i} \, k_{\ell j}(\boldsymbol{x}_i, \boldsymbol{x}_j) \, s_{\ell j,j},
\end{aligned}
\tag{32}
$$

where $s_{\ell j,i} := \boldsymbol{a}_\ell^\top \boldsymbol{y}_i \in \mathbb{R}$. For each fixed $(\ell, j)$, the matrix $\big[k_{\ell j}(\boldsymbol{x}_i, \boldsymbol{x}_j)\big]_{i,j=1}^n$ is positive semi-definite, so $\sum_{i,j} s_{\ell j,i} \, k_{\ell j}(\boldsymbol{x}_i, \boldsymbol{x}_j) \, s_{\ell j,j} \geq 0$. Summing over $(\ell, j)$ preserves nonnegativity, hence the Gram matrix induced by $\mathbf{K}$ is positive semi-definite.

**Separable kernel (Definition 2):**  The kernel is

$$
\mathbf{K}(\boldsymbol{x}, \boldsymbol{x}') = \boldsymbol{a} \, \boldsymbol{a}^\top \, k(\boldsymbol{x}, \boldsymbol{x}'), \qquad \boldsymbol{x}, \boldsymbol{x}' \in \mathcal{X},
\tag{33}
$$

where $k$ is a scalar positive semi-definite kernel. Since $\big(\boldsymbol{a}\boldsymbol{a}^\top\big)^\top = \boldsymbol{a}\boldsymbol{a}^\top$ and $k(\boldsymbol{x}, \boldsymbol{x}') = k(\boldsymbol{x}', \boldsymbol{x})$, we have $\mathbf{K}(\boldsymbol{x}, \boldsymbol{x}')^\top = \mathbf{K}(\boldsymbol{x}', \boldsymbol{x})$. For arbitrary $\{\boldsymbol{x}_i\}_{i=1}^n$ and $\{\boldsymbol{y}_i\}_{i=1}^n$, $\sum_{i,j=1}^n \boldsymbol{y}_i^\top \mathbf{K}(\boldsymbol{x}_i, \boldsymbol{x}_j) \boldsymbol{y}_j = \sum_{i,j} \big(\boldsymbol{a}^\top \boldsymbol{y}_i\big) k(\boldsymbol{x}_i, \boldsymbol{x}_j) \big(\boldsymbol{a}^\top \boldsymbol{y}_j\big) \geq 0$, because the Gram matrix $\big[k(\boldsymbol{x}_i, \boldsymbol{x}_j)\big]$ is positive semi-definite.

Therefore, both kernel classes are symmetric and generate positive semi-definite Gram matrices for any finite set of inputs, i.e., they are valid tensor-output kernels on $\mathcal{X}$. $\qquad\square$

## F  THE PROOF OF LEMMA 1

*Proof.* First, for a tensor-output system $\mathrm{vec}(\mathbf{f}(\boldsymbol{x}))$ following a TOGP defined in (3), denote the derivative field of $\mathrm{vec}(\mathbf{f}(\boldsymbol{x}))$ to the $j$-coordinate element of $\boldsymbol{x}$ as $\boldsymbol{g}_j(\boldsymbol{x}) := \partial \mathrm{vec}(\mathbf{f}(\boldsymbol{x}))/\partial x_j$, where $\mathrm{vec}\big(\mathbf{f}(\boldsymbol{x})\big) \in \mathbb{R}^T$ and $j \in \{1, \dots, d\}$. According to the derivative property of GP (Santner et al., 2003), we have that $\boldsymbol{g}_j(\boldsymbol{x}) \in \mathbb{R}^\top$ is a multivariate-output GP with mean $\hat{\boldsymbol{\mu}}_n^\nabla(x_j) := \frac{\partial}{\partial x_j} \boldsymbol{\mu}_n(\boldsymbol{x})$ and covariance $\hat{\mathbf{K}}_n^\nabla(x_j, x_j') := \mathrm{Cov}(\mathbf{g}_j(\boldsymbol{x}), \mathbf{g}_j(\boldsymbol{x}')) = \frac{\partial^2}{\partial x_j \partial x_j'} \sigma^2 \mathbf{K}(\boldsymbol{x}, \boldsymbol{x}')$.

At round $n+1$, the observed data is denoted as $\mathbf{X}_n$ and $\boldsymbol{Y}_n$, then we have the posterior distribution of $\boldsymbol{g}_j(\boldsymbol{x})$ is a $T$-dimensional Gaussian with mean and covariance

$$
\hat{\boldsymbol{\mu}}_n^\nabla(x_j) = \frac{\partial \hat{\boldsymbol{\mu}}_n(\boldsymbol{x})}{\partial x_j}; \quad \hat{\mathbf{K}}_n^\nabla(x_j, x_j') = \frac{\partial^2 \hat{\mathbf{K}}_n(\boldsymbol{x}, \boldsymbol{x})}{\partial x_j \, \partial x_j'}.
\tag{34}
$$

It is easy to verify that $\hat{\mathbf{K}}_n^\nabla(x_j, x_j')$ is positive semi-definite for every $\boldsymbol{x}$.

For any fixed $\boldsymbol{x} \in \mathcal{X}$, given $\mathbf{X}_n$ and $\boldsymbol{Y}_n$, the random vector

$$
\mathbf{g}_j(\boldsymbol{x}) - \hat{\boldsymbol{\mu}}_n^\nabla(x_j) \sim \mathcal{N}\big(\mathbf{0}, \hat{\mathbf{K}}_n^\nabla(\boldsymbol{x}, \boldsymbol{x}; j)\big).
\tag{35}
$$

Applying the Gaussian Lipschitz concentration (Proposition 2.5.2 and Theorem 5.2.2 in Papaspiliopoulos (2020)) to the norm $\|\cdot\|_2$ yields, for all $t > 0$,

$$
\Pr\bigg( \big\|\mathbf{g}_j(\boldsymbol{x}) - \hat{\boldsymbol{\mu}}_n^\nabla(x_j)\big\| \geq \sqrt{\mathrm{tr}\big(\hat{\mathbf{K}}_n^\nabla(x_j, x_j)\big)} + t \bigg) \leq \exp\bigg( -\frac{t^2}{2 \, \lambda_{\max}^{(n)}(x_j)} \bigg),
\tag{36}
$$

where $\lambda_{\max}^{(n)}(\boldsymbol{x}; j)$ is the largest eigenvalue of $\hat{\mathbf{K}}_n^\nabla(\boldsymbol{x}, \boldsymbol{x}; j)$.

Let $\mathcal{D}_M = \{\boldsymbol{x}_1, \dots, \boldsymbol{x}_M\} \subset \mathcal{X}$ be any finite discretization. Using (36) with $t(\boldsymbol{x}) = \sqrt{2 \, \lambda_{\max}^{(n)}(x_j) \, \log(M/\delta)}$, and applying the union bound, we obtain with probability at least $1 - \delta$,

$$
\big\|\mathbf{g}_j(\boldsymbol{x}) - \hat{\boldsymbol{\mu}}_n^\nabla(x_j)\big\| \leq \sqrt{\mathrm{tr}\big(\hat{\mathbf{K}}_n^\nabla(x_j, x_j)\big)} + \sqrt{2 \, \lambda_{\max}^{(n)}(x_j) \, \log(M/\delta)}, \qquad \forall \, \boldsymbol{x} \in \mathcal{D}_M.
\tag{37}
$$

Let $C_\nabla := \sup_{\boldsymbol{x} \in \mathcal{X}} \sqrt{\mathrm{tr}\big(\hat{\mathbf{K}}_n^\nabla(x_j, x_j)\big)}$ and $\Lambda_\nabla := \sup_{\boldsymbol{x} \in \mathcal{X}} \lambda_{max}^{(n)}(x_j)$, then (37) implies that, with probability at least $1 - \delta$,

$$\|\mathbf{g}_j(\boldsymbol{x})\| \leq \|\hat{\boldsymbol{\mu}}_n^\nabla(x_j)\| + C_\nabla + \sqrt{2\,\Lambda_\nabla\,\log(M/\delta)}, \qquad \forall\,\boldsymbol{x} \in \mathcal{D}_M. \tag{38}$$

Assume the kernel is sufficiently smooth so that $\mathbf{g}_j(\cdot)$ has almost surely Lipschitz sample paths. Then there exist absolute constants $a, b > 0$ (depending on the Lipschitz modulus and a covering-number bound of $\mathcal{X}$) such that for any $L' > 0$,

$$\Pr\left(\sup_{\boldsymbol{x} \in \mathcal{X}} \|\mathbf{g}_j(\boldsymbol{x})\| > L' + C_\nabla\right) \leq a\,\exp\left(-\frac{(L')^2}{b^2}\right). \tag{39}$$

This follows by combining the net bound (38) with a standard chaining argument to pass from a finite net to the full domain; the Gaussian tail is preserved with a possible adjustment of absolute constants into $a, b$. $\qquad\square$

## G  THE PROOF OF THEOREM 1

*Proof.* The proof consists of two main parts. We first establish a concentration inequality for $\mathrm{vec}(\mathbf{f}(\boldsymbol{x}))$ for any $\boldsymbol{x} \in \mathcal{X}$, and then use this result to derive an upper bound for the regret.

**Part 1. Concentration inequality.**  We begin by proving a concentration inequality for $\mathrm{vec}(\mathbf{f}(\boldsymbol{x}))$ evaluation on a discrete set of points in the domain $\mathcal{X}$. We then extend the result to neighborhoods of these discrete points, and ultimately to the entire space $\mathcal{X}$.

We first prove a basic concentration inequality for a general $T$-dimensional Gaussian distribution, i.e., $\boldsymbol{Z} \sim \mathcal{N}(\boldsymbol{\mu}, \mathbf{K})$. If we define $\boldsymbol{U} = \boldsymbol{Z} - \boldsymbol{\mu}$ and set another $T$-dimensional standard Gaussian distribution $\boldsymbol{V} \sim \mathcal{N}(0, \mathbf{I}_T)$, we can obtain

$$\boldsymbol{U} \overset{d}{=} \mathbf{K}^{1/2}\boldsymbol{V}, \quad \|\boldsymbol{U}\| = \|\mathbf{K}^{1/2}\boldsymbol{V}\|.$$

Define the function $f(\boldsymbol{U}) \triangleq \|\mathbf{K}^{1/2}\boldsymbol{V}\|$. For any $\boldsymbol{U}, \boldsymbol{U}'$, we have

$$|f(\boldsymbol{U}) - f(\boldsymbol{U}')| \leq \|\mathbf{K}^{1/2}(\boldsymbol{U} - \boldsymbol{U}')\| \leq \|\mathbf{K}^{1/2}\|_{op}\,\|\boldsymbol{U} - \boldsymbol{U}'\| = \sqrt{\lambda_{\max}(\mathbf{K})}\,\|\boldsymbol{U} - \boldsymbol{U}'\|,$$

where $\lambda_{\max}(\mathbf{K})$ is the maximum eigenvalue of $\mathbf{K}$. Therefore $f$ is a Lipschitz function with constant $L = \sqrt{\lambda_{\max}(\mathbf{K})}$.

According to Proposition 2.5.2 and Theorem 5.2.2 in Papaspiliopoulos (2020), we have the following concentration inequality for a Lipschitz function of a standard Gaussian distribution:

$$\Pr\left(f(\boldsymbol{U}) \geq \mathbb{E}[f(\boldsymbol{U})] + t\right) \leq \exp\left(-\frac{t^2}{2L^2}\right).$$

Substituting $f$ and $L$, we obtain

$$\Pr\left(\|\boldsymbol{Z} - \boldsymbol{\mu}\| \geq \mathbb{E}[\|\boldsymbol{Z} - \boldsymbol{\mu}\|] + t\right) \leq \exp\left(-\frac{t^2}{2\lambda_{\max}(\mathbf{K})}\right).$$

Since

$$\mathbb{E}[\|\boldsymbol{Z} - \boldsymbol{\mu}\|] \leq \sqrt{\mathbb{E}[\|\boldsymbol{Z} - \boldsymbol{\mu}\|^2]} = \sqrt{\mathrm{tr}(\mathbf{K})},$$

we get the final result for a general Gaussian distribution:

$$\Pr\left(\|\boldsymbol{Z} - \boldsymbol{\mu}\| \geq \sqrt{\mathrm{tr}(\mathbf{K})} + t\right) \leq \exp\left(-\frac{t^2}{2\lambda_{\max}(\mathbf{K})}\right). \tag{40}$$

Define the discrete set $\mathcal{D}_M = \{\boldsymbol{x}_1, \ldots, \boldsymbol{x}_M\} \subset \mathcal{X}$. According to the above concentration inequality, we have

$$\Pr\left(\|\mathrm{vec}(\mathbf{f}(\boldsymbol{x})) - \hat{\boldsymbol{\mu}}_n(\boldsymbol{x})\| > \sqrt{\mathrm{tr}(\hat{\mathbf{K}}_n(\boldsymbol{x}, \boldsymbol{x}))} + z\right) \leq \exp\left(-\frac{z^2}{2\lambda_{\max}^{(n)}(\boldsymbol{x})}\right), \quad \forall\,\boldsymbol{x} \in \mathcal{D}_M,$$

where $\lambda_{\max}^{(n)}(\boldsymbol{x})$ is the maximum eigenvalue of $\hat{\mathbf{K}}_n(\boldsymbol{x}, \boldsymbol{x})$.

By setting $z = \sqrt{2\lambda_{\max}^{(n)}(\boldsymbol{x}) \log \frac{M}{\delta}}$, we obtain

$$\Pr\left(\|\mathrm{vec}(\mathbf{f}(\boldsymbol{x})) - \hat{\boldsymbol{\mu}}_n(\boldsymbol{x})\| \leq \beta_n \sqrt{\lambda_{\max}^{(n)}(\boldsymbol{x})}\right) \geq 1 - \delta, \quad \forall \boldsymbol{x} \in \mathcal{D}_M, \tag{41}$$

where $\beta_n = \sqrt{\sup_{\boldsymbol{x} \in \mathcal{X}} \frac{\mathrm{tr}(\hat{\mathbf{K}}_n(\boldsymbol{x}, \boldsymbol{x}))}{\lambda_{\max}^{(n)}(\boldsymbol{x})}} + \sqrt{2 \log \frac{M}{\delta}} = \sqrt{C_n} + \sqrt{2 \log \frac{M}{\delta}}$.

From Lemma 1, we obtain that $\|\mathrm{vec}(\mathbf{f}(\boldsymbol{x})) - \mathrm{vec}(\mathbf{f}(\boldsymbol{x}'))\| \leq (L' + C^\nabla)\|\boldsymbol{x} - \boldsymbol{x}'\|_1$ holds with probability at least $1 - a \exp(-L'^2/b^2)$ for all $\boldsymbol{x}, \boldsymbol{x}' \in \mathcal{X}$.

Then, at round $n$, we set the size of $\mathcal{D}_{M(n)}$ as $(\tau_n)^d$, i.e., $M(n) = (\tau_n)^d$. For $\boldsymbol{x} \in \mathcal{D}_{M(n)}$, we have $\|\boldsymbol{x} - [\boldsymbol{x}]_n\|_1 \leq \frac{rd}{\tau_n}$, where $[\boldsymbol{x}]_n$ is the closest point in $\mathcal{D}_{M(n)}$ to $\boldsymbol{x}$.

Using the above aligns, if we set $L' = b\sqrt{\log \frac{da}{\delta}}$, we obtain

$$\|\mathrm{vec}(\mathbf{f}(\boldsymbol{x})) - \mathrm{vec}(\mathbf{f}([\boldsymbol{x}]_n))\| \leq \left(b\sqrt{\log \frac{da}{\delta}} + C^\nabla\right)\|\boldsymbol{x} - [\boldsymbol{x}]_n\|_1 \leq \left(b\sqrt{\log \frac{da}{\delta}} + C^\nabla\right)\frac{rd}{\tau_n},$$

with probability at least $1 - \delta$ for all $\boldsymbol{x} \in \mathcal{X}$.

Choosing $\tau_n = rdn^2(b\sqrt{\log \frac{da}{\delta}} + C^\nabla)$, we have

$$\Pr\left(\|\mathrm{vec}(\mathbf{f}(\boldsymbol{x})) - \mathrm{vec}(\mathbf{f}([\boldsymbol{x}]_M))\| \leq \frac{1}{n^2}\right) \geq 1 - \delta, \quad \forall \boldsymbol{x} \in \mathcal{X}.$$

Combining the above results, we obtain

$$\Pr\left(\|\mathrm{vec}(\mathbf{f}(\boldsymbol{x}^\star)) - \hat{\boldsymbol{\mu}}([\boldsymbol{x}^\star]_n)\| \leq \beta_n \sqrt{\lambda_{\max}^{(n)}([\boldsymbol{x}^\star]_n)} + \frac{1}{n^2}\right) \geq 1 - \delta,$$

where $[\boldsymbol{x}^\star]_n$ is the closest point in $\mathcal{D}_{M(n)}$ to $\boldsymbol{x}^\star$, and $\beta_n = \sqrt{C_n} + 2d \log\left(\frac{rdn^2(b\sqrt{\log(da/\delta)} + C^\nabla)}{\delta}\right)$.

**Part 2. Regret bound.** According to the Lipschitz property of $h$, we have
$$r_n = h(\boldsymbol{x}^\star) - h(\boldsymbol{x}_n) \leq L\|\mathrm{vec}(\mathbf{f}(\boldsymbol{x}^\star)) - \mathrm{vec}(\mathbf{f}(\boldsymbol{x}_n))\|, \tag{42}$$
where $L > 0$ is the Lipschitz constant.

Since

$$\hat{\boldsymbol{\mu}}_{n-1}(\boldsymbol{x}_n) + \beta_n \sqrt{\lambda_{\max}^{(n-1)}(\boldsymbol{x}_n)} \geq \hat{\boldsymbol{\mu}}_{n-1}([\boldsymbol{x}^\star]_n) + \beta_n \sqrt{\lambda_{\max}^{(n-1)}([\boldsymbol{x}^\star]_n)} \geq \mathrm{vec}(\mathbf{f}(\boldsymbol{x}^\star)) - \frac{1}{n^2},$$

we have

$$r_n \leq L\left(2\beta_n \sqrt{\lambda_{\max}^{(n-1)}(\boldsymbol{x}_n)} + \frac{1}{n^2}\right). \tag{43}$$

We first consider the first term:
$$4\beta_n^2 \lambda_{\max}^{(n-1)}(\boldsymbol{x}_n) \leq 4\beta_N^2 \eta\left(\eta^{-1} \lambda_{\max}^{(n-1)}(\boldsymbol{x}_n)\right) \leq 4\beta_N^2 \eta C_2 \log\left(1 + \eta^{-1} \lambda_{\max}^{(n-1)}(\boldsymbol{x}_n)\right)$$
$$\leq 4\beta_N^2 \eta C_2 \log\left|\mathbf{I}_T + \eta^{-1} \hat{\mathbf{K}}_{n-1}(\boldsymbol{x}_n, \boldsymbol{x}_n)\right|,$$

where $C_2$ is a constant.

Define $C_1 = 4\eta C_2$. Then

$$\sum_{n=1}^N 4\beta_n^2 \lambda_{\max}^{(n-1)}(\boldsymbol{x}_n) \leq C_1 N \beta_N^2 \sum_{n=1}^N \log\left|\mathbf{I}_T + \eta^{-1} \hat{\mathbf{K}}_{n-1}(\boldsymbol{x}_n, \boldsymbol{x}_n)\right| \leq C_1 N \beta_N^2 \gamma_N(\mathbf{K}, \eta),$$

where the last inequality holds by the definition of $\gamma_N(\mathbf{K}, \eta)$.

Since $\sum_{n=1}^N \frac{1}{n^2} \leq \frac{\pi^2}{6}$, we have the final result:

$$\sum_{n=1}^N r_n \leq L\left(\sqrt{C_1 N \gamma_N(\mathbf{K}, \eta)}\, \beta_N + \frac{\pi^2}{6}\right). \tag{44}$$

$$\square$$

## H    THE PROOF OF PROPOSITION 2

*Proof.* We start from the definition of the (maximum) information gain:

$$\gamma_n(\mathbf{K}, \eta) = \max_{\mathbf{X}_n} \frac{1}{2} \log \det\big(\mathbf{I}_{nT} + \eta^{-1}\sigma^2\,\mathbf{K}_n\big), \quad \mathbf{K}_n := \big[\mathbf{K}(\boldsymbol{x}_i, \boldsymbol{x}_j)\big]_{i,j=1}^n \in \mathbb{R}^{nT \times nT}. \quad (45)$$

Under the separable kernel in Definition 2, we have

$$\mathbf{K}(\boldsymbol{x}, \boldsymbol{x}') = \big(\text{vec}(\mathbf{A})\text{vec}(\mathbf{A})^\top\big)\,k(\boldsymbol{x}, \boldsymbol{x}') =: \mathbf{B}\,k(\boldsymbol{x}, \boldsymbol{x}'), \quad (46)$$

where $\mathbf{B} := \text{vec}(\mathbf{A})\text{vec}(\mathbf{A})^\top \in \mathbb{R}^{T \times T}$. Then, the Gram matrix factorizes as a Kronecker product

$$\mathbf{K}_n = \mathbf{K}_x \otimes \mathbf{B}, \qquad \mathbf{K}_x = [k(\boldsymbol{x}_i, \boldsymbol{x}_j)]_{i,j=1}^n \in \mathbb{R}^{n \times n}. \quad (47)$$

Let $\{\alpha_i\}_{i=1}^n$ be the eigenvalues of $\mathbf{K}_x$ and $\{\beta_j\}_{j=1}^T$ be the eigenvalues of $\mathbf{B}$. By the spectral property of the Kronecker product, the eigenvalues of $\mathbf{K}_x \otimes \mathbf{B}$ are $\{\alpha_i\beta_j : i = 1, \ldots, n, \ j = 1, \ldots, T\}$. Therefore, we have

$$\log \det\big(\mathbf{I}_{nT} + \eta^{-1}\sigma^2(\mathbf{K}_x \otimes \mathbf{B})\big) = \sum_{i=1}^n \sum_{j=1}^T \log\big(1 + c\,\alpha_i\,\beta_j\big), \qquad c := \eta^{-1}\sigma^2. \quad (48)$$

For each fixed $i$, the function $u \mapsto \log(1 + c\,\alpha_i\,u)$ is non-decreasing for $u \geq 0$, hence

$$\sum_{j=1}^T \log\big(1 + c\,\alpha_i\,\beta_j\big) = \sum_{j:\,\alpha_j > 0} \log\big(1 + c\,\alpha_i\,\beta_j\big) \leq T \cdot \log\big(1 + c\,\alpha_i\,\beta_{\max}\big), \quad (49)$$

where $\beta_{\max} := \max_j \beta_j$. Summing over $i = 1, \ldots, n$ and multiplying by $1/2$ gives

$$\gamma_n(\mathbf{K}, \eta) \leq \frac{T}{2} \sum_{i=1}^n \log\big(1 + c\,\beta_{\max}\,\lambda_i\big) = T \cdot \gamma_n\big(k^\sharp, \eta\big), \quad (50)$$

where $k^\sharp := \beta_{\max}\,k$ is simply a rescaled version of $k$. Since rescaling by a positive constant does not change the asymptotic order of the information gain, we obtain the general comparison bound

$$\gamma_n(\mathbf{K}, \eta) \leq T \cdot \gamma_n(k, \eta).$$

In our specific setting, the tightest bound is $\gamma_n(\mathbf{K}, \eta) = \mathcal{O}(T\gamma_n(k, \eta))$. Finally, we substitute known results for the scalar kernel $k$: (i) If $k$ is the Gaussian (squared exponential) kernel in $d$ dimensions, then $\gamma_n(k, \eta) = \mathcal{O}\big((\log n)^{d+1}\big)$, which gives $\gamma_n(\mathbf{K}, \eta) = \mathcal{O}\big(T(\log n)^{d+1}\big)$.

(ii) If $k$ is a Matérn kernel with smoothness parameter $\nu > 1$, then $\gamma_n(k, \eta) = \mathcal{O}\big(n^{\frac{d(d+1)}{2\nu+d(d+1)}} \log n\big)$, which gives $\gamma_n(\mathbf{K}, \eta) = \mathcal{O}\big(T\,n^{\frac{d(d+1)}{2\nu+d(d+1)}} \log n\big)$. $\qquad \square$

## I    THE PROOF OF THEOREM 2

Denote $\tilde{h}(\boldsymbol{x}, \boldsymbol{\lambda}) = H_f \tilde{\boldsymbol{f}}(\boldsymbol{x}, \boldsymbol{\lambda})$, then we have

$$\tilde{h}(\boldsymbol{x}^\star, \boldsymbol{\lambda}^\star) - \tilde{h}(\boldsymbol{x}_n, \boldsymbol{\lambda}_n) = \Big[\tilde{h}(\boldsymbol{x}^\star, \boldsymbol{\lambda}^\star) - \tilde{h}(\boldsymbol{x}_n, \boldsymbol{\lambda}^\star)\Big] + \Big[\tilde{h}(\boldsymbol{x}_n, \boldsymbol{\lambda}^\star) - \tilde{h}(\boldsymbol{x}_n, \boldsymbol{\lambda}_n)\Big]$$
$$= r_{1n} + r_{2n}.$$

For the first item, according to the Cauchy-Schwarz inequality, we have

$$H_f \tilde{\boldsymbol{f}}(\boldsymbol{x}, \boldsymbol{\lambda}) - H_f \tilde{\boldsymbol{\mu}}_n(\boldsymbol{x}, \boldsymbol{\lambda}) \leq H \|\tilde{\boldsymbol{f}}(\boldsymbol{x}, \boldsymbol{\lambda}) - \tilde{\boldsymbol{\mu}}_n(\boldsymbol{x}, \boldsymbol{\lambda})\|. \quad (51)$$

Similar to the proof of Theorem 1, we provide the following lemma.

**Lemma 2.** *Under Assumption 1–3, suppose the noise vectors* $\{vec(\boldsymbol{\varepsilon}_i)\}_{i \geq 1}$ *are independently and identically distributed in* $N(0, \sigma^2 \mathbf{I}_k)$. *Then, for any* $\delta \in (0, 1]$, *with probability at least* $1 - \delta$, $\|\tilde{\boldsymbol{f}}(\boldsymbol{x}, \boldsymbol{\lambda}) - \tilde{\boldsymbol{\mu}}_n(\boldsymbol{x}, \boldsymbol{\lambda})\|_2 \leq \tilde{\beta}_n \|\tilde{\mathbf{K}}_n(\boldsymbol{x}, \boldsymbol{x}; \boldsymbol{\lambda}, \boldsymbol{\lambda})\|^{1/2}$ *holds uniformly over all* $\boldsymbol{x} \in \mathcal{X}$ *and* $\boldsymbol{\lambda} \in \Lambda$ *and* $i \geq 1$, *where* $\beta_n = \sqrt{\tilde{C}_n} + 2d \log(\frac{rdn^2(b\sqrt{\log(da/\delta)}+\tilde{C}_\nabla)}{l}\delta)$, $\tilde{C}_n = \sup_{\boldsymbol{x} \in \mathcal{X}} \frac{tr(\tilde{\mathbf{K}}_n(\boldsymbol{x}, \boldsymbol{x}; \boldsymbol{\lambda}_n^\star, \boldsymbol{\lambda}_n^\star)))}{\lambda_{max}^{(n)}(\boldsymbol{x}, \boldsymbol{\lambda}_n^\star)}$, *and* $\tilde{C}_\nabla = \sup_{\boldsymbol{x} \in \mathcal{X}} \sqrt{tr(\tilde{\mathbf{K}}_n^\nabla(x_j, x_j))}$.

From Lemma 2 and (51), we have

$$H_f \tilde{\boldsymbol{f}}(\boldsymbol{x}, \boldsymbol{\lambda}) - H_f \tilde{\boldsymbol{\mu}}_n(\boldsymbol{x}, \boldsymbol{\lambda}) \leq H \|\tilde{\boldsymbol{f}}(\boldsymbol{x}, \boldsymbol{\lambda}) - \tilde{\boldsymbol{\mu}}_n(\boldsymbol{x}, \boldsymbol{\lambda})\| \leq H \tilde{\beta}_n \|\tilde{\mathbf{K}}_n(\boldsymbol{x}, \boldsymbol{x}; \boldsymbol{\lambda}, \boldsymbol{\lambda})\|^{1/2}.$$

Then, we have

$$
\begin{aligned}
r_{1n} &= \tilde{h}(\boldsymbol{x}^\star, \boldsymbol{\lambda}^\star) - \tilde{h}(\boldsymbol{x}_n, \boldsymbol{\lambda}^\star) \\
&\leq H_f(\tilde{\boldsymbol{\mu}}_n(\boldsymbol{x}^\star, \boldsymbol{\lambda}^\star)) + H \tilde{\beta}_n \|\tilde{\mathbf{K}}_{n-1}(\boldsymbol{x}^\star, \boldsymbol{x}^\star)\|^{1/2} - H_f(\tilde{\boldsymbol{f}}(\boldsymbol{x}_n, \boldsymbol{\lambda}^\star)) \\
&\leq 2H \tilde{\beta}_{n-1} \|\tilde{\mathbf{K}}_{n-1}(\boldsymbol{x}_n, \boldsymbol{x}_n; \boldsymbol{\lambda}^\star, \boldsymbol{\lambda}^\star)\|^{1/2}.
\end{aligned}
$$

Similar to Theorem 1, we obtain

$$
\begin{aligned}
R_{1N} := \sum_{n=1}^N r_{1n} &\leqslant 2H \tilde{\beta}_{n-1} \|\tilde{\mathbf{K}}_{n-1}(\boldsymbol{x}_n, \boldsymbol{x}_n; \boldsymbol{\lambda}^\star, \boldsymbol{\lambda}^\star)\|^{1/2} \\
&\leq H \left( \sqrt{C_3 N \gamma_N(\tilde{\mathbf{K}}, \eta)} \tilde{\beta}_N + \frac{\pi^2}{6} \right),
\end{aligned}
$$

where $C_3 > 0$ is a constant.

For the second item, followed in Accabi et al. (2018), we have

$$R_{2N} \leq H \sqrt{C_4 T N \tilde{\gamma}_n(\tilde{\mathbf{K}}, \eta)} \tilde{\rho}_N,$$

where $C_4 > 0$ is a constant.

Then we have the cumulative regret over $N$ rounds is bounded by

$$
\begin{aligned}
r_n &\leq r_{1n} + r_{2n} \\
&\leq H \left( \sqrt{C_3 \gamma_N(\tilde{\mathbf{K}}, \eta) N} \tilde{\beta}_N + \frac{\pi^2}{6} + \sqrt{C_4 T N \tilde{\gamma}_N(\tilde{\mathbf{K}}, \eta)} \tilde{\rho}_N \right).
\end{aligned}
$$

## J  THE PROOF OF PROPOSITION 3

*Proof.* Based on the definition of the (maximum) information gain, we have

$$\gamma_n(\tilde{\mathbf{K}}, \eta) = \max_{\mathbf{X}_n} \frac{1}{2} \log \det(\mathbf{I}_m + c\, \tilde{\mathbf{K}}_n), \qquad c := \eta^{-1} \sigma^2, \tag{52}$$

where the Gram matrix takes the form $\tilde{\mathbf{K}}_n = \mathbf{E}\mathbf{K}_n\mathbf{E}^\top$ and $\mathbf{K}_n = \mathbf{B} \otimes \mathbf{K}_x$, with $\mathbf{K}_x = [k(\boldsymbol{x}_i, \boldsymbol{x}_j)]_{i,j=1}^n \in \mathbb{R}^{n \times n}$, $\mathbf{B} = vec(\mathbf{A})vec(\mathbf{A})^\top \in \mathbb{R}^{T \times T}$.

By Sylvester's identity, we have

$$\det\left(\mathbf{I}_{nk} + c\, \mathbf{E}_n\mathbf{K}_n\mathbf{E}_n^\top\right) = \det\left(\mathbf{I}_{nT} + c\, \mathbf{K}_n^{1/2}\mathbf{E}_n^\top\mathbf{E}_n\mathbf{K}_n^{1/2}\right). \tag{53}$$

Since $\mathbf{E}_n$ selects rows, $0 \preceq E_n^\top E_n \preceq \mathbf{I}_{nT}$, hence

$$
\begin{aligned}
\log \det\left(\mathbf{I}_{nk} + c\, \mathbf{E}_n\mathbf{K}_n\mathbf{E}_n^\top\right) &= \log \det\left(\mathbf{I}_{nT} + c\, \mathbf{K}_n^{1/2}\mathbf{E}_n^\top\mathbf{E}_n\mathbf{K}_n^{1/2}\right) \\
&\leq \log \det\left(\mathbf{I}_{nT} + c\, \mathbf{K}_n\right).
\end{aligned} \tag{54}
$$

Then, followed in Proposition 1, we have

$$\gamma_n(\tilde{\mathbf{K}}_n, \eta) = \mathcal{O}\left(T(\log n)^{d+1}\right) \text{ for the Gaussian kernel,} \tag{55}$$

$$\gamma_n(\tilde{\mathbf{K}}_n, \eta) = \mathcal{O}\left(Tn^{\frac{d(d+1)}{2\nu+d(d+1)}} \log n\right) \text{ for the Matérn kernel.} \tag{56}$$

From (54), we also obtain that

$$\tilde{\gamma}_n(\tilde{\mathbf{K}}_n, \eta) = \mathcal{O}\big(T(\log n)^{d+1}\big) \text{ for the Gaussian kernel,} \tag{57}$$

$$\tilde{\gamma}_n(\tilde{\mathbf{K}}_n, \eta) = \mathcal{O}\Big(T n^{\frac{d(d+1)}{2\nu+d(d+1)}} \log n\Big) \text{ for the Matérn kernel.} \tag{58}$$

$\square$

## K   DETAILED SETTINGS OF THE EXPERIMENTS

### K.1   DETAILED SETTINGS OF SYNTHETIC EXPERIMENTS

The setting of simulations are as follows:

- **Compute resources:** All experiments were run on a Windows 10 Pro (Build 19045) desktop with an Intel Core i9-7900X CPU (3.10 GHz) and 32 GB RAM.

- **Kernel setting:** We use the Matérn kernel function with the smoothing parameter $\nu = 5/2$ as the input kernel for different GPs.

- **Data setting:** For the GP prediction, we generate $n_{train} = 10d$ training samples for estimating hyperparameters and $n_{test} = 5d$ testing samples for predicting. In the BO and CBBO framework, we generate $n_0 = 5d$ initial design points based on Latin Hypercube Sampling (LHS), and $N = 10d$ sequential design points.

- **Criteria:** To compare the GPs prediction performance of different methods, we use the following criteria:

  (1) Negative Log-Likelihood (NLL): It measures how well a probabilistic model fits the observed data. For given data $\mathbf{X}_n$ and $\mathbf{Y}_n$, the NLL is defined as $\text{NLL} = \frac{1}{2}\log(\tau^2\mathbf{K}_n + \sigma^2\mathbf{I}_{nT}) + \frac{1}{2}\mathbf{Y}_n^\top(\tau^2\mathbf{K}_n + \sigma^2\mathbf{I}_{nT})^{-1}\mathbf{Y}_n$.

  (2) Mean Absolute Error (MAE): $\text{MAE} = \frac{1}{n_{test}}\sum_{i=1}^{n_{test}} \|\frac{\mathbf{y}_i - \hat{\mathbf{y}}_i}{\mathbf{y}_i}\|$.

  (3) Covariance Operator Norm ($\|Cov\|$): $\|\hat{\mathbf{K}}_n\| = \sup_{\|\boldsymbol{v}\|=1} \|\hat{\mathbf{K}}_n\|$.

  For the BO framework, we use the the mean squared error of inputs ($\text{MSE}_x$) and the mean absolute error of outputs $\text{MAE}_y$ to compare the optimization performance of different methods. Here $\text{MSE}_x = \|\boldsymbol{x}^\star - \boldsymbol{x}_N^\star\|^2$, and $\text{MAE}_y = \|\frac{\mathbf{f}^\star - \mathbf{f}_N^\star}{\mathbf{f}^\star}\|$ over $N$ rounds. For the CBBO framework, we add the Acc criterion to compare the match between the optimal super arm and the super arm chosen over N rounds, that is, $Acc = \mathbb{I}_{\boldsymbol{\lambda}^\star = \boldsymbol{\lambda}_N}/k$, where $\mathbb{I}$ is a indictor function.

### K.2   DETAILED SETTINGS OF CASE STUDIES

The detailed datasets of case studies are as follows:

**Chemistry reaction (CHEM):** Reaction optimization is fundamental to synthetic chemistry, from improving yields in industrial processes to selecting reaction conditions for drug candidate synthesis. According to Shields et al. (2021), we aim to evaluate the reaction parameters ($x_1$: concentration, $x_2$: temperature) to improve the experimental yields ($\mathbf{y} : 4 \times 3 \times 3$) of palladium-catalysed direct arylation reaction under varying bases (Mode 1), ligands (Mode 2), and solvents (Mode 3).

**PS/PAN material (MAT):** Electrospun polystyrene/polyacrylonitrile (PS/PAN) materials are commonly used as potential oil sorbents for marine oil spill remediation. From Wang et al. (2020), we aim to optimize the fabrication parameters of PS/PAN materials, including spinneret speed ($x_1$), collector distance ($x_2$), applied voltage ($x_3$), and fiber diameter ($x_4$), to improve their oil absorption capacity ($\mathbf{y} : 5 \times 4 \times 4$) under varying PS content (Mode 1), mass fraction (Mode 2), and $SiO_2$ content (Mode 3).

**3D printing (PRINT):** Material extrusion-based three-dimensional printed products have been widely used in aerospace, automotive, and other fields. Following Zhai et al. (2023), we focus on selecting appropriate process parameters ($x_1$: layer thickness, $x_2$: platform temperature, $x_3$: nozzle temperature, $x_4$: infill density, and $x_5$: printing speed) to reduce variations in part quality

($\mathbf{y} : 3 \times 4 \times 3$) caused by different printer nozzles (Mode 1) and printing geometries (Mode 2). The quality (Mode 3) is evaluated in terms of compression deformation, compressive strength, and the printing cost.

**Renewable energy (REEN):** Climate change affects the availability and reliability of renewable energy sources such as wind, solar, and hydropower. We employ the operational energy dataset from the Copernicus Climate Change Service (`https://cds.climate.copernicus.eu/datasets/sis-energy-derived-reanalysis?tab=overview`) to explore the climate conditions that are most beneficial to renewable energy generation in various European nations. The climate-related variables, used as input features, include air temperature ($x_1$), precipitation ($x_2$), surface incoming solar radiation ($x_3$), wind speed at 10 meters ($x_4$) and 100 meters ($x_5$), and mean sea level pressure ($x_6$). The energy-related indicators ($\mathbf{y} : 10 \times 2$) collected from ten European countries (Mode 1), used as outputs, include the capacity factor ratio of solar photovoltaic power generation and wind power generation onshore (Mode 2).

## L    THE COMPARISON OF COMPUTATIONAL COMPLEXITY FOR BASELINES

To provide a clearer comparison, we summarize the computational complexity of GP training, BO, and CBBO for the baseline methods sMTGP, MLGP, and MVGP, together with our proposed method, in the table below. Here $m_h$, $m_{h1}$, $m_{h2}$, and $m_{hl}$ for $l = 1, \cdots, m$ denote the numbers of

Table 5: The computational complexity of different methods for GP training, BO, and CBBO.

| Task | Method | Computational complexity |
|---|---|---|
| GP Training | TOGP | $\mathcal{O}\big((n^3 T^3 + n^2 T^2 m_h) \log n\big)$ |
| | sMTGP | $\mathcal{O}\big((n^3 + \sum_{l=1}^m t_l^3 + n^2 m_h + \sum_{l=1}^m t_l^2 m_{hl}) \log n\big)$ |
| | MLGP | $\mathcal{O}\big((n^3 + \sum_{l=1}^m t_l^3 + n^2 m_h + \sum_{l=1}^m t_l^2 m_{hl}) \log n\big)$ |
| | MVGP | $\mathcal{O}\big((n^3 + T^3 + n^2 m_{h1} + T^2 m_{h2}) \log n\big)$ |
| BO (Round $n$) | TOGP | $\mathcal{O}\big((n^3 T^3 + n^2 T^2 m_h) \log n\big)$ |
| | sMTGP | $\mathcal{O}\big((n^3 + \sum_{l=1}^m t_l^3 + n^2 m_h + \sum_{l=1}^m t_l^2 m_{hl}) \log n\big)$ |
| | MLGP | $\mathcal{O}\big((n^3 + \sum_{l=1}^m t_l^3 + n^2 m_h + \sum_{l=1}^m t_l^2 m_{hl}) \log n\big)$ |
| | MVGP | $\mathcal{O}\big((n^3 + T^3 + n^2 m_{h1} + T^2 m_{h2}) \log n\big)$ |
| CBBO (Round $n$) | TOGP | $\mathcal{O}\big((n^3 k^3 + n^2 k T m_h) \log n + k T^3\big)$ |
| | sMTGP | $\mathcal{O}\big((n^3 + k^3 + n^2 m_h + k \sum_{l=1}^m t_l m_{hl}) \log n + k \sum_{l=1}^m t_l^3\big)$ |
| | MLGP | $\mathcal{O}\big((n^3 + k^3 + n^2 m_h + k \sum_{l=1}^m t_l m_{hl}) \log n + k \sum_{l=1}^m t_l^3\big)$ |
| | MVGP | $\mathcal{O}\big((n^3 + T^3 + n^2 m_{h1} + k T m_{h2}) \log n + k T^3\big)$ |

hyperparameters for the corresponding methods.

Furthermore, we report the empirical computational time for GP training across all methods under the three experimental settings. The results are summarized below.

Table 6: The runtime (s) of different methods for GP training in the three synthetic settings.

| | TOGP | sMTGP | MLGP | MVGP |
|---|---|---|---|---|
| **Setting (1)** | 266.04 | 27.26 | 30.57 | **6.50** |
| **Setting (2)** | 69.85 | 16.08 | 21.99 | **1.31** |
| **Setting (3)** | 900.13 | 851.34 | 891.73 | **769.50** |

We further report the empirical running time of BO and CBBO under all baseline methods. The results are summarized in the table below.

Finally, we provide the prediction and optimization performance of TOGP based on the Nyström low-rank approximation (SVD-TOGP) in Remark 3 under the three synthetic settings. The results are summarized below.

Table 7: The runtime (s) of different methods for BO and CBBO in the three synthetic settings.

| Method/Task | Setting (1) BO | Setting (1) CBBO | Setting (2) BO | Setting (2) CBBO | Setting (3) BO | Setting (3) CBBO |
|---|---|---|---|---|---|---|
| TOBO/TOCBBO | 6968.25 | 8017.08 | 1588.53 | 2348.16 | 6035.61 | 11699.09 |
| sMTGP-UCB | 1516.31 | 953.82 | 117.78 | 179.11 | 5291.26 | 9436.90 |
| MLGP-UCB | 1545.41 | 981.86 | 133.93 | 185.49 | 5452.90 | 9696.60 |
| MVGP-UCB | 340.44 | 437.29 | 14.86 | 20.76 | 3006.97 | 3199.47 |
| TOGP-RS | 6176.84 | 7990.44 | 1573.36 | 2338.97 | 5786.02 | 11501.01 |
| sMTGP-RS | 1503.24 | 939.27 | 106.93 | 164.95 | 5273.84 | 9421.08 |
| MLGP-RS | 1530.32 | 971.25 | 109.14 | 180.70 | 5252.32 | 9431.96 |
| MVGP-RS | **334.79** | **386.86** | **7.70** | **6.70** | **2997.75** | **3116.23** |

Table 8: Summary of TOGP and SVD-TOGP performance for GP training, BO and CBBO in the three synthetic settings.

| Task | Criterion | Setting (1) SVD-TOGP | Setting (1) TOGP | Setting (2) SVD-TOGP | Setting (2) TOGP | Setting (3) SVD-TOGP | Setting (3) TOGP |
|---|---|---|---|---|---|---|---|
| GP Training | NLL | 557.09 | **503.00** | **-18.39** | -18.10 | -3778.74 | **-3923.10** |
| | MAE | 0.1756 | **0.1571** | 0.1099 | **0.1052** | 0.1436 | **0.1372** |
| | $\|Cov\|$ | 3.9627 | **2.0200** | 0.6261 | **0.0400** | **0.0881** | 2.8200 |
| | Time (s) | **21.80** | 266.04 | **10.83** | 69.85 | **436.17** | 900.13 |
| BO | $MSE_x$ | 0.0001 | **0.0000** | 0.0003 | **0.0003** | 0.0002 | **0.0001** |
| | $MAE_y$ | 0.0041 | **0.0008** | 0.0356 | **0.0350** | 0.0051 | **0.0050** |
| | Ins Regret | 0.0040 | **0.0001** | 0.0002 | **0.0002** | 0.0402 | **0.0302** |
| | Time (s) | **831.02** | 6968.25 | **162.78** | 1588.53 | **3604.66** | 6035.61 |
| CBBO | $MSE_x$ | 0.0024 | **0.0023** | **0.0000** | **0.0000** | 0.0035 | **0.0021** |
| | $MAE_y$ | 0.0180 | **0.0172** | **0.0000** | **0.0000** | 0.0246 | **0.0145** |
| | Acc | **1** | **1** | **1** | **1** | **1** | **1** |
| | Ins Regret | 1.0944 | **0.9807** | **0.0000** | **0.0000** | 1.8571 | **1.4406** |
| | Time (s) | **655.83** | 8017.08 | **134.22** | 2348.16 | **4446.12** | 11699.09 |

# M    ADDITIONAL RESULTS OF SYNTHETIC EXPERIMENTS

Table 9: The additional optimization performance of different methods in three synthetic settings.

| $k$ | | Setting (1) $MSE_x$ | $MAE_y$ | Acc | Setting (2) $MSE_x$ | $MAE_y$ | Acc | Setting (3) $MSE_x$ | $MAE_y$ | Acc |
|---|---|---|---|---|---|---|---|---|---|---|
| $T/3$ | TOCBBO | **0.0054** | **0.0509** | **1.00** | **0.0001** | **0.0036** | **1.00** | **0.0021** | **0.0312** | **1.00** |
| | SMTGP-UCB | 0.0057 | 0.0942 | **1.00** | 0.0548 | 0.3336 | **1.00** | 0.0340 | 0.8616 | 0.71 |
| | MLGP-UCB | 0.0677 | 0.9775 | 0.33 | 0.0434 | 0.6961 | 0.50 | 0.3549 | 0.8205 | 0.29 |
| | MVGP-UCB | 0.0325 | 0.7530 | 0.33 | 0.0019 | 0.0191 | **1.00** | 0.0405 | 0.3647 | 0.71 |
| | TOGP-RS | 0.0208 | 1.3279 | 0.83 | 0.0206 | 0.4347 | 0.50 | 0.0709 | 1.5521 | 0.57 |
| | SMTGP-RS | 0.0203 | 1.2625 | 0.50 | 0.1201 | 0.4443 | 0.50 | 0.0254 | 1.0703 | 0.50 |
| | MLGP-RS | 0.0619 | 1.0296 | 0.67 | 0.1191 | 0.5081 | **1.00** | 0.1816 | 1.1735 | 0.64 |
| | MVGP-RS | 0.0634 | 0.9733 | 0.67 | 0.0469 | 0.4444 | 0.50 | 0.0574 | 1.3837 | 0.43 |
| $2T/3$ | TOCBBO | **0.0006** | **0.0125** | **1.00** | **0.0010** | **0.0076** | **1.00** | **0.0103** | **0.1163** | **0.96** |
| | SMTGP-UCB | 0.0066 | 0.0585 | **1.00** | 0.0012 | 0.0177 | **1.00** | 0.0122 | 1.9890 | 0.89 |
| | MLGP-UCB | 0.0323 | 3.6747 | 0.82 | 0.2229 | 1.4532 | 0.75 | 0.0423 | 4.0294 | 0.81 |
| | MVGP-UCB | 0.0047 | 0.4765 | **1.00** | 0.0022 | 0.0309 | **1.00** | 0.0324 | 1.4985 | 0.93 |
| | TOGP-RS | 0.0128 | 4.6512 | 0.82 | 0.0202 | 1.1872 | **1.00** | 0.0218 | 10.4919 | 0.70 |
| | SMTGP-RS | 0.0128 | 4.6512 | 0.82 | 0.2229 | 1.4532 | 0.75 | 0.0110 | 6.0795 | 0.70 |
| | MLGP-RS | 0.0323 | 3.6747 | 0.82 | 0.2229 | 1.4532 | 0.75 | 0.0423 | 4.0294 | 0.81 |
| | MVGP-RS | 0.0128 | 4.6512 | 0.82 | 0.0439 | 0.8309 | **1.00** | 0.0423 | 4.0294 | 0.81 |

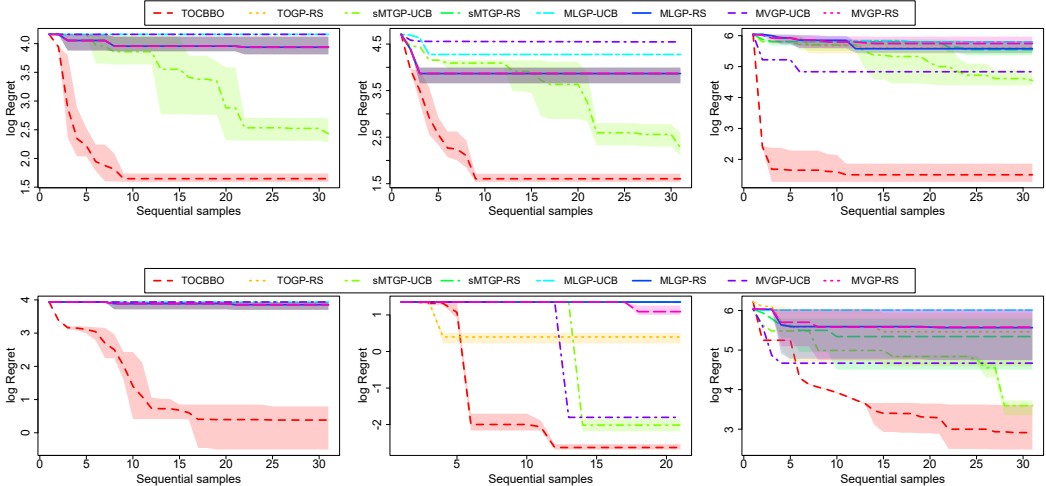

Figure 4: Each round's logarithmic instantaneous regret of different methods in the Setting (1) (Left), (2) (Middle), and (3) (Right) when $k = T/3$ (Top row) and $k = 2T/3$ (Bottom row).

Table 10: Summary across CHEM, MAT, and PRINT datasets with 10 repetitions.

| Method | CHEM | MAT | PRINT |
|---|---|---|---|
| TOBO | 100 (0.00) | 150.07 (0.00) | -25.03 (0.00) |
| sMTGP-UCB | 99.91 (0.10) | 147.97 (2.24) | -25.03 (0.00) |
| MLGP-UCB | 99.29 (0.75) | 147.90 (2.32) | -25.13 (0.11) |
| MVGP-UCB | 96.32 (4.89) | 147.20 (3.44) | -25.27 (0.32) |
| TOGP-RS | 93.84 (1.56) | 147.83 (2.32) | -25.27 (0.23) |
| sMTGP-RS | 96.63 (3.74) | 146.22 (4.30) | -25.03 (0.13) |
| MLGP-RS | 99.41 (0.76) | 148.28 (1.64) | -25.27 (0.23) |
| MVGP-RS | 96.32 (3.72) | 147.20 (2.80) | -25.27 (0.33) |

In the BO setting, we additionally include a single-objective GP-UCB baseline, which directly assumes that $L_f \mathbf{f}$ follows a GP and uses the UCB acquisition function to select query points sequentially. The optimization results under the three settings in our numerical experiments are summarized below.

Table 11: The results of Single GP-UCB in three synthetic settings.

| | Criterion | Single GP-UCB | TOBO |
|---|---|---|---|
| | $\text{MSE}_x$ | 0.0004 | **0.0000** |
| Setting (1) | Ins Regret | 0.0459 | **0.0001** |
| | Time | **457.34** | 6968.25 |
| | $\text{MSE}_x$ | 0.4761 | **0.0003** |
| Setting (2) | Ins Regret | 0.2093 | **0.0002** |
| | Time | **187.30** | 1588.53 |
| | $\text{MSE}_x$ | 0.0084 | **0.0001** |
| Setting (3) | Ins Regret | 0.1964 | **0.0302** |
| | Time | **415.61** | 6035.61 |

We include an independent GP-UCB baseline (Ind GP-UCB), which assumes that each tensor element follows its own GP model and applies the proposed CMAB-UCB2 acquisition function to sequentially select both query points and super-arms. The optimization results under the three synthetic settings are summarized below.

Table 12: The results of the Ind GP-UCB in three synthetic settings.

|  | Criterion | Ind GP-UCB | TOBO |
|---|---|---|---|
| **Setting (1)** | MSE$_x$ | 0.2121 | **0.0023** |
|  | MAE$_y$ | 0.6739 | **0.0172** |
|  | Acc | 0.67 | **1** |
|  | Ins Regret | 36.301 | **0.9807** |
|  | Time (s) | **328.56** | 8017.08 |
| **Setting (2)** | MSE$_x$ | 0.0731 | **0.0000** |
|  | MAE$_y$ | 0.0486 | **0.0000** |
|  | Acc | **1** | **1** |
|  | Ins Regret | 0.1561 | **0.0000** |
|  | Time (s) | **110.77** | 2348.16 |
| **Setting (3)** | MAE$_y$ | 0.0139 | **0.0021** |
|  | Acc | 0.86 | **1** |
|  | Ins Regret | 54.2714 | **1.4406** |
|  | Time (s) | **3296.72** | 11699.09 |

For non-separable multi-output GPs (Fricker et al., 2013), we additionally provide the results under the three synthetic settings below.

Table 13: The results of the non-separable MOGP in three synthetic settings.

| Task | Criterion | Setting (1) | Setting (2) | Setting (3) |
|---|---|---|---|---|
| **GP** | NLL | 507.19 | -39.33 | -3094.10 |
|  | MAE | 0.1923 | 0.1493 | 0.1591 |
|  | $\|\text{Cov}\|$ | 6.7398 | 0.5618 | 20.5800 |
|  | Time (s) | 453.11 | 151.31 | 1397.68 |
| **BO** | MSE$_x$ | 0.0014 | 0.0005 | 0.0244 |
|  | MAE$_y$ | 0.0536 | 0.0503 | 0.6277 |
|  | Ins Regret | 1.0284 | 0.0030 | 6.6283 |
|  | Time (s) | 9109.08 | 2101.45 | 7881.31 |
| **CBBO** | MSE$_x$ | 0.0073 | 0.0683 | 0.0091 |
|  | MAE$_y$ | 0.0643 | 0.0390 | 0.1257 |
|  | Acc | 1 | 1 | 0.86 |
|  | Ins Regret | 1.7389 | 0.1551 | 51.4646 |
|  | Time (s) | 10462.18 | 3107.82 | 14175.50 |

As shown, TOBO achieves consistently better optimization accuracy than Single GP-UCB, Ind GP-UCB, and non-separable MOGP-UCB across all settings. Although the runtime of TOBO is higher, we further introduce an efficient variant named SVD-TOBO, which employs a low-rank eigen-decomposition to approximate the TOGP covariance in Remark 3 and significantly reduces runtime.

To evaluate the sensitivity of our method with respect to different choices of $L_f$ and $H_f$, we consider three types of operators defined as follows (Chugh, 2020).

- **Sum operator (used in the main manuscript):** $L_f \mathbf{f}(\boldsymbol{x}) = \sum f_{i_1,\ldots,i_m}(\boldsymbol{x})$, $H_f \tilde{\mathbf{f}}(\boldsymbol{x}, \boldsymbol{\lambda}) = \sum \tilde{f}_{i_1,\ldots,i_m}(\boldsymbol{x}, \boldsymbol{\lambda})$.
- **Weighted sum operator:** $L_f \mathbf{f}(\boldsymbol{x}) = \sum w_{i_1,\ldots,i_m} f_{i_1,\ldots,i_m}(\boldsymbol{x})$, $H_f \tilde{\mathbf{f}}(\boldsymbol{x}, \boldsymbol{\lambda}) = \sum w_{i_1,\ldots,i_m} \tilde{f}_{i_1,\ldots,i_m}(\boldsymbol{x}, \boldsymbol{\lambda})$, where $w_{i_1,\ldots,i_m} \sim U(0,1)$.
- **Exponential weighted operator:** $L_f \mathbf{f}(\boldsymbol{x}) = \sum e^{pw_{i_1,\ldots,i_m}-1} e^{pf_{i_1,\ldots,i_m}(\boldsymbol{x})}$, $H_f \tilde{\mathbf{f}}(\boldsymbol{x}, \boldsymbol{\lambda}) = \sum e^{pw_{i_1,\ldots,i_m}-1} e^{p\tilde{f}_{i_1,\ldots,i_m}(\boldsymbol{x},\boldsymbol{\lambda})}$, where $p = 2$ and $w_{i_1,\ldots,i_m} \sim U(0,1)$.

The results under Setting (1) in our numerical experiments are summarized below. It can be seen that our method is robust across different choices of $L_f$ and $H_f$, and both TOBO and TOCBBO consistently achieve strong optimization performance.

Table 14: The results of different operators in three synthetic settings.

|  | Criterion | Sum operator | Weighted sum operator | Exponential weighted operator |
|---|---|---|---|---|
| **BO** | $MSE_x$ | 0.0000 | 0.0000 | 0.0000 |
|  | $MAE_y$ | 0.0008 | 0.0083 | 0.0072 |
|  | Ins Regret | 0.0001 | 0.0044 | 0.0533 |
| **CBBO** | $MSE_x$ | 0.0023 | 0.0037 | 0.0025 |
|  | $MAE_y$ | 0.0172 | 0.0329 | 0.0548 |
|  | Acc | 1 | 1 | 1 |
|  | Ins Regret | 0.1964 | 0.0107 | 0.4972 |

## N   REAL-WORLD APPLICATION: SEMICONDUCTOR MANUFACTURING PROCESS

A motivating example arises from semiconductor manufacturing, where each wafer consists of numerous dies (chips) arranged in a two-dimensional grid. During the Chip Probing (CP) phase, a critical stage for functional quality control, each die is evaluated based on multiple quality variables such as voltage, current, leakage, and power consumption. These variables are spatially correlated across neighboring dies due to physical effects such as process variation and mechanical stress, forming a naturally structured tensor output. To ensure high yield and reliability, manufacturers aim to adjust process control parameters (inputs) so that all quality variables across the wafer remain within target specifications. This leads to an optimization problem where the output is a three-mode tensor: the first two modes index the die positions on the wafer, and the third mode captures multiple quality variables. Such a scenario cannot be effectively modeled by scalar- or vector-output BO approaches, as they would lose essential structural information.

In practice, each wafer may contain hundreds of dies. However, resource and time constraints often make it infeasible to measure all quality variables across all die positions on every wafer. Manufacturers instead selectively measure a subset of output entries, such as centrally located dies, those more prone to failure, or historically most informative regions. Similarly, only a subset of quality variables may be measured if certain tests are time-consuming or costly. This results in a partially observed tensor, where only part of the full output is available at each iteration.

We incorporate this example into the revised paper and conduct a corresponding case study. The input is $\boldsymbol{x} = (x_1, x_2, x_3)$ representing process parameters, and the output $\mathbf{f}(\boldsymbol{x}) \in \mathbb{R}^{5 \times 5 \times 3}$ denotes die-wise quality variables on the wafer. A black-box semiconductor simulator is employed as the true system. We generate 5 observations as the initial design and then sequentially select 20 queried points based on different BO methods. Tables 15 and 16 compare the performance of TOBO and TOCBBO against several baselines. Our methods significantly outperform the alternatives in terms of input accuracy ($MAE_x$), regret, and final objective value, demonstrating their superiority.

|  | TOBO | sMTGP-UCB | MLGP-UCB | MVGP-UCB |
|---|---|---|---|---|
| $MAE_x$ | **0.0651 (0.0016)** | 0.1997 (0.0020) | 0.1579 (0.0059) | 0.2669 (0.0046) |
| Regret | **0.1702 (0.0007)** | 0.2400 (0.0014) | 0.1956 (0.0088) | 0.3818 (0.0029) |
| Objective | **0.8298 (0.0009)** | 0.7600 (0.0031) | 0.8044 (0.0079) | 0.6182 (0.0033) |

Table 15: Performance comparison of TOBO with baseline methods in terms of $MAE_x$, regret, and objective (mean and standard deviation).

|  | TOCBBO | sMTGP-UCB | MLGP-UCB | MVGP-UCB |
|---|---|---|---|---|
| $MAE_x$ | **0.1453 (0.0466)** | 0.4313 (0.0592) | 0.5702 (0.0747) | 0.6758 (0.0844) |
| Regret | **0.1431 (0.0388)** | 0.2483 (0.0522) | 0.4338 (0.0918) | 0.3461 (0.0709) |
| Objective | **0.6085 (0.0436)** | 0.5290 (0.0500) | 0.4282 (0.1147) | 0.4794 (0.0912) |

Table 16: Performance comparison of TOCBBO with baseline methods in terms of $MAE_x$, regret, and objective (mean and standard deviation).

These results show that the proposed methods can effectively optimize under partially observed tensor data, highlighting their practical relevance for semiconductor manufacturing.

