# OpenReview forum: "Combinatorial Bandit Bayesian Optimization for Tensor Outputs"
_ICLR.cc/2026/Conference — ICLR 2026 Poster_

### Official Review · Reviewer_uCf6 · 2025-10-23

**Soundness:** 3
**Presentation:** 3
**Contribution:** 2
**Rating:** 6
**Confidence:** 1

**Summary:**

The paper introduces Tensor-Output Bayesian Optimization, a framework for optimizing black-box functions with tensor-valued outputs using a tensor-output Gaussian process. It proposes both separable and non-separable tensor-output kernels that capture input-dependent cross-mode correlations, and develops a UCB acquisition tailored to scalarized tensor objectives. The work further extends to a combinatorial setting where only a subset of tensor entries contributes to the objective, using a partially observed TOGP and a CMAB-UCB2 criterion for joint input and subset selection. Theoretical sublinear regret bounds are provided, and experiments on synthetic and real tasks show consistent improvements over vectorized multi-output GP baselines.

**Strengths:**

Modeling tensor outputs directly with non-separable, input-dependent kernels is a clear step beyond vectorized MOBO, enabling richer cross-mode structure to be exploited. The CBBO formulation and the PTOGP-based acquisition rule are principled. The regret analyses leverage information gain and cover both full and partial observation regimes, giving the method theoretical footing. Empirical results are broad, with synthetic and multiple real datasets, and show strong gains over standard multi-task and multivariate GP baselines. The appendices give thorough algorithmic and complexity details.

**Weaknesses:**

The computational complexity $O(n^3 T^3 log(n))$ of TOBO is cubic in both n **and** tensor size, which may severely limit practical scale.

The scalarization operators $L_f$ and $H_f$ are central in practice, but their choices, sensitivity, and domain-specific instantiations are only lightly discussed.

Baselines omit state-of-the-art tensor GP surrogates with Kronecker structure and non-separable multi-output kernels beyond the selected MTGP/MVGP/MLGP such as proposed in "Scalable High-Order Gaussian Process Regression" (line 91).

**Questions:**

- How are $L_f$ and $H_f$ instantiated across tasks, and how sensitive are results to these choices or to non-Lipschitz scalarizations encountered in practice?
- How are tensor ranks chosen for the core tensors (line 208), and what is the performance/overfitting trade-off as ranks vary under separable vs non-separable kernels?

---

> ### Author Response · Authors · 2025-11-21
> **Response to Weakness 1**
>
> Thank you very much for your valuable suggestions. We have made point-by-point responses to address all raised questions and concerns. The **rebuttal revision** has also been submitted. All changes in the rebuttal version of the manuscript have been **red** for clarity.
>
> ---
>
> ***Weakness 1: Regarding the high cubic complexity of TOBO in both (n) and tensor size***
>
> **Response to Weakness 1:** Thank you for raising this question. Although the computational complexity of training TOGP is $O(n^3T^3)$, and the computational costs of TOBO and TOCBBO are also dominated by $O(n^3T^3)$, the overall computational burden remains affordable in practice. In Setting (3), where the tensor dimension is $4 \times 5 \times 2$, the total computation time for sequentially adding $30$ new points on top of $20$ initial design points is 6035.61 seconds conducted on a Windows workstation equipped with an Intel Core i9-9960X CPU (16 cores) and 128 GB RAM server. More importantly, TOGP, TOBO, and TOCBBO achieve significantly better fitting and optimization performance than all baselines (see Tables 2 and 3, lines 425–457 on page 8). **The additional computational cost is therefore well justified by the significant improvements in accuracy.** Moreover, BO is typically applied to expensive black-box optimization scenarios. In such settings, the cost of data acquisition often far exceeds the computational cost of training the surrogate model, and the optimization performance is far more critical than the running time.
>
> **For large tensor outputs where the computational cost may become prohibitive, we have also introduced a Nyström low-rank approximation strategy to improve the scalability of TOGP.** Specifically, based on data $\mathbf{X_n}$ and $\boldsymbol Y_n$, consider the spectral decomposition $\mathbf K_n=\mathbf U_n\mathbf\Lambda_n\mathbf U_n^\top$, where $\mathbf\Lambda_n = \mathrm{diag}(\lambda_1,\lambda_2,\ldots,\lambda_{nT})$ denotes eigenvalues ordered as $\lambda_1\ge\lambda_2\ge\ldots\ge\lambda_{nT}>0$. We approximate $\mathbf K_n$ using the leading $l$ eigenpairs ($l\ll nT$), that is, $\mathbf K_n\approx\mathbf U_l\mathbf \Lambda_l\mathbf U_l^\top$, where $\mathbf U_l \in \mathbb R^{nT\times l}$ contains the first $l$ eigenvectors and $\mathbf\Lambda_l=\mathrm{diag}(\lambda_1,\ldots,\lambda_l)$. The rank $l$ is selected by cumulative explained variance:
> $$l=\min_{l_0}\{l_0 \in \{1,\ldots,nT\}:\sum_{i=1}^{l_0} \lambda_i/\sum_{i=1}^{nT}\lambda_i\ge c\}.$$
> The Nyström method approximates $\mathbf{K}_n$ by selecting $n_l$ ($\ll nT$) landmark columns and extending a small eigendecomposition, which yields a computational cost of evaluating $(\mathbf K_n+\eta \mathbf I)^{-1}$ as $\mathcal{O}(nTn_l^2 +n_l^3)$ (Williams & Seeger 2000). Thus, the overall computational complexity for training TOGP becomes $\mathcal O\left((nTn_l^2 + n_l^3 + n^2 T^2 m_h)\log n\right)$. We have added it as a remark in Section 3.1 of our **rebuttal revision** (lines 252-261, page 5).
>
> Furthermore, we provide the prediction and optimization performance of TOGP based on the Nyström low-rank approximation (SVD-TOGP) under the three synthetic settings. The results are summarized below.
>
> |||Setting (1)||Setting (2)||Setting (3)||
> |-|-|-|-|-|-|-|-|
> |**Task**|**Criterion**|**SVD-TOGP**|**TOGP**|**SVD-TOGP**|**TOGP**|**SVD-TOGP**|**TOGP**|
> |**GP**|NLL|557.09|503.00|-18.39|-18.10|-3778.74|-3923.1|
> ||MAE|0.1756|0.1571|0.1099|0.1052|0.1436|0.1372|
> ||$\Vert Cov\Vert$|3.9627| 2.0200|0.6261|0.0400|0.0881|2.8200|
> ||Time (s)|**21.8**|266.04|**10.83**|69.85|**436.17**|900.13|
> |**BO**|MSE$_x$|0.0001|0.0000|0.0003|0.0003|0.0002|0.0001|
> ||MAE$_y$|0.0041|0.0008|0.0356|0.0350|0.0051|0.0050|
> ||Ins Regret|0.0040|0.0001|0.0002|0.0002|0.0402|0.0302|
> ||Time (s)|**831.02**|6968.25|**162.78**|1588.53|**3604.66**|6035.61|
> |**CBBO**|MSE$_x$|0.0024|0.0023|0.0000|0.0000|0.0035|0.0021|
> ||MAE$_y$|0.0180|0.0172|0.0000|0.0000|0.0246|0.0145|
> ||Acc|1|1|1|1|1|1|
> ||Ins Regret|1.0944|0.9807|0.0000|0.0000|1.8571|1.4406|
> ||Time (s)|**655.83**|8017.08|**134.22**|2348.16|**4446.12**|11699.09|
>
> **We can see that SVD-TOGP not only greatly reduces the computational time but also achieves higher accuracy than all baselines (see Tables 2-3 in our manuscript, lines 425-457 on pages 8-9).** Thus, the SVD-TOGP not only preserves predictive accuracy but also significantly improves the computational efficiency. The results have been added in Appendix L of our **rebuttal revision** (lines 1372-1392, page 26).

---

> ### Author Response · Authors · 2025-11-21
> **Response to Weaknesses 2-3**
>
> ***Weakness 2: Regarding the limited discussion of the scalarization operators and their sensitivity***
>
> **Response to Weakness 2:** Thank you for raising this question. The choices of $L_f$ and $H_f$ are case-specific and determined by the optimization objective. In practice, if the goal is to optimize the overall average behavior of the tensor output, a sum operator can be used. If different locations or modes should contribute according to their relative importance, a weighted sum operator is more suitable. When certain positions need to be emphasized more strongly, an exponential weighted operator can be employed (Chugh, 2020). Based on your suggestion, we additionally consider these three types of operators to evaluate the sensitivity of our method with respect to different choices, defined as follows.
> * Sum operator (used in the main manuscript): $L_f\mathbf{f}(\boldsymbol{x})=\sum f_{i_1,\ldots,i_m}(\boldsymbol x)$; $H_f\tilde{\mathbf{f}}(\boldsymbol{x},\boldsymbol{\lambda})=\sum \tilde f_{i_1,\ldots,i_m}(\boldsymbol x,\boldsymbol{\lambda})$.
>
> * Weighted sum operator: $L_f\mathbf{f}(\boldsymbol{x})=\sum w_{i_1,\ldots,i_m}f_{i_1,\ldots,i_m}(\boldsymbol x)$; $H_f\tilde{\mathbf{f}}(\boldsymbol{x},\boldsymbol{\lambda})=\sum w_{i_1,\ldots,i_m}\tilde f_{i_1,\ldots,i_m}(\boldsymbol x,\boldsymbol{\lambda})$, where $w_{i_1,\ldots,i_m}\sim U(0,1)$.
>
> * Exponential weighted operator: $L_f\mathbf{f}(\boldsymbol{x})=\sum exp\{(pw_{i_1,\ldots,i_m}-1)\}exp\{(pf_{i_1,\ldots,i_m}(\boldsymbol x))\}$; $H_f\tilde{\mathbf{f}}(\boldsymbol{x},\boldsymbol{\lambda})=\sum exp\{(pw_{i_1,\ldots,i_m}-1)\}exp\{(p\tilde f_{i_1,\ldots,i_m}(\boldsymbol x,\boldsymbol{\lambda}))\}$, where $p=2$ and $w_{i_1,\ldots,i_m}\sim U(0,1)$.
>
> The results under Setting (1) in our numerical experiments are summarized below.
>
> |Task|Criterion|Sum operator|Weighted sum operator|Exponential weighted operator|
> |-|-|-|-|-|
> |**BO**|MSE$_x$|0.0000|0.0000|0.0000|
> ||MAE$_y$|0.0008|0.0083|0.0072|
> ||Ins Regret|0.0001|0.0044|0.0533|
> |**CBBO**|MSE$_x$|0.0023|0.0037|0.0025|
> ||MAE$_y$|0.0172|0.0329|0.0548|
> ||Acc|1|1|1|
> ||Ins Regret|0.1964|0.0107|0.4972|
>
> It can be seen that our method is robust across different choices of $L_f$ and $H_f$, and both TOBO and TOCBBO consistently achieve strong optimization performance. The results have been added in Appendix L of our **rebuttal revision** (lines 1538-1576, page 29).
>
> ---
>
> ***Weakness 3: Regarding the omission of tensor GP baselines using Kronecker structure and non-separable kernels***
>
> **Response to Weakness 3:** Thank you for raising this question. The TOGP model proposed in *Scalable High-Order Gaussian Process Regression* has the same covariance structure as sMTGP, assuming the covariance structure of the tensor output is fully separable and using a Kronecker product to define both inter-mode and input correlations. Therefore, the performance of sMTGP is equivalent to that of the TOGP model used in that work. **For the non-separable multi-output GP (NS-MOGP) (Fricker et al., 2013), we additionally provide the results under the three synthetic settings below.**
> |Task|Creterion| Setting (1) | Setting (2) | Setting (3) |
> |-|-|-|-|-|
> |**GP**|NLL|507.19|-39.33|-3094.10|
> ||MAE|0.1923|0.1493|0.1591|
> ||$\Vert Cov\Vert$|6.7398|0.5618|20.5800|
> ||Time (s)|453.11 |151.31 | 1397.68|
> |**BO**|MSE$_x$|0.0014	|0.0005	|0.0244|
> ||MAE$_y$| 0.0536|	0.0503|	0.6277|
> ||Ins Regret| 1.0284|	0.0030|	6.6283|
> ||Time (s)|9109.08 | 2101.45 | 7881.31|
> |**CBBO**|MSE$_x$| 0.0073|	0.0683|	0.0091|
> ||MAE$_y$| 0.0643|0.0390|	0.1257|
> ||Acc| 1|	1	|0.86|
> ||Ins Regret| 1.7389|0.1551|51.4646|
> ||Time (s)|10462.18 |3107.82 |14175.50|
>
> The results have been added in Appendix M of our **rebuttal revision** (lines 1512-1531, page 29). The empirical results show that the NS-MOGP performs worse than our proposed method. In addition, the NS-MOGP are known to be suboptimal in high-dimensional settings, and they do not take advantage of the inherent low-rank structures in tensor-valued outputs (Fricker et al., 2013). Moreover, the runtime of NS-MOGP is substantially longer, as it involves a large number of hyperparameters, which increases the difficulty of parameter estimation and further limits their practical performance.

---

> ### Author Response · Authors · 2025-11-21
> **Response to Questions 1-2**
>
> ***Question 1: Regarding how $L_f$ and $H_f$ are chosen across tasks and their sensitivity***
>
> **Response to Question 1:** Thank you for raising this question. We have answered this question in **Response to Weakness 2**.
>
> ---
>
> ***Question 2: Regarding the choice of tensor ranks and their performance under separable and non-separable kernels***
>
> **Response to Question 2:** Thank you for your comment. In our work, the tensor ranks of the core tensors $\mathbf A_l$ are selected through cross-validation using MAE as the evaluation criterion for $l=1,\cdots,m$. Specifically, we consider a candidate set of ranks $\mathbf R_c=\{(\boldsymbol r_1,\cdots,\boldsymbol r_c)\}$, compute the predictive MAE on the held-out validation set for each candidate, and choose the rank that achieves the lowest validation MAE. Formally, $\boldsymbol{r}^\star=\arg\min_{\boldsymbol{r}\in\mathbf R_c} MAE$, where $MAE = \frac{1}{n_{test}} \sum_{i=1}^{n_{test}} \Vert \frac{\mathbf f_i - \hat{\mathbf f}_i(\boldsymbol{r})}{\mathbf f_i}\Vert$. This procedure ensures that the selected rank provides sufficient expressive power while avoiding overfitting, and it aligns with standard practice in tensor decomposition and GP model selection. We have added this clarification as a remark in Section 3.1 of our **rebuttal revision** (lines 245–250, page 5).
>
> ---
>
> **References:**
>
> [1] Williams C, Seeger M. Using the Nyström method to speed up kernel machines[J]. Advances in neural information processing systems, 2000, 13.
>
> [2] Chugh T. Scalarizing functions in Bayesian multiobjective optimization[C]. 2020 IEEE Congress on Evolutionary Computation (CEC). IEEE, 2020: 1-8.
>
> [3] Zhe S, Xing W, Kirby R M. Scalable high-order gaussian process regression[C]. The 22nd International Conference on Artificial Intelligence and Statistics. PMLR, 2019: 2611-2620.
>
> [4] Fricker T E, Oakley J E, Urban N M. Multivariate Gaussian process emulators with nonseparable covariance structures[J]. Technometrics, 2013, 55(1): 47-56.

---

> > ### Comment · Reviewer_uCf6 · 2025-11-22
> > **Response to the rebuttal**
> >
> > Thank you for the extensive answer to the review. As you can see from the confidence score, this is not my field of expertise, and the AC has been notified about that. I do appreciate the thoroughness of your response and the addition of comparisons on the operator choice and computation times. Also, thank you for pointing towards the Nyström low-rank approximation for dealing with the complexity of higher-dimensional problems.
> >
> > I will leave my score at 6 for being most neutral, but want to emphasize that this is no judgement of your work which I am unable to assess. My rating is weak accept and not weak reject because the quality of presentation, writing, and thoroughness of answers is commendable.
> >
> > Please point the area chair to this comment in case the rating has a bearing on the final decision.

---

### Official Review · Reviewer_f43k · 2025-11-01

**Soundness:** 3
**Presentation:** 3
**Contribution:** 4
**Rating:** 8
**Confidence:** 4

**Summary:**

The paper investigates optimization problems involving tensor-output objective functions, where each function evaluation returns a multi-dimensional tensor instead of a vector, as in standard multi-output settings. The authors introduce a Tensor-Output Gaussian Process (TOGP) that captures both dependencies across inputs and structured correlations within and between tensor modes of outputs. The proposed TOGP generalizes and introduces two types of kernels: a separable kernel, which assumes constant correlations across tensor modes, and a non-separable kernel, which allows correlations within and across tensor modes to vary with the inputs.

The authors further apply the TOGP to Bayesian Optimization (BO) in two scenarios. In the fully observed setting (TOBO), all tensor entries are available at each iteration, and optimization proceeds via an Upper Confidence Bound acquisition function derived from the TOGP posterior. In the partially observed setting (TOCBBO), only a subset of tensor entries is observed per iteration, transforming the problem into a Combinatorial Bandit Bayesian Optimization (CBBO) framework. Here, the algorithm must select both the next input and the subset of tensor entries (the super-arm) to observe. The paper provides theoretical guarantees, including proofs of kernel validity and sublinear regret bounds for both TOBO and TOCBBO. Extensive evaluations include three synthetic and four real-world benchmark problems to compare against many multi-output GP baselines.

**Strengths:**

-	The idea is sound, which generalizes the well-known LMC model in multi-output (vector-output) GP into the tensor-input GPs.
-	The paper is clearly written, with theoretical proof for the kernel validity and regret bound under certain assumptions.
-	The empirical performance shows that TOGP has potential in modelling tensor-output functions, when compared to vector-output GPs.

**Weaknesses:**

Some experiments are missing:
- Lack of single-objective BO baselines: Since the target problem settings rely on a scalarization operator (L_f or H_f in the CBBO case) to convert tensor outputs into scalar objectives, standard single-objective BO methods (e.g., single-task GP combined with a UCB acquisition function) should, in principle, operate normally. The paper would be stronger if it compared TOGP-based optimization with such single-task GP baselines to demonstrate the benefit of modelling complex tensor dependencies rather than relying on a simpler approach.
- Lack of single-objective BO baselines in the Bandit BO setting: Similarly, in the Bandit BO setting, the presence of the scalarization operator (L_f or H_f) allows single-objective BO algorithms to be applied directly. Prior works (e.g., Nguyen et al., 2020; Ru et al., 2020; Huang et al., 2022) already use this setup effectively, and including these as baselines would provide a fairer and more comprehensive comparison for TOCBBO.
- Lack of multi-objective BO baselines: Because TOGP inherently models multiple correlated outputs, it would be valuable to assess its performance under a multi-objective optimization framework. Comparing against the SOTA MOBO methods using metrics such as the hypervolume indicator (e.g., Maddox et al., 2021; Daulton et al., 2022) would further complement the paper.

**Questions:**

1. What is the runtime trade-off of the proposed TOGP? Since TOGP models complex interactions across multiple tensor modes, it likely incurs higher computational costs than both single-task GPs and multi-output GPs based on the LMC, due to the larger number of hyperparameters involved. Additionally, given that the problem settings already require a scalarization operator (L_f or H_f) to map tensor outputs to scalar objectives, which allows single-task GP to work just fine, it would be useful to analyze the accuracy–efficiency trade-off - that is, how much performance gain the tensor-output model provides compared to the reduced computational cost relative to a simpler single-task GP baseline.
2. Does the proposed TOBO algorithm account for the diversity of solutions? Given that the target problems have multiple outputs, some of which may be conflicting, can TOBO handle such cases, identify a set of solutions that satisfy different objectives or trade-offs? This is a common consideration in MOBO literature, where the goal is not only to find the best single solution but also to capture the Pareto diversity among competing objectives.

---

> ### Author Response · Authors · 2025-11-21
> **Response to Weaknesses 1-2**
>
> Thank you very much for your valuable suggestions. We have made point-by-point responses to address all raised questions and concerns. The **rebuttal revision** has also been submitted. All changes in the rebuttal version of the manuscript have been **red** for clarity.
>
> ---
>
> ***Weakness 1: Regarding the lack of standard single-objective BO baselines***
>
> **Response to Weakness 1:** Thank you for raising this question. In the BO setting, we additionally include a single-objective GP-UCB baseline, which directly assumes that $L_f\mathbf f$ follows a GP and uses the UCB acquisition function to select query points sequentially. We evaluate optimization accuracy using two criteria defined as follows.
> * Mean Square Error between $\boldsymbol{x}^\star$ and $\boldsymbol{x}_n$: MSE$_x=\Vert\boldsymbol{x}^\star-\boldsymbol{x}_n\Vert^2$.
> * Ins regret: $r_n = L_f\mathbf f(\boldsymbol x^\star)-L_f\mathbf f(\boldsymbol x_{n})$.
>
> The optimization results under the three settings in our numerical experiments are summarized below.
> |Setting|Critetion|Single GP-UCB|TOBO|
> |-|-|-|-|
> |**Setting (1)**|MSE$_x$|0.0004|**0.0000**|
> ||Ins Regret|0.0459|**0.0001**|
> ||Time|**457.34**|6968.25|
> |**Setting (2)**|MSE$_x$|0.4761|**0.0003**|
> ||Ins Regret|0.2093|**0.0002**|
> ||Time|**187.3**|1588.53|
> |**Setting (3)**|MSE$_x$|0.0084|**0.0001**|
> ||Ins Regret|0.1964|**0.0302**|
> ||Time|**415.61**|6035.61|
>
> The results have been added in Appendix M of our rebuttal revision (lines 1465–1477, page 28). As shown, TOBO consistently achieves better optimization accuracy than Single GP-UCB across all settings. Moreover, TOGP provides predictive means and uncertainties for every element of the tensor output, which enables its direct application to the CBBO framework. In contrast, Single GP-UCB only models the scalarization-based objective and therefore cannot be used in the CBBO setting.
>
> Although the runtime of TOBO is higher, we further introduce an efficient variant, SVD-TOBO, which applies a low-rank eigen-decomposition to approximate the TOGP covariance. The Nyström method approximates $\mathbf{K}_n$ by selecting $n_l$ ($\ll nT$) landmark columns and extending a small eigendecomposition, which reduces the computational complexity of evaluating $(\mathbf K_n+\eta \mathbf I)^{-1}$ from $\mathcal{O}(n^3T^3)$ to $\mathcal{O}(nTn_l^2 +n_l^3)$ (Williams & Seeger 2000). The detailed statement and corresponding results for SVD-TOBO are provided in **Response to Question 1**.
>
> ---
>
> ***Weakness 2: Regarding the lack of single-objective BO baselines in the bandit BO setting***
>
> **Response to Weakness 2:** Thank you for raising this question. In the CBBO setting, using a single GP as the surrogate of $L_f \mathbf f$ cannot be used to select the super-arms of $\mathbf f$ because it does not provide the posterior distribution of $\mathbf f$ at each tensor element. Existing work on bandits with structured outputs (such as Nguyen et al. 2020, Ru et al. 2020, Huang et al. 2022) typically employs an independent GP model for each element of $\mathbf f$. Thus, we add an independent GP-UCB baseline (Ind GP-UCB), which assumes that each tensor element follows its own GP model and uses the proposed CMAB-UCB2 acquisition function to sequentially select both query points and super-arms. The optimization results under the three settings used in our numerical experiments are summarized below.
>
> |Setting|Critetion|Ind GP-UCB|TOBO|
> |-|-|-|-|
> |**Setting (1)**|MSE$_x$|0.2121	|**0.0023**|
> ||MAE$_y$|0.6739|	**0.0172**|
> ||Acc|0.67	|**1**|
> ||Ins Regret|36.301|	**0.9807**|
> ||Time|**328.56**|	8017.08|
> |**Setting (2)**|MSE$_x$|0.0731|	**0.0000**|
> ||MAE$_y$|0.0486	|**0.0000**|
> ||Acc|**1**	|**1**|
> ||Ins Regret|0.1561	|**0.0000**|
> ||Time|**110.77**	|2348.16|
> |**Setting (3)**||MAE$_y$|0.0139	|**0.0021**|
> ||Acc|0.86|	**1**|
> ||Ins Regret|54.2714	|**1.4406**|
> ||Time|**3296.72**|	11699.09|
>
> The results have been added in Appendix M of our rebuttal revision (lines 1491–1506, page 28). As shown, TOBO significantly outperforms Ind GP-UCB in terms of optimization accuracy across all settings. Although the runtime of TOBO is higher, we further introduce an efficient variant, SVD-TOBO, which applies a low-rank eigen-decomposition to approximate the TOGP covariance. The Nyström method approximates $\mathbf{K}_n$ by selecting $n_l$ ($\ll nT$) landmark columns and extending a small eigendecomposition, which reduces the computational complexity of evaluating $(\mathbf K_n+\eta \mathbf I)^{-1}$ from $\mathcal{O}(n^3T^3)$ to $\mathcal{O}(nTn_l^2 +n_l^3)$ (Williams & Seeger 2000). The detailed statement and corresponding results for SVD-TOBO are provided in **Response to Question 1**.

---

> ### Author Response · Authors · 2025-11-21
> **Response to Weakness 3 and Question 1 (Part A)**
>
> ***Weakness 3: Regarding the lack of multi-objective BO baselines***
>
> **Response to Weakness 3:** Thank you for raising this question. However, existing hypervolume-based and entropy search-based MOBO methods are not applicable when the number of objectives $T>3$ due to their prohibitive computational complexity. The tensors used in our experiments have sizes $T = 6, 16, 40$, which correspond to objective dimensions far beyond the practical limits of these MOBO methods. Specifically,
>
> * Hypervolume-based MOBO methods do not scale to tensor outputs. These methods require computing the dominated hypervolume in an $T$-dimensional objective space, and their computational complexity grows super-polynomially with $T$. For example, the q-expected hypervolume improvement (qEHVI) method has a complexity of $\mathcal O(TNK(2^q-1))$ (Daulton et al., 2020). Here, $K$ denotes the number of hyperrectangles in the decomposition of the non-dominated region. When $T=2$, $K$ already grows super-polynomial in $T$, and when $T\geq4$, the number of boxes required for a decomposition of the non-dominated region is unknown (Yang et al., 2019). It is widely known that the complexity of hypervolume computation grows exponentially with both $T$ and the size of the Pareto set (Daulton et al., 2020). Thus, hypervolume-based acquisition functions are not applicable to TOBO.
>
> * Entropy search-based MOBO methods also do not scale to tensor outputs. In particular, these methods require estimating the entropy of either the optimal objective vector or the Pareto set. For example, joint entropy search (JES) uses independent GPs as surrogates and has a total computational cost of $\mathcal O(Tn^3+Sp^{\lfloor T/2\rfloor+1})$ (Tu et al., 2022). Here $S$ is the number of Monte Carlo samples, and $p$ is the size of the sampled Pareto set. The term $p^{\lfloor T/2 \rfloor + 1}$ grows extremely rapidly with $T$, making the box decomposition step computationally prohibitive even for moderate objective dimensions. Thus, entropy search-based MOBO methods are also not applicable to our tensor-output setting.
>
> Therefore, these methods are not applicable in our setting, and that is the reason why we did not compare the performance with them.
>
> ---
>
> ***Question 1: Regarding the request to clarify the runtime cost of TOGP and its accuracy-efficiency trade-off against single-task GP baselines***
>
> **Response to Question 1:** Thank you for your valuable comment. We have provided the runtime for GP training of different methods under the three synthetic settings used in our numerical experiments. The results are summarized below.
> |Setting|TOGP|sMTGP|MLGP|MVGP|
> |-|-|-|-|-|
> |**Setting (1)**|266.04|27.26|30.57|**6.50**|
> |**Setting (2)**|69.85|16.08|21.99|**1.31**|
> |**Setting (3)**|900.13|851.34|891.73|**769.50**|
>
> We also provide the runtime of different methods for BO and CBBO under the three synthetic settings. The results are summarized below.
> |Setting|Method|TOBO/TOCBBO|TOGP-RS|sMTGP-UCB|sMTGP-RS|MLGP-UCB|MLGP-UCB|MVGP-UCB|MVGP-UCB|
> |--|--|--|--|--|--|--|--|--|--|
> |**Setting (1)**|BO|6968.25|6176.84|1516.31|1503.24|1545.41|1530.32|340.44|**334.79**|
> ||CBBO|8017.08|7990.44|953.82|939.27|981.86|971.25|437.29|**386.86**||
> |**Setting (2)**|BO|1588.53|1573.36|117.78|106.93|133.93|109.14|14.86|**7.70**||
> ||CBBO|2348.16|2338.97|179.11|164.95|185.49|180.70|20.76|**6.70**||
> |**Setting (3)**|BO|6035.61|5786.02|5291.26|5273.84|5452.90|5252.32|3006.97|**2997.75**|
> ||CBBO|11699.09|11501.01|9436.90|9421.08|9696.60|9431.96|3199.47|**3116.23**|
>
> We have included these computational results in Appendix L of our **rebuttal revision** (lines 1323-1371, pages 25-26). It can be observed that the proposed TOGP has the longest training time among all methods. This is mainly due to two reasons. First, in our numerical experiments, we use the non-separable tensor-output kernel introduced in Definition 1 to build a more general TOGP model, which leads to a computational complexity of $O(n^3T^3)$. In TOBO and TOCBBO, since the hyperparameters are updated at each round, the cost of selecting the query point at round $n$ is also $O(n^3T^3)$. In contrast, the baseline methods adopt kernels with different levels of separability, where the covariance matrix can be represented using Kronecker products, resulting in lower computational cost. Second, all experiments were conducted on a Windows workstation equipped with an Intel Core i9-9960X CPU (16 cores) and 128 GB RAM. With a more powerful server, the running time would be further reduced.

---

> ### Author Response · Authors · 2025-11-21
> **Response to Question 1 (Part B)**
>
> Despite this, the computational cost of TOGP remains affordable. In Setting (3), where the tensor dimension is $4 \times 5 \times 2$, the total computation time for sequentially adding $30$ new points on top of $20$ initial design points is 6035.61 seconds. More importantly, TOGP, TOBO, and TOCBBO achieve significantly better fitting and optimization performance than all baselines (see Tables 2 and 3, lines 425–457 on page 8). **The additional computational cost is therefore well justified by the significant improvements in accuracy.** Moreover, BO is typically applied to expensive black-box optimization scenarios. In such settings, the cost of data acquisition often far exceeds the computational cost of training the surrogate model, and the optimization performance is far more critical than the running time.
>
> **For large tensor outputs where the computational cost may become prohibitive, we have also introduced a Nyström low-rank approximation strategy to improve the scalability of TOGP.** Specifically, based on data $\mathbf{X_n}$ and $\boldsymbol Y_n$, consider the spectral decomposition $\mathbf K_n=\mathbf U_n\mathbf\Lambda_n\mathbf U_n^\top$, where $\mathbf\Lambda_n = \mathrm{diag}(\lambda_1,\lambda_2,\ldots,\lambda_{nT})$ denotes eigenvalues ordered as $\lambda_1\ge\lambda_2\ge\ldots\ge\lambda_{nT}>0$. We approximate $\mathbf K_n$ using the leading $l$ eigenpairs ($l\ll nT$), that is, $\mathbf K_n\approx\mathbf U_l\mathbf \Lambda_l\mathbf U_l^\top$, where $\mathbf U_l \in \mathbb R^{nT\times l}$ contains the first $l$ eigenvectors and $\mathbf\Lambda_l=\mathrm{diag}(\lambda_1,\ldots,\lambda_l)$. The rank $l$ is selected by cumulative explained variance:
> $$l=\min_{l_0}\{l_0 \in \{1,\ldots,nT\}:\sum_{i=1}^{l_0} \lambda_i/\sum_{i=1}^{nT}\lambda_i\ge c\}.$$
> The Nyström method approximates $\mathbf{K}_n$ by selecting $n_l$ ($\ll nT$) landmark columns and extending a small eigendecomposition, which yields a computational cost of evaluating $(\mathbf K_n+\eta \mathbf I)^{-1}$ as $\mathcal{O}(nTn_l^2 +n_l^3)$ (Williams & Seeger 2000). Thus, the overall computational complexity for training TOGP becomes $\mathcal O\left((nTn_l^2 + n_l^3 + n^2 T^2 m_h)\log n\right)$. We have added it as a remark in Section 3.1 of our **rebuttal revision** (lines 252-261, page 5).
>
> Finally, we provide the prediction and optimization performance of TOGP based on the Nyström low-rank approximation (SVD-TOGP) under the three synthetic settings. The results are summarized below.
>
> |||Setting (1)||Setting (2)||Setting (3)||
> |-|-|-|-|-|-|-|-|
> |**Task**|**Criterion**|**SVD-TOGP**|**TOGP**|**SVD-TOGP**|**TOGP**|**SVD-TOGP**|**TOGP**|
> |**GP**|NLL|557.09|503.00|-18.39|-18.10|-3778.74|-3923.1|
> ||MAE|0.1756|0.1571|0.1099|0.1052|0.1436|0.1372|
> ||$\Vert Cov\Vert$|3.9627| 2.0200|0.6261|0.0400|0.0881|2.8200|
> ||Time (s)|**21.8**|266.04|**10.83**|69.85|**436.17**|900.13|
> |**BO**|MSE$_x$|0.0001|0.0000|0.0003|0.0003|0.0002|0.0001|
> ||MAE$_y$|0.0041|0.0008|0.0356|0.0350|0.0051|0.0050|
> ||Ins Regret|0.0040|0.0001|0.0002|0.0002|0.0402|0.0302|
> ||Time (s)|**831.02**|6968.25|**162.78**|1588.53|**3604.66**|6035.61|
> |**CBBO**|MSE$_x$|0.0024|0.0023|0.0000|0.0000|0.0035|0.0021|
> ||MAE$_y$|0.0180|0.0172|0.0000|0.0000|0.0246|0.0145|
> ||Acc|1|1|1|1|1|1|
> ||Ins Regret|1.0944|0.9807|0.0000|0.0000|1.8571|1.4406|
> ||Time (s)|**655.83**|8017.08|**134.22**|2348.16|**4446.12**|11699.09|
>
> **We can see that SVD-TOGP not only greatly reduces the computational time but also achieves higher accuracy than all baselines (see Tables 2-3 in our manuscript, lines 425-457 on pages 8-9).** Thus, the SVD-TOGP not only preserves predictive accuracy but also significantly improves computational efficiency. The results have been added in Appendix L of our **rebuttal revision** (lines 1372-1392, page 26).

---

> ### Author Response · Authors · 2025-11-21
> **Response to Question 2**
>
> ***Question 2: Regarding the concern about whether TOBO accounts for output diversity***
>
> **Response to Question 2:** Thank you for raising this concern. The proposed TOBO and TOCBBO frameworks are scalarization-based methods that aim to find the best single solution rather than capturing Pareto diversity. The operators $L_f$ and $H_f$ used in TOBO and TOCBBO allow users to encode their preferences. For example, assigning different weights to different tensor modes, which enables prioritization across the output structure. In addition, we emphasize that existing MOBO methods designed to capture Pareto diversity, including hypervolume-based and entropy search-based methods, are not applicable in our setting due to their prohibitive computational complexity (Daulton et al., 2020; Tu et al., 2020). Therefore, scalarization-based strategies provide the only computationally feasible design for TOBO. The detailed explanations have been answered in **Response to Weakness 3**.
>
> We have added this clarification in Section 2 of our **rebuttal revision** (lines 128–130, page 3). We also added a discussion in Section 6 (lines 538–539, page 10) noting that developing computationally efficient hypervolume-based or entropy search-based criteria for TOBO and TOCBBO can be explored in the future.
>
> ---
> **References:**
>
> [1] Nguyen D, Gupta S, Rana S, et al. Bayesian optimization for categorical and category-specific continuous inputs[C]. Proceedings of the AAAI Conference on Artificial Intelligence. 2020, 34(04): 5256-5263.
>
> [2] Ru B, Alvi A, Nguyen V, et al. Bayesian optimisation over multiple continuous and categorical inputs[C]//International Conference on Machine Learning. PMLR, 2020: 8276-8285.
>
> [3] Huang C, Lee J, Zhang Y, et al. Mixed-input Bayesian optimization method for structural damage diagnosis[J]. IEEE Transactions on Reliability, 2022, 72(2): 678-691.
>
> [4] Williams C, Seeger M. Using the Nyström method to speed up kernel machines[J]. Advances in neural information processing systems, 2000, 13.
>
> [5] Daulton S, Balandat M, Bakshy E. Differentiable expected hypervolume improvement for parallel multi-objective Bayesian optimization[J]. Advances in neural information processing systems, 2020, 33: 9851-9864.
>
> [6] Yang K, Emmerich M, Deutz A, et al. Efficient computation of expected hypervolume improvement using box decomposition algorithms[J]. Journal of Global Optimization, 2019, 75(1): 3-34.
>
> [7] Tu B, Gandy A, Kantas N, et al. Joint entropy search for multi-objective Bayesian optimization[J]. Advances in Neural Information Processing Systems, 2022, 35: 9922-9938.

---

### Official Review · Reviewer_AnPz · 2025-11-01

**Soundness:** 3
**Presentation:** 2
**Contribution:** 2
**Rating:** 6
**Confidence:** 3

**Summary:**

This paper presents two variants of GP-UCB for BO of tensor-valued objectives. Both model the objective using a tensor-output GP (TOGP) with the goal of capturing additional structure present in the tensor representation, with a kernel that is somewhat novel to capture correlation between modes. The first algorithm uses scalarization - basically a weighted sum of the ouputs - while the second does the same on a subset of outputs (partial observations) at each round. Convergence rates are analysed to show sub-linear regret, and some experimental results are presented.

**Strengths:**

The major strength/novelty of this paper appears to be the TOGP model, in particular the kernel used. And the kernel used for this GP is novel, at least to my understanding.

That said, at least the first algorithm presented does not seem particularly novel to me: it is basically a scalarizing multi-objective BO algorithm. Each element of the tensor represents an objective, and a linearization operator takes the place of scalarization. The second algorithm is arguably more novel, as it assumes partial observations of the tensor components at each iteration; however, past this, it again boils down to a scalarized MOBO formulation.

Experiments seem to have improved since last time I reviewed this paper, and experimental result seem sound, so I am inclined toward accepting.

**Weaknesses:**

See above

**Questions:**

Relevant questions were answered last time I reviewed this paper.

---

> ### Author Response · Authors · 2025-11-21
> **Response**
>
> Thank you very much for your valuable comments and encouragement. In this revision, we have clarified the contributions more clearly.
>
> * First, we propose a tensor-output Gaussian process (TOGP) with two classes of tensor-output kernels that explicitly incorporate tensor structure by extending the linear model of coregionalization from vector-valued outputs to tensor-valued outputs. The proposed kernels capture rich dependencies across tensor modes and across the input domain. Using the TOGP model as a surrogate, we develop a TOBO framework based on the upper confidence bound (UCB) acquisition. We establish a sublinear regret bound for TOBO, which is the first regret analysis for tensor-valued outputs under a Bayesian framework. It is nontrivial because it requires extending scalar concentration inequalities to a tensor-output setting and developing new concentration bounds tailored to the proposed model.
>
> * Second, we formulate a novel problem setting, referred to as CBBO. To address this setting, we design the TOCBBO framework by introducing a CMAB-UCB2 acquisition function, which integrates the UCB criterion for input selection with the CMAB-UCB criterion for super-arm selection. We further establish a sublinear regret bound for TOCBBO. This theoretical result is also nontrivial, as it requires extending scalar concentration inequalities to tensor-valued outputs.
>
> In addition, we emphasize that existing MOBO methods designed to capture Pareto diversity, including hypervolume-based and entropy search-based methods, are not applicable in our setting due to their prohibitive computational complexity. Therefore, scalarization-based strategies provide the only computationally feasible design for tensor-output BO.
>
> * Hypervolume-based MOBO methods do not scale to tensor outputs. In particular, these methods rely on computing the dominated hypervolume in an $M$-dimensional objective space. Their computational complexity grows super-polynomially with the number of objectives $T$. For example, f q-expected hypervolume improvement (qEHVI) has a complexity of $\mathcal O(TNK(2^q-1))$ (Daulton et al., 2020). Here $K$ is the number of hyperrectangles in the decomposition of the non-dominated region. When $T=2$, $K$ is super-polynomial in $T$. When $T\geq4$, the number of boxes required for a decomposition of the non-dominated region is unknown (Yang et al., 2019). It is widely known that the complexity of hypervolume computation grows exponentially with both $T$ and the size of the Pareto set (Daulton et al., 2020). In our TOBO framework, the tensor-output dimension is given by  $T=t_1\times\cdots t_m$. The experimental settings include tensors of sizes $T = 6, 16, 40$, which correspond to objective dimensions far beyond what hypervolume-based MOBO can handle in practice. Thus, hypervolume-based acquisition functions are not applicable to TOBO.
>
> * Entropy search-based MOBO methods also do not scale to tensor outputs. In particular, these methods require estimating the entropy of either the optimal objective vector or the Pareto set. For example, joint entropy search (JES) uses independent GPs as surrogates and has a total computational cost of $\mathcal O(Tn^3+Sp^{\lfloor T/2\rfloor+1})$ (Tu et al., 2022). Here $S$ is the number of Monte Carlo samples, and $p$ is the size of the sampled Pareto set. The term $p^{\lfloor T/2 \rfloor + 1}$ grows extremely rapidly with $T$, making the box decomposition step computationally prohibitive even for moderate objective dimensions. Thus, entropy search-based MOBO methods are also not applicable to our tensor-output setting.
>
> In summary, existing MOBO methods that aim to capture Pareto diversity are not suitable for tensor-output objectives due to fundamental computational barriers. For this reason, the proposed TOBO and TOCBBO adopt a scalarization-based strategy, which is the only tractable method for TOBO and TOCBBO. We have added this clarification in Section 2 of our **rebuttal revision** (lines 128–130, page 3). We also added a discussion in Section 6 (lines 538–539, page 10) noting that developing computationally efficient hypervolume-based or entropy search-based criteria for TOBO and TOCBBO can be explored in the future.
>
> Finally, we appreciate your positive evaluation of the improved experiments. Thank you again for your constructive and valuable feedback.
>
> ---
>
> **References:**
>
> [1] Daulton S, Balandat M, Bakshy E. Differentiable expected hypervolume improvement for parallel multi-objective Bayesian optimization[J]. Advances in neural information processing systems, 2020, 33: 9851-9864.
>
> [2] Yang K, Emmerich M, Deutz A, et al. Efficient computation of expected hypervolume improvement using box decomposition algorithms[J]. Journal of Global Optimization, 2019, 75(1): 3-34.
>
> [3] Tu B, Gandy A, Kantas N, et al. Joint entropy search for multi-objective Bayesian optimization[J]. Advances in Neural Information Processing Systems, 2022, 35: 9922-9938.

---

### Official Review · Reviewer_9BYU · 2025-11-03

**Soundness:** 3
**Presentation:** 2
**Contribution:** 3
**Rating:** 6
**Confidence:** 4

**Summary:**

- The paper introduces Tensor-Output Gaussian Processes (TOGPs) that model structured tensor-valued outputs with novel separable and non-separable kernels capturing input-dependent correlations.
- Building on TOGP, the authors propose Tensor-Output Bayesian Optimization (TOBO) using a UCB acquisition function for efficient optimization of expensive black-box tensor-output functions.
- They further extend the framework to a Combinatorial Bandit Bayesian Optimization (TOCBBO) setting where only a subset of tensor outputs can be observed.
- Both TOBO and TOCBBO are theoretically analyzed, with sublinear regret bounds.
- Extensive synthetic and real-world experiments demonstrate that the proposed methods outperform several baselines in predictive accuracy and optimization effectiveness.

**Strengths:**

- The manuscript is well written with polished language that enhances readability and effectively communicates complex technical ideas.
- It introduces a novel Tensor-Output Gaussian Process (TOGP) framework with separable and non-separable kernels.
- The framework extends Bayesian Optimization to handle tensor-valued functions.

**Weaknesses:**

- The paper's main novelty lies in the Gaussian Process modeling, while the Bayesian Optimization layer mainly adapts standard UCB and CMAB-UCB2 frameworks without introducing a fundamentally new acquisition strategy.
- The experimental evaluation emphasizes predictive accuracy metrics (MSE, MAE, NLL) more than optimization-oriented ones, creating a mismatch with the paper’s stated BO focus.
- The paper lacks runtime or computational efficiency results, reporting only asymptotic complexity analyses without empirical timing comparisons.

**Questions:**

- The paper's primary contribution appears to lie in the Gaussian Process modeling rather than in the Bayesian Optimization framework itself. Could the authors clarify whether the main contribution is intended to advance GP modeling or BO methodology?
- Given that ICLR allows for a revision, the authors might be able to revise or extend the work to more clearly strengthen its contribution to the BO literature?
- While theoretical computational complexities are analyzed, no empirical runtime or efficiency results are provided. Could the authors report or estimate the practical computational overhead of TOGP and TOCBBO compared with existing multi-output GP or BO methods?
- How does the improved predictive accuracy of TOGP quantitatively translate into better optimization results such as faster regret decay or fewer evaluations? Have the authors conducted any ablation or correlation analysis to demonstrate this link?
- Could the authors consider extending their framework to more meaningful tensor structures, such as spatiotemporal tensors that capture both spatial and temporal dependencies?

**Details Of Ethics Concerns:**

There is no particular ethical concern.

---

> ### Author Response · Authors · 2025-11-21
> **Response to Weaknesses 1-2**
>
> Thank you very much for your valuable suggestions. We have provided point-by-point responses to address all questions and concerns. The **rebuttal revision** has also been submitted. All changes in the revised rebuttal manuscript have been highlighted in **red** for clarity.
>
> ---
>
> ***Weakness 1: Regarding the concern that the BO layer mainly applies standard UCB and CMAB-UCB2 without introducing a new acquisition strategy***
>
> **Response to Weakness 1:** Thank you for raising this question. In our work, the main contributions focus on two aspects.
> * First, we propose a tensor-output Gaussian process (TOGP) with two classes of tensor-output kernels that explicitly incorporate tensor structure by extending the linear model of coregionalization from vector-valued outputs to tensor-valued outputs. The proposed kernels capture rich dependencies across tensor modes and across the input domain. Using the TOGP model as a surrogate, we develop a TOBO framework based on the upper confidence bound (UCB) acquisition. We establish a sublinear regret bound for TOBO, which is the first regret analysis for tensor-valued outputs under a Bayesian framework. It is nontrivial because it requires extending scalar concentration inequalities to a tensor-output setting and developing new concentration bounds tailored to the proposed model.
>
> * Second, we formulate a novel problem setting, referred to as CBBO. To address this setting, we design the TOCBBO framework by introducing a CMAB-UCB2 acquisition function, which integrates the UCB criterion for input selection with the CMAB-UCB criterion for super-arm selection. We further establish a sublinear regret bound for TOCBBO. This theoretical result is also nontrivial, as it requires extending scalar concentration inequalities to tensor-valued outputs.
>
> Therefore, the focus of this work is not on designing a new acquisition function. Our primary contribution is to develop a unified BO framework and a CBBO framework with theoretical guarantees for complex black-box tensor-output functions. We have clarified this point in Section 1 of our **rebuttal revision** (lines 76-91, page 2). The design of new acquisition functions can also be explored within this framework. For example, one may combine the TOGP model with EI or PI with theoretical guarantees and further extend them to the TOCBBO framework. We have added this discussion in Section 6 of our **rebuttal revision** (lines 533-538, page 10).
>
> ---
>
> ***Weakness 2: Regarding the concern that the experiments emphasize predictive accuracy more than optimization-focused metrics***
>
> **Response to Weakness 2:** We actually conducted extensive numerical studies and case studies to evaluate the performance of different methods from an optimization perspective. The BO performance criteria (see Table 3 and Figures 1-3, lines 441-485 and 498-522 on pages 9-10) include:
> * Mean Square Error between $\boldsymbol{x}^\star$ and $\boldsymbol{x}_n$: $MSE_x=\Vert\boldsymbol{x}^\star-\boldsymbol{x}_n\Vert^2$.
>
> * Mean Absolute Error between $\mathbf{f}(\boldsymbol{x}^\star)$ and $\mathbf{f}(\boldsymbol{x}_n)$: $MAE_y=\frac{\Vert\mathbf{f}(\boldsymbol{x}^\star)-\mathbf{f}(\boldsymbol{x}_n)\Vert}{\Vert\mathbf{f}(\boldsymbol{x}^\star)\Vert}$.
>
> * Instantaneous regret: $r_n = L_f\mathbf f(\boldsymbol x^\star)-L_f\mathbf f(\boldsymbol x_{n})$.
>
> The CBBO performance criteria (see Table 3 and Figures 1-3, lines 441-485 and 498-522 on pages 9-10) include:
> * Mean Square Error between $\boldsymbol{x}^\star$ and $\boldsymbol{x}_n$: $MSE_x=\Vert\boldsymbol{x}^\star-\boldsymbol{x}_n\Vert^2$.
>
> * Mean Absolute Error between $\tilde{\mathbf f}(\boldsymbol{x}^\star,\boldsymbol\lambda^\star)$ and $\tilde{\mathbf f}(\boldsymbol{x}_n,\boldsymbol\lambda_n)$: $MAE_y=\frac{\Vert\tilde{\mathbf f}(\boldsymbol{x}^\star,\boldsymbol\lambda^\star)-\tilde{\mathbf f}(\boldsymbol{x}_n,\boldsymbol\lambda_n)\Vert}{\Vert\tilde{\mathbf f}(\boldsymbol{x}^\star,\boldsymbol\lambda^\star)\Vert}$.
>
> * Accuracy: $Acc=\mathbb I\{(\boldsymbol\lambda^\star=\boldsymbol\lambda_n)\}/T$.
>
> * Instantaneous regret: $r_n = H_f\tilde{\mathbf f}(\boldsymbol x^\star)-H_f\tilde{\mathbf f}(\boldsymbol x_{n})$.
>
> Therefore, the emphasis of our experimental evaluation is on BO and CBBO optimization criteria rather than predictive accuracy alone. The GP fitting criteria are included only to verify that the surrogate model is well calibrated and suitable for the supporting optimization tasks, whereas the BO and CBBO criteria directly measure optimization quality. Moreover, if there are additional optimization evaluation criteria that can be considered, we would sincerely appreciate any further suggestions you may have.

---

> ### Author Response · Authors · 2025-11-21
> **Response to Weakness 3 (Part A)**
>
> ***Weakness 3: Regarding the lack of empirical runtime or computational efficiency results***
>
> **Response to Weakness 3:** We have provided the runtime for GP training of different methods under the three synthetic settings used in our numerical experiments. The results are summarized below.
> |Setting|TOGP|sMTGP|MLGP|MVGP|
> |-|-|-|-|-|
> |**Setting (1)**|266.04|27.26|30.57|**6.50**|
> |**Setting (2)**|69.85|16.08|21.99|**1.31**|
> |**Setting (3)**|900.13|851.34|891.73|**769.50**|
>
> We also provide the runtime of different methods for BO and CBBO under the three synthetic settings. The results are summarized below.
> |Setting|Method|TOBO/TOCBBO|TOGP-RS|sMTGP-UCB|sMTGP-RS|MLGP-UCB|MLGP-UCB|MVGP-UCB|MVGP-UCB|
> |--|--|--|--|--|--|--|--|--|--|
> |**Setting (1)**|BO|6968.25|6176.84|1516.31|1503.24|1545.41|1530.32|340.44|**334.79**|
> ||CBBO|8017.08|7990.44|953.82|939.27|981.86|971.25|437.29|**386.86**||
> |**Setting (2)**|BO|1588.53|1573.36|117.78|106.93|133.93|109.14|14.86|**7.70**||
> ||CBBO|2348.16|2338.97|179.11|164.95|185.49|180.70|20.76|**6.70**||
> |**Setting (3)**|BO|6035.61|5786.02|5291.26|5273.84|5452.90|5252.32|3006.97|**2997.75**|
> ||CBBO|11699.09|11501.01|9436.90|9421.08|9696.60|9431.96|3199.47|**3116.23**|
>
> We have included these computational results in Appendix L of our **rebuttal revision** (lines 1323-1371, pages 25-26). It can be observed that the proposed TOGP has the longest training time among all methods. This is mainly due to two reasons. First, in our numerical experiments, we use the non-separable tensor-output kernel introduced in Definition 1 to build a more general TOGP model, which leads to a computational complexity of $O(n^3 T^3)$. In TOBO and TOCBBO, since the hyperparameters are updated at each round, the cost of selecting the query point at round $n$ is also $O(n^3 T^3)$. In contrast, the baseline methods adopt kernels with different levels of separability, where the covariance matrix can be represented using Kronecker products, resulting in lower computational cost. Second, all experiments were conducted on a Windows workstation equipped with an Intel Core i9-9960X CPU (16 cores) and 128 GB RAM. With a more powerful server, the running time would be further reduced.
>
> Despite this, the computational cost of TOGP remains affordable. In Setting (3), where the tensor dimension is $4 \times 5 \times 2$, the total computation time for sequentially adding $30$ new points on top of $20$ initial design points is 6035.61 seconds. More importantly, TOGP, TOBO, and TOCBBO achieve significantly better fitting and optimization performance than all baselines (see Tables 2 and 3, lines 425–457 on page 8). **The additional computational cost is therefore well justified by the significant improvements in accuracy.** Moreover, BO is typically applied to expensive black-box optimization scenarios. In such settings, the cost of data acquisition often far exceeds the computational cost of training the surrogate model, and the optimization performance is far more critical than the running time.

---

> ### Author Response · Authors · 2025-11-21
> **Response to Weakness 3 (Part B)**
>
> **For large tensor outputs where the computational cost may become prohibitive, we have also introduced a Nyström low-rank approximation strategy to improve the scalability of TOGP.** Specifically, based on data $\mathbf{X_n}$ and $\boldsymbol Y_n$, consider the spectral decomposition $\mathbf K_n=\mathbf U_n\mathbf\Lambda_n\mathbf U_n^\top$, where $\mathbf\Lambda_n = \mathrm{diag}(\lambda_1,\lambda_2,\ldots,\lambda_{nT})$ denotes eigenvalues ordered as $\lambda_1\ge\lambda_2\ge\ldots\ge\lambda_{nT}>0$. We approximate $\mathbf K_n$ using the leading $l$ eigenpairs ($l\ll nT$), that is, $\mathbf K_n\approx\mathbf U_l\mathbf \Lambda_l\mathbf U_l^\top$, where $\mathbf U_l \in \mathbb R^{nT\times l}$ contains the first $l$ eigenvectors and $\mathbf\Lambda_l=\mathrm{diag}(\lambda_1,\ldots,\lambda_l)$. The rank $l$ is selected by cumulative explained variance:
> $$l=\min_{l_0}\{l_0 \in \{1,\ldots,nT\}:\sum_{i=1}^{l_0} \lambda_i/\sum_{i=1}^{nT}\lambda_i\ge c\}.$$
> The Nyström method approximates $\mathbf{K}_n$ by selecting $n_l$ ($\ll nT$) landmark columns and extending a small eigendecomposition, which yields a computational cost of evaluating $(\mathbf K_n+\eta \mathbf I)^{-1}$ as $\mathcal{O}(nTn_l^2 +n_l^3)$ (Williams & Seeger 2000). Thus, the overall computational complexity for training TOGP becomes $\mathcal O\left((nTn_l^2 + n_l^3 + n^2 T^2 m_h)\log n\right)$. We have added it as a remark in Section 3.1 of our **rebuttal revision** (lines 252-261, page 5).
>
> Finally, we provide the prediction and optimization performance of TOGP based on the Nyström low-rank approximation (SVD-TOGP) under the three synthetic settings. The results are summarized below.
>
> |||Setting (1)||Setting (2)||Setting (3)||
> |-|-|-|-|-|-|-|-|
> |**Task**|**Criterion**|**SVD-TOGP**|**TOGP**|**SVD-TOGP**|**TOGP**|**SVD-TOGP**|**TOGP**|
> |**GP**|NLL|557.09|503.00|-18.39|-18.10|-3778.74|-3923.1|
> ||MAE|0.1756|0.1571|0.1099|0.1052|0.1436|0.1372|
> ||$\Vert Cov\Vert$|3.9627| 2.0200|0.6261|0.0400|0.0881|2.8200|
> ||Time (s)|**21.8**|266.04|**10.83**|69.85|**436.17**|900.13|
> |**BO**|MSE$_x$|0.0001|0.0000|0.0003|0.0003|0.0002|0.0001|
> ||MAE$_y$|0.0041|0.0008|0.0356|0.0350|0.0051|0.0050|
> ||Ins Regret|0.0040|0.0001|0.0002|0.0002|0.0402|0.0302|
> ||Time (s)|**831.02**|6968.25|**162.78**|1588.53|**3604.66**|6035.61|
> |**CBBO**|MSE$_x$|0.0024|0.0023|0.0000|0.0000|0.0035|0.0021|
> ||MAE$_y$|0.0180|0.0172|0.0000|0.0000|0.0246|0.0145|
> ||Acc|1|1|1|1|1|1|
> ||Ins Regret|1.0944|0.9807|0.0000|0.0000|1.8571|1.4406|
> ||Time (s)|**655.83**|8017.08|**134.22**|2348.16|**4446.12**|11699.09|
>
> **We can see that SVD-TOGP not only greatly reduces the computational time but also achieves higher accuracy than all baselines (see Tables 2-3 in our manuscript, lines 425-457 on pages 8-9).** Thus, the SVD-TOGP not only preserves predictive accuracy but also significantly improves computational efficiency. The results have been added in Appendix L of our **rebuttal revision** (lines 1372-1392, page 26).

---

> ### Author Response · Authors · 2025-11-21
> **Response to Questions 1-5**
>
> ***Question 1: Regarding the request to clarify whether the main contribution lies in GP modeling or in the BO methodology***
>
> **Response to Question 1:** Thank you for raising this question. We have answered this question in **Response to Weekness 1**.
>
> ---
> ***Question 2: Regarding the suggestion to strengthen the contribution to the BO literature in a revision***
>
> **Response to Question 2:** Thank you for your valuable suggestion. We clarify our main contributions compared to existing BO methods in Section 1 of our **rebuttal revision** (lines 92-99, page 2) as follows:
>
> Notably, compared to existing TOGP methods (Belyaev et al., 2015; Kia et al., 2018; Zhe et al.,2019), our model provides a more general kernel construction framework, as their tensor-output kernels correspond to specific choices of low-rank tensor decompositions, while our LMC-based formulation allows arbitrary tensor constraints to be incorporated into the coregionalization matrix. Compared to existing BO methods (Srinivas et al.,2009; Belakaria et al., 2019; Chowdhury & Gopalan, 2021), our work is the first to establish a BO framework for tensor outputs and further extend it to the proposed CBBO setting. Moreover, our contributions lie in deriving regret bounds for both TOBO and TOCBBO based on concentration inequalities tailored to the proposed TOGP under the Bayesian framework.
>
> ---
> ***Question 3: Regarding the request for empirical runtime or efficiency comparisons for TOGP and TOCBBO***
>
> **Response to Question 3:** Thank you for raising this question. We have answered this question in **Response to Weekness 3**.
>
> ---
> ***Question 4: Regarding the request to show how improved predictive accuracy translates into better optimization performance via ablation or correlation analysis***
>
> **Response to Question 4:** Thank you for raising this concern. Existing work on tensor outputs focuses only on GP modeling and does not address BO or CBBO. Therefore, in our numerical experiments and case studies, we combine sMTGP with the UCB criterion and the CMAB-UCB2 criterion for comparison. The sMTGP model assumes that the covariance structure of the tensor output is fully separable and uses a Kronecker product to define both inter-mode and input correlations. As a result, the baseline sMTGP-UCB can be regarded as an ablation study of our proposed TOBO and TOCBBO for evaluating optimization performance. Our numerical experiments and case studies consistently show that TOBO and TOCBBO achieve faster reductions in instantaneous regret and converge more rapidly than sMTGP-UCB.
>
> ---
>
> ***Question 5: Regarding the suggestion to extend the framework to more meaningful tensor structures such as spatiotemporal tensors***
>
> **Response to Question 5:** Thank you for your valuable comment. Our proposed TOGP framework can naturally model more meaningful tensor structures. Taking a spatiotemporal system as an example, the outputs can be represented as a two-mode tensor. Specifically, suppose $f(\boldsymbol x) \in \mathbb{R}^{k \times m}$ denotes the system dynamics evaluated at spatial locations $\boldsymbol{S}= \{s_1, \cdots, s_k\}$ and time points $\boldsymbol{T} = \{t_1, \cdots, t_m\}$. The proposed TOGP can be used to model $\mathbf f$ by capturing spatial and temporal dependencies simultaneously. The matrix $\operatorname{vec}(\mathbf A)\operatorname{vec}(\mathbf A)^{\top}$ in equations (6) and (7) describes these correlations across the output correlations, and the corresponding kernel hyperparameters are estimated through maximum likelihood. We have added a corresponding discussion in Section 6 of our **rebuttal revision** (lines 538–539, page 10).
>
> ---
>
> **References:**
>
> [1] Williams C, Seeger M. Using the Nyström method to speed up kernel machines[J]. Advances in neural information processing systems, 2000, 13.
>
> [2] Belyaev M, Burnaev E, Kapushev Y. Gaussian process regression for structured data sets[C]. International Symposium on Statistical Learning and Data Sciences. Cham: Springer International Publishing, 2015: 106-115.
>
> [3] Kia S M, Beckmann C F, Marquand A F. Scalable multi-task Gaussian process tensor regression for normative modeling of structured variation in neuroimaging data[J]. arXiv preprint arXiv:1808.00036, 2018.
>
> [4] Zhe S, Xing W, Kirby R M. Scalable high-order gaussian process regression[C]. The 22nd International Conference on Artificial Intelligence and Statistics. PMLR, 2019: 2611-2620.
>
> [5] Srinivas N, Krause A, Kakade S M, et al. Gaussian process optimization in the bandit setting: No regret and experimental design[J]. arXiv preprint arXiv:0912.3995, 2009.
>
> [6] Belakaria S, Deshwal A, Doppa J R. Max-value entropy search for multi-objective Bayesian optimization[J]. Advances in neural information processing systems, 2019, 32.
>
> [7] Chowdhury S R, Gopalan A. No-regret algorithms for multi-task bayesian optimization[C]. International Conference on Artificial Intelligence and Statistics. PMLR, 2021: 1873-1881.

---

### Official Review · Reviewer_6Rc1 · 2025-11-03

**Soundness:** 2
**Presentation:** 2
**Contribution:** 2
**Rating:** 4
**Confidence:** 4

**Summary:**

This paper proposes a novel Bayesian optimization framework for systems with tensor-valued outputs, a setting previously unaddressed in the BO literature. The core idea involves a Tensor-output Gaussian Process (TOGP) with two new kernel classes that capture complex, input-dependent correlations within the tensor structure, avoiding the limitations of simple vectorization. The authors also introduce a more challenging Combinatorial Bandit BO (CBBO) setting where only a subset of tensor elements contributes to the objective, and propose TOCBBO to handle it via a CMAB-UCB2 acquisition function that decouples continuous input selection from combinatorial super-arm selection. Theoretical guarantees are provided for the sub-linear regret of both methods. Extensive synthetic and real-world experiments demonstrate superior performance over baselines that use standard multi-output GPs.

**Strengths:**

1. This paper introduces the first Bayesian optimization framework specifically designed for tensor-output systems, developing novel tensor-output Gaussian processes that capture complex structural dependencies through specialized kernel designs.
2. This paper provides strong theoretical guarantees, establishing sublinear regret bounds for both the standard tensor-output setting and the more challenging combinatorial bandit scenario with partial observations.
3. The proposed methods show robust performance and consistently outperform existing approaches across diverse real-world applications.

**Weaknesses:**

1. This paper does not provide computational complexity comparisons with existing methods. Additionally, the high O(n^3T^3) complexity of TOGP itself represents a significant limitation, as strategies for scaling to large tensor outputs are not discussed.
2. The theoretical analysis relies on the assumption that the true function is a sample from the proposed TOGP. There is little discussion of whether this holds in practice or how to assess its validity in real-world applications.
3. A more detailed explanation of Equation (3) in line 161 would be helpful. For instance, it would be useful to clarify whether the kernel function K(x,x′) is designed to capture both similarities in the input space X and correlations among the internal elements of the output tensor.
4. One concern is that the modeling and algorithm design are claimed to target tensor-type data, yet the actual formulations (e.g., Equations (4), (5), and (8)) are carried out at the matrix level. Would this process compromise or distort the intrinsic tensor structure?

**Questions:**

See the weaknesses part above.

---

> ### Author Response · Authors · 2025-11-21
> **Response to Weakness 1 (Part A)**
>
> Thank you very much for your valuable suggestions. We have provided point-by-point responses to address all the raised questions and concerns. The **rebuttal revision** has also been submitted. All changes in the revised rebuttal manuscript have been highlighted in **red** for clarity.
>
> ---
>
> ***Weakness 1: Regarding the concern that the paper lacks computational complexity comparisons and does not address the scalability of the $\mathcal O(n^3 T^3)$ TOGP model***
>
> **Response to Weakness 1**:
> Thank you for raising this point. To provide a clearer comparison, we summarize the computational complexity of GP training, BO, and CBBO for the baseline methods sMTGP, MLGP, and MVGP, together with our proposed method, in the table below.
> |Method|GP Training|BO (Round n)|CBBO (Round n)|
> |-|-|-|-|
> |TOGP|$\mathcal O\left((n^3T^3+n^2T^2m_h)\log n\right)$|$\mathcal O\left((n^3T^3+n^2T^2m_h)\log n\right)$|$\mathcal O((n^3k^3+n^2kTm_h)\log n+kT^3)$|
> |sMTGP| $\mathcal O\left((n^3+\sum_{l=1}^m t_l^3+n^2m_{h}+\sum_{l=1}^mt_l^2m_{hl})\log n\right)$ | $\mathcal O\left((n^3+\sum_{l=1}^m t_l^3+n^2m_{h}+\sum_{l=1}^mt_l^2m_{hl})\log n\right)$ |$\mathcal O\left((n^3+k^3+n^2m_{h}+k\sum_{l=1}^mt_lm_{hl})\log n+k\sum_{l=1}^mt_l^3\right)$ |
> |MLGP| $\mathcal O\left((n^3+\sum_{l=1}^m t_l^3+n^2m_{h}+\sum_{l=1}^mt_l^2m_{hl})\log n\right)$ | $\mathcal O\left((n^3+\sum_{l=1}^m t_l^3+n^2m_{h}+\sum_{l=1}^mt_l^2m_{hl})\log n\right)$ |$\mathcal O\left((n^3+k^3+n^2m_{h}+k\sum_{l=1}^mt_lm_{hl})\log n+k\sum_{l=1}^mt_l^3\right)$|
> |MVGP| $\mathcal O\left((n^3+T^3+n^2m_{h1}+T^2m_{h2})\log n\right)$ | $\mathcal O\left((n^3+T^3+n^2m_{h1}+T^2m_{h2})\log n\right)$ |$\mathcal O\left((n^3+T^3+n^2m_{h1}+kTm_{h2})\log n+kT^3\right)$|
>
> Here $m_h$, $m_{h1}$, $m_{h2}$, and $m_{hl}$ for $l=1,\cdots,m$ denote the number of hyperparameters for the corresponding methods.
>
> Furthermore, we provide the runtime (seconds) for GP training of different methods under the three synthetic settings used in our numerical experiments. The results are summarized below.
> ||TOGP|sMTGP|MLGP|MVGP|
> |-|-|-|-|-|
> |**Setting (1)**|266.04|27.26|30.57|**6.50**|
> |**Setting (2)**|69.85|16.08|21.99|**1.31**|
> |**Setting (3)**|900.13|851.34|891.73|**769.50**|
>
> We also provide the runtime (seconds) of different methods for BO and CBBO under the three synthetic settings. The results are summarized below.
> ||Method|TOBO/TOCBBO|TOGP-RS|sMTGP-UCB|sMTGP-RS|MLGP-UCB|MLGP-UCB|MVGP-UCB|MVGP-UCB|
> |--|--|--|--|--|--|--|--|--|--|
> |**Setting (1)**|BO|6968.25|6176.84|1516.31|1503.24|1545.41|1530.32|340.44|**334.79**|
> ||CBBO|8017.08|7990.44|953.82|939.27|981.86|971.25|437.29|**386.86**||
> |**Setting (2)**|BO|1588.53|1573.36|117.78|106.93|133.93|109.14|14.86|**7.70**||
> ||CBBO|2348.16|2338.97|179.11|164.95|185.49|180.70|20.76|**6.70**||
> |**Setting (3)**|BO|6035.61|5786.02|5291.26|5273.84|5452.90|5252.32|3006.97|**2997.75**|
> ||CBBO|11699.09|11501.01|9436.90|9421.08|9696.60|9431.96|3199.47|**3116.23**|
>
> We have included these computational results in Appendix L of our **rebuttal revision** (lines 1323-1371, pages 25-26). It can be observed that the proposed TOGP has the longest training time among all methods. This is mainly due to two reasons. First, in our numerical experiments, we use the non-separable tensor-output kernel introduced in Definition 1 to build a more general TOGP model, which leads to a computational complexity of $O(n^3 T^3)$. In TOBO and TOCBBO, since the hyperparameters are updated at each round, the cost of selecting the query point at round $n$ is also $O(n^3 T^3)$. In contrast, the baseline methods adopt kernels with different levels of separability, where the covariance matrix can be represented using Kronecker products, resulting in lower computational cost. Second, all experiments were conducted on a Windows workstation equipped with an Intel Core i9-9960X CPU (16 cores) and 128 GB RAM. With a more powerful server, the running time would be further reduced.
>
> Despite this, the computational cost of TOGP remains affordable. In Setting (3), where the tensor dimension is $4 \times 5 \times 2$, the total computation time for sequentially adding $30$ new points on top of $20$ initial design points is 6035.61 seconds. More importantly, TOGP, TOBO, and TOCBBO achieve significantly better fitting and optimization performance than all baselines (see Tables 2 and 3, lines 425–457 on page 8). **The additional computational cost is therefore well justified by the significant improvements in accuracy.** Moreover, BO is typically applied to expensive black-box optimization scenarios. In such settings, the cost of data acquisition often far exceeds the computational cost of training the surrogate model, and the optimization performance is far more critical than the running time.

---

> ### Author Response · Authors · 2025-11-21
> **Response to Weakness 1 (Part B) and Weakness 2**
>
> **For large tensor outputs where the computational cost may become prohibitive, we have also introduced a Nyström low-rank approximation strategy to improve the scalability of TOGP.** Specifically, based on data $\mathbf{X_n}$ and $\boldsymbol Y_n$, consider the spectral decomposition $\mathbf K_n=\mathbf U_n\mathbf\Lambda_n\mathbf U_n^\top$, where $\mathbf\Lambda_n = \mathrm{diag}(\lambda_1,\lambda_2,\ldots,\lambda_{nT})$ denotes eigenvalues ordered as $\lambda_1\ge\lambda_2\ge\ldots\ge\lambda_{nT}>0$. We approximate $\mathbf K_n$ using the leading $l$ eigenpairs ($l\ll nT$), that is, $\mathbf K_n\approx\mathbf U_l\mathbf \Lambda_l\mathbf U_l^\top$, where $\mathbf U_l \in \mathbb R^{nT\times l}$ contains the first $l$ eigenvectors and $\mathbf\Lambda_l=\mathrm{diag}(\lambda_1,\ldots,\lambda_l)$. The rank $l$ is selected by cumulative explained variance:
> $$l=\min_{l_0}\{l_0 \in \{1,\cdots,nT\}:\sum_{i=1}^{l_0} \lambda_i/\sum_{i=1}^{nT}\lambda_i\ge c\}.$$
> The Nyström method approximates $\mathbf{K}_n$ by selecting $n_l$ ($\ll nT$) landmark columns and extending a small eigendecomposition, which yields a computational cost of evaluating $\left(\mathbf{K}_n+\eta \mathbf{I}\right)^{-1}$ as $\mathcal{O}(nTn_l^2 +n_l^3)$ (Williams & Seeger 2000). Thus, the overall computational complexity for training TOGP becomes $\mathcal{O}\left((nT n_l^2 + n_l^3 + n^2 T^2 m_h)\log n\right)$. We have added it as a remark in Section 3.1 of our **rebuttal revision** (lines 252-261, page 5).
>
> Finally, we provide the prediction and optimization performance of TOGP based on the Nyström low-rank approximation (SVD-TOGP) under the three synthetic settings. The results are summarized below.
>
> |||Setting (1)||Setting (2)||Setting (3)||
> |-|-|-|-|-|-|-|-|
> |**Task**|**Criterion**|**SVD-TOGP**|**TOGP**|**SVD-TOGP**|**TOGP**|**SVD-TOGP**|**TOGP**|
> |**GP**|NLL|557.09|503.00|-18.39|-18.10|-3778.74|-3923.1|
> ||MAE|0.1756|0.1571|0.1099|0.1052|0.1436|0.1372|
> ||$\Vert Cov\Vert$|3.9627| 2.0200|0.6261|0.0400|0.0881|2.8200|
> ||Time (s)|**21.8**|266.04|**10.83**|69.85|**436.17**|900.13|
> |**BO**|MSE$_x$|0.0001|0.0000|0.0003|0.0003|0.0002|0.0001|
> ||MAE$_y$|0.0041|0.0008|0.0356|0.0350|0.0051|0.0050|
> ||Ins Regret|0.0040|0.0001|0.0002|0.0002|0.0402|0.0302|
> ||Time (s)|**831.02**|6968.25|**162.78**|1588.53|**3604.66**|6035.61|
> |**CBBO**|MSE$_x$|0.0024|0.0023|0.0000|0.0000|0.0035|0.0021|
> ||MAE$_y$|0.0180|0.0172|0.0000|0.0000|0.0246|0.0145|
> ||Acc|1|1|1|1|1|1|
> ||Ins Regret|1.0944|0.9807|0.0000|0.0000|1.8571|1.4406|
> ||Time (s)|**655.83**|8017.08|**134.22**|2348.16|**4446.12**|11699.09|
>
> **We can see that SVD-TOGP not only greatly reduces the computational time but also achieves higher accuracy than all baselines (see Tables 2-3 in our manuscript, lines 425-457 on pages 8-9).** Thus, the SVD-TOGP not only preserves predictive accuracy but also significantly improves the computational efficiency. The results have been added in Appendix L of our **rebuttal revision** (lines 1372-1392, page 26).
>
> ---
>
> ***Weakness 2: Regarding the assumption that the true function is a sample from the proposed TOGP and the lack of discussion on its practical validity***
>
> **Response to Weakness 2**: Thank you for raising this question. We fully understand your concern. In real-world applications, the true function is a black box and does not necessarily follow a GP model. However, **assuming that the true function follows a TOGP model is a standard Bayesian assumption used in the literature on Bayesian optimization and bandit problems** (Srinivas et al. 2009; Accabi et al. 2018). This assumption provides a sufficient condition for establishing concentration inequalities and proving sublinear regret bounds, which characterize the theoretical behavior of the algorithm.
>
> In practice, **even when the true function does not follow a TOGP model, such as in the three synthetic settings used in our experiments, the posterior mean still provides accurate predictions and the posterior variance remains well calibrated** (see Table 2, lines 425–431 on page 8). Furthermore, both TOBO and TOCBBO achieve rapidly decreasing instantaneous regret and consistently outperform all baselines (see Table 3, lines 441–457 on page 8). Therefore, the GP assumption is used solely for theoretical analysis and does not restrict the practical applicability of our methods.

---

> ### Author Response · Authors · 2025-11-21
> **Response to Weaknesses 3-4**
>
> ***Weakness 3: Regarding the request for a clearer explanation of Equation (3)***
>
> **Response to Weakness 3:** Thank you for your suggestion. The proposed $\mathbf K(\boldsymbol x,\boldsymbol x')$ can capture both the correlation across inputs and the correlation within the tensor output. We have added a more detailed explanation of the structure of $\mathbf K(\boldsymbol x,\boldsymbol x')$ after introducing the non-separable and separable tensor-output kernels in Section 3.1 of our **rebuttal revision** (lines 222-227, page 5) as follows:
>
> Note that $\mathbf K(\boldsymbol x,\boldsymbol x')$ is designed to capture both the correlation across inputs and the correlation within the tensor output. For the non-separable tensor-output kernel in (6), the base kernel $k_{lj}(\boldsymbol x,\boldsymbol x')$ models the covariance between the inputs $\boldsymbol x$ and $\boldsymbol x'$, while the matrix $\operatorname{vec}(\mathbf A_l)\operatorname{vec}(\mathbf A_l)^{\top}$ describes the covariance structure among the elements of the output tensor. For the separable tensor output kernel in (7), $k(\boldsymbol x,\boldsymbol x')$ captures the input correlation, and $\operatorname{vec}(\mathbf A)\operatorname{vec}(\mathbf A)^{\top}$ specifies the mode-wise covariance structure within the tensor output.
>
> ---
>
> ***Weakness 4: Regarding the concern that the matrix-level formulations in Equations (4), (5), and (8) may compromise the intrinsic tensor structure***
>
> **Response to Weakness 4:** Thank you for your insightful comment. For GP modeling with tensor outputs, the covariance structure must be specified through a matrix-valued kernel together with appropriate tensor structural constraints, such as low-rank assumptions. In fact, all existing tensor-output GP models directly vectorize the tensor and build the covariance structure using a low-rank tensor decomposition, such as CP and Tucker methods (Belyaev et al. 2015; Kia et al. 2018; Zhe et al. 2019). For example, Kia et al. (2018) assumes that the covariance matrix within each mode is separable under a Tucker decomposition to build a tensor-output GP.
>
> In our work, we propose more general classes of tensor-output kernels based on the linear model of coregionalization (LMC). Under the LMC framework, the tensor structure is encoded in the coregionalization matrix $\operatorname{vec}(\mathbf{A}_l)\operatorname{vec}(\mathbf{A}_l)^\top$, $l=1,\cdots,m$. By imposing tensor constraints on $\mathbf{A}_l$, such as CP or TT decompositions, the resulting kernel naturally preserves the desired tensor structures while reducing the number of hyperparameters. Compared with Belyaev et al. (2015), Kia et al. (2018), and Zhe et al. (2019), our model provides a more general kernel construction framework, as their tensor-output kernels correspond to specific choices of low-rank tensor decompositions, while our LMC-based formulation allows arbitrary tensor constraints to be incorporated into the coregionalization matrix.
>
> Therefore, the proposed kernel preserves the intrinsic tensor structure while offering a structured framework for building matrix-valued kernels in which the coregionalization matrix is explicitly constrained to satisfy a chosen tensor decomposition. This leads to a coherent and flexible modeling framework for tensor-valued outputs.
>
> ---
>
> **References:**
>
> [1] Williams C, Seeger M. Using the Nyström method to speed up kernel machines[J]. Advances in neural information processing systems, 2000, 13.
>
> [2] Srinivas N, Krause A, Kakade S M, et al. Gaussian process optimization in the bandit setting: No regret and experimental design[J]. arXiv preprint arXiv:0912.3995, 2009.
>
> [3] Accabi G M, Trovo F, Nuara A, et al. When gaussian processes meet combinatorial bandits [C]. 14th European Workshop on Reinforcement Learning. 2018: 1-11.
>
> [4] Belyaev M, Burnaev E, Kapushev Y. Gaussian process regression for structured data sets[C]. International Symposium on Statistical Learning and Data Sciences. Cham: Springer International Publishing, 2015: 106-115.
>
> [5] Kia S M, Beckmann C F, Marquand A F. Scalable multi-task Gaussian process tensor regression for normative modeling of structured variation in neuroimaging data[J]. arXiv preprint arXiv:1808.00036, 2018.
>
> [6] Zhe S, Xing W, Kirby R M. Scalable high-order gaussian process regression[C]. The 22nd International Conference on Artificial Intelligence and Statistics. PMLR, 2019: 2611-2620.

---

### Author Response · Authors · 2025-12-01
**Summary**

We sincerely appreciate the time, effort, and thoughtful feedback provided by the reviewers (**6Rc1, 9BYU, AnPz, f43k, uCf6**) in evaluating our work. We are encouraged by the recognition that **TOGP is novel** in capturing complex structural dependencies through a specialized tensor-output kernel (**6Rc1, 9BYU, AnPz, f43k, uCf6**), that **TOCBBO is novel** in addressing the CBBO problem with partial observations (**AnPz, uCf6**), and that both TOBO and TOCBBO are supported by solid theoretical guarantees with sublinear regret bounds (**6Rc1, f43k, uCf6**).

---

**1. Consensus on Strengths**
* **TOGP kernel design.** All reviewers agreed that TOGP introduces a novel kernel construction that effectively models tensor-output functions and captures complex structural dependencies.

* **CBBO formulation.** Reviewers **9BYU**, **AnPz**, and **uCf6** highlighted the novelty of the TOCBBO framework.

* **Theoretical guarantees.** Reviewers **6Rc1**, **f43k**, and **uCf6** emphasized that TOBO and TOCBBO offer strong theoretical guarantees with sublinear regret bounds.

* **Empirical performance.** Reviewers **6Rc1**, **AnPz**, **f43k**, and **uCf6** noted that the proposed methods consistently outperform baselines across both synthetic and real datasets.

---

**2. Response to Common Concerns**
* **Computational Cost (6Rc1, 9BYU, f43k):**
  * We added a comparison of the asymptotic complexity analyses of all baselines.
  * We compared the empirical runtime of all methods in three synthetic experiments to evaluate overall performance. Although our method has the longest runtime, it remains affordable. The additional computational cost is justified by the significant improvement in accuracy.
  * We further proposed a more scalable TOGP with reduced computational cost. The improved TOGP preserves predictive accuracy and significantly improves computational efficiency, achieving the second fastest runtime among all methods.

* **Scalability of TOGP (6Rc1, f43k, uCf6):**
  * We introduced a Nyström low-rank approximation strategy to improve the scalability of TOGP, referred to as SVD-TOGP, which reduces the training complexity from $\mathcal O(n^3T^3)$ to $\mathcal O(nT n_l^2 + n_l^3)$ with $n_l \ll nT$.
  * We compared the empirical performance of SVD-TOGP with all baselines. The results show that SVD-TOGP greatly reduces computational cost, runs faster than sMTGP and MLGP, and achieves higher accuracy than all baselines.

* **Expanded Baselines (f43k, uCf6):** We added three additional baselines (single-GP, single-objective GP, and non-separable multi-output GP) to compare with our method in three synthetic experiments. The results show that all three baselines perform worse than the proposed method.

* **Comparison with MOBO (AnPz, f43k):** We clarified that MOBO methods based on Pareto optimality are limited to output dimensions less than 4 due to prohibitive computational complexity and cannot be applied to tensor-output BO frameworks.

---

**3. Summary of Revisions**
* **Motivation and contributions:** The rebuttal revision clarifies our contributions more clearly and highlights the novelty of our method in **Section 1**. We also added a statement on the limitations of existing MOBO methods for tensor outputs in **Section 2**.

* **Method Details:** The rebuttal revision adds details on the tensor-output kernel $\mathbf K(\cdot,\cdot)$, the rank selection criterion for tensor $\mathbf A$, and the Nyström low-rank approximation strategy in **Section 3**.

* **Discussion:** The rebuttal revision adds further discussion in **Section 7**.

* **Expanded Experimental Results:** The rebuttal revision adds a comparison of the computational complexity and empirical runtime for the baselines in **Appendix L**, results for the added baselines (single-GP, single-objective GP, non-separable multi-output GP), and a sensitivity analysis of $L_f$ and $H_f$ in **Appendix M**.

---

**4. Response from Reviewers**

* Reviewer **AnPz**, who had reviewed an earlier version of our work submitted to another conference, highlighted the novelty of the proposed TOGP and TOCBBO for addressing CBBO. The reviewer also noted that the experimental results have improved, **leading to an inclination toward accepting the paper**.

* Only Reviewer **uCf6** provided a follow-up response after reading our rebuttal and confirmed that all concerns raised in the initial review were fully addressed. The reviewer stated that the score would be left at 6 as a neutral rating due to limited expertise and emphasized that this should not be interpreted as a judgment of the work. Reviewer uCf6 further noted that **the rating should be viewed as a weak accept rather than a weak reject** given the quality of the presentation, writing, and the thoroughness of our responses.

---

We believe that the revisions prompted by the reviewers' feedback have substantially improved the clarity, rigor, and overall quality of our work.

---

### Meta-Review · Area_Chair_FZMK · 2026-01-07

**Summary:**

This paper proposes a BO method for tensor-output function with practical algorithms. Overall, the setting is novel although the technical contribution largely builds on previous work. Having said that, the practical significance is noteworthy and I'd not view its technical incremental as a show stopper. Most reviewers are also largely in favor of accepting this paper. Reviewer 6Rc1 raised 4 weaknesses (computational complexity, assumption that black-box function follows a GP, inquiries for additional clarification regarding Eq. (3) and the matrix vs tensor output setup in the experiment). The authors have provided very comprehensive rebuttal. I have read the rebuttal and believe it addresses all these concerns. So overall, I don't see any critical concern left for this paper. Accordingly, I recommend acceptance. The authors are encouraged to incorporate all rebuttal materials into the final version. Particularly the materials in response to Reviewer 6Rc1.

**Reviewer Concerns:**

As I detailed above, most reviewers were positive and the rebuttal in my own reading has addressed all concerns. The only review leaning on rejection is from Reviewer 6Rc1 but these weaknesses are largely editorial in nature and have been well-addressed. I believe Reviewer 6Rc1 would agree with my assessment and increase the score accordingly to acceptance.

**Reviewer Scores:**

The original scores are 8664. The rebuttal is effective and comprehensive in my own reading. I expect the reviewer who gave a 4 would upgrade. Hence, this is a clear acceptance case.

---

### Decision · Program_Chairs · 2026-01-26

Accept (Poster)